# *TREAT*: A Code LLMs Trustworthiness / Reliability Evaluation and Testing Framework

## Abstract

Large foundation models are fundamentally transforming the software engineering landscape, demonstrating exceptional capabilities across diverse tasks such as code generation, debugging, and testing. Despite this rapid progress, a significant gap remains in how to comprehensively evaluate these models' trustworthiness in real-world software engineering scenarios. Existing benchmarks suffer from limited task scope and fail to incorporate critical evaluation aspects such as the robustness and reliability of models. To bridge this gap, we present an evaluation framework called **TREAT** (Code LLMs **T**rustworthiness / **R**eliability **E**valuation **A**nd **T**esting) that provides a holistic assessment of model performance in code intelligence tasks. Our evaluation framework addresses key limitations in existing approaches with four main improvements: (1) Multi-Task Holistic Evaluation that spans diverse software engineering activities rather than limited coding tasks; (2) Multi-Language and Multi-Modality Assessment that extends beyond traditional single-language, text-only benchmarks to include multi-modality coding tasks; (3) Robustness Assessment that evaluates model reliability under semantically-preserving code transformations; and (4) Rigorous Evaluation Methodology that enhances the trustworthiness of evaluation results through diverse evaluation prompts and adaptive solution extraction. Based on this evaluation framework, we assess 26 state-of-the-art models and uncover both their strengths and limitations, yielding several key insights: ❶ Current models show substantial performance variation across programming tasks, especially on tasks like code review and vulnerability detection; ❷ Multi-modal language models demonstrate specific performance limitations in UI code generation and edit; ❸ Existing models exhibit severe robustness issues on coding tasks; ❹ Our multi-prompt evaluation method can mitigate potential evaluation bias from single prompts and obtain more reliable results. Our project page is available at https://code-treat.vercel.app/.

## 1 Introduction

The landscape of software engineering is being fundamentally reshaped by large foundation models, particularly Large Language Models (LLMs) and Multimodal Large Language Models (MLLM) (Hou et al., 2024; Lyu et al., 2025). These models can understand natural language instructions and convert them into executable code, bridging the gap between human intent and software implementation. Advanced models like OpenAI's GPT series (Hurst et al., 2024) and Anthropic's Claude (Anthropic, 2024) have demonstrated remarkable proficiency across diverse software engineering tasks, from code generation and debugging (Li et al., 2024b; Wang et al., 2025a) to documentation and testing (Gao et al., 2023; Xie et al., 2023). This evolution is driving the development of intelligent tools that are transforming software engineering practices. As these models become increasingly integrated into critical software development workflows, understanding their trustworthiness and reliability has become increasingly critical.

Despite these impressive achievements, the rapid advancement of LLMs in software engineering has created substantial challenges for model evaluation. Although numerous models have emerged in both academia and industry, there is a lack of comprehensive evaluation methodologies that can assess model capabilities across diverse real-world software engineering scenarios. Existing evaluation approaches (Jain et al., 2025; Yang et al., 2024d) are often constrained to narrow, task-specific

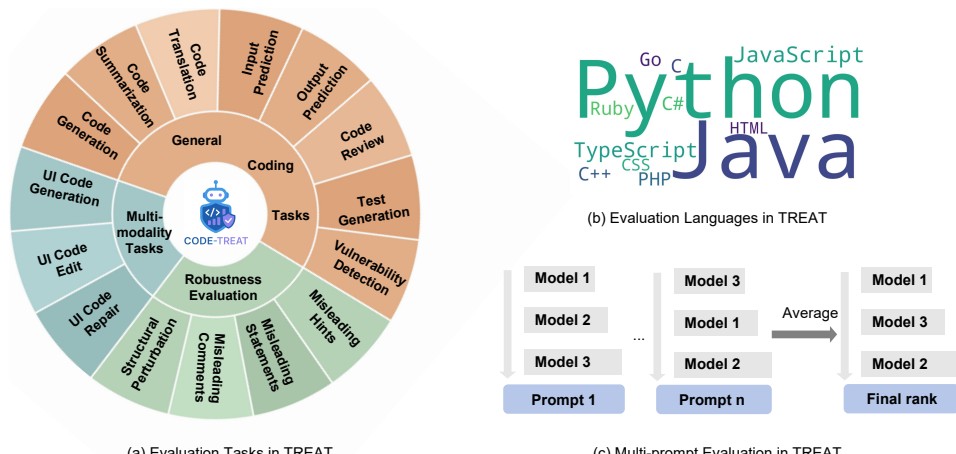

Figure 1: Overview of the TREAT evaluation framework.

benchmarks that fail to capture the complexity and diversity of practical software development work-flows. Specifically, these benchmarks lack assessments for some critical software quality assurance tasks such as code review and code vulnerability detection. Moreover, existing benchmarks often focus solely on single-modal and normal inputs, failing to incorporate important aspects such as multi-modality processing capabilities and the mode's robustness and reliability. These gaps make it difficult to assess model's trustworthiness in real-world development scenarios, posting major challenges for researchers and practitioners to determine optimal model selection for specific software engineering scenarios.

To address the challenges, we present TREAT, the first holistic evaluation framework for LLMs in code intelligence tasks. It tackles the aforementioned problems with the following features: ❶ **Multi-Task Holistic Evaluation.** Unlike existing benchmarks that focus on narrow and task-specific assessments such as code generation, as shown in Figure 1 (a), TREAT provides a holistic benchmark spanning the software engineering activities in the development lifecycle. It encompasses multiple task categories, which enables researchers to assess model capabilities across diverse scenarios. ❷ **Multi-Language and Multi-Modality Assessment.** TREAT expands evaluation scope beyond traditional single-language, text-only benchmarks. As shown in Figure 1 (b), our framework systematically evaluates models across multiple programming languages and incorporates multi-modality tasks that bridge visual design and software implementation. We incorporate tasks such as UI code generation and edit, which are essential given the multimodal environment of modern software development environments. ❸ **Robustness Evaluation.** Considering the importance of trustworthy Code LLMs in software engineering, as shown in Figure 1 (a), TREAT also incorporates systematic robustness evaluation through various code transformation methods, which evaluates model stability under semantically-preserving perturbations. ❹ **Rigorous Evaluation Methodology.** We establish a rigorous evaluation methodology that enhances the fairness and reliability of the evaluation results. As shown in Figure 1 (c), we employ a multi-prompt evaluation strategy to reduce potential evaluation bias. Additionally, we employ an adaptive answer extraction method to better align benchmark evaluation with real-world developer usage.

Based on our evaluation framework, we have assessed 26 state-of-the-art models including both open-source and commercial models across different sizes. Based on this study and following analysis, we present the following novel empirical findings:

1. Current state-of-the-art models exhibit substantial performance variation and specialization across different programming tasks (Figure 2), with no single model achieving consistent best performance across all coding scenarios.

2. MLLMs exhibit different performance bottlenecks across different UI tasks, with UI code generation primarily limited by syntactic compilation issues while code edit and repair tasks are constrained by insufficient visual understanding and precise modification capabilities.

3. Existing large language models exhibit severe robustness issues on coding tasks, with an average performance decline of 14.1% under semantically-preserving code perturbations.

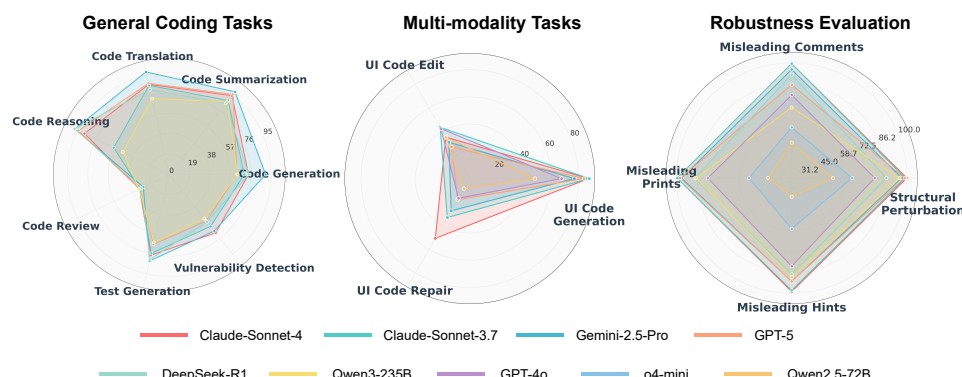

Figure 2: Performance comparison of leading models on TREAT. The results of multi-modality tasks are normalized for visualization.

4. Our multi-prompt evaluation method can effectively mitigate evaluation bias caused by single prompts, providing more reliable and trustworthy assessment results.

The main contributions of this paper can be summarized as follows:

1. **Comprehensive Benchmark.** We introduce TREAT, the first holistic evaluation framework spanning the software development lifecycle. It encompasses over 10+ tasks and languages, enabling a comprehensive assessment of LLM's generalization capabilities across diverse settings.

2. **Holistic and Rigorous Evaluation.** We establish a holistic evaluation methodology that incorporates multi-modality assessment and robustness evaluation through semantically-preserving code transformations. The evaluation process employs multiple diverse prompts to reduce potential evaluation bias.

3. **Empirical Analysis.** Through evaluation of 25+ state-of-the-art models, we reveal novel findings such as significant performance variations across tasks and unreliable performance under robustness assessment.

## 2 RELATED WORK

### 2.1 LARGE LANGUAGE MODELS FOR CODE

Large language models (LLMs) for code have rapidly advanced tasks such as code generation, completion, and reasoning. Several prominent models have emerged in this domain. For example, OpenAI's GPT series has garnered recognition for its proficiency in code generation and debugging capabilities, while Google's Gemini models excel at tackling complex algorithmic problems. Anthropic's Claude (Anthropic, 2024) has achieved impressive performance, exhibiting exceptional aptitude for tasks demanding sophisticated code reasoning. More recent models like DeepSeek-V3 (DeepSeek-AI et al., 2025b) and DeepSeek-R1 (DeepSeek-AI et al., 2025a) have reached performance levels that rival leading closed-source models. Qwen3 (Yang et al., 2025a) series features powerful agentic coding capabilities and is designed to handle complex software development workflows.

### 2.2 CODE INTELLIGENCE EVALUATION FOR LARGE LANGUAGE MODELS

The evaluation of Code LLMs has undergone significant evolution, transforming from simple code generation benchmarks such as HumanEval (Chen et al., 2021) and MBPP (Austin et al., 2021) to more sophisticated and realistic benchmarks. For example, LiveCodeBench (Jain et al., 2025) deals with the data contamination problem through the use of contemporary contest problems; Big-CodeBench (Zhuo et al., 2024) focuses on library-aware code generation capabilities. Although some recent benchmarks have expanded to include additional evaluation tasks, they remain constrained in scope and scale. Different from these benchmarks, our TREAT evaluation framework provides a holistic evaluation of model performance encompassing multi-language support, multi-task evaluation, multi-modality capabilities, and robustness assessment.

Table 1: Comparison with existing evaluation benchmarks.

| Benchmark | Size | Languages | Evaluation Tasks | Multi Prompt | Multi Modality |
|---|---|---|---|---|---|
| MBPP (Austin et al., 2021) | 378 | Python | Code Gen | ✗ | ✗ |
| HumanEval (Chen et al., 2021) | 164 | Python | Code Gen | ✗ | ✗ |
| LiveCodeBench (Jain et al., 2025) | 1,055 | Python | Code Gen, Input Pre, Output Pre, Code Rep | ✗ | ✗ |
| BigcodeBench (Zhuo et al., 2024) | 1,140 | Python | Code Gen | ✗ | ✗ |
| FullstackBench (Cheng et al., 2024) | 3,374 | 16 languages | Code Gen | ✗ | ✗ |
| CoCo-Bench (Yin et al., 2025) | 705 | Python, Java, C++, SQL | Code Gen, Code Rev, Code Und, Code Mod | ✗ | ✗ |
| AutoCodeBench (Chou et al., 2025) | 3,920 | 20 languages | Code Gen | ✗ | ✗ |
| SWE-Bench Multimodal (Yang et al., 2024c) | 517 | JavaScript | Issue Resolution | ✗ | ✓ |
| DyCodeEval (Chen et al.) | 8,070 | Python | Code Gen | ✓ | ✗ |
| **TREAT** | **9,908** | **12 languages** | **Code Gen, Code Rev, Test Gen, etc. (10+ tasks)** | ✓ | ✓ |

## 3  TREAT BENCHMARK CONSTRUCTION METHODOLOGY

To comprehensively evaluate code intelligence tasks, we construct our TREAT evaluation framework with a generic methodology based on the software development lifecycle. As shown in Table 1, compared with existing benchmarks, TREAT encompasses over 10 evaluation tasks and is the only benchmark that employs multiple-prompt evaluation, multi-modality capabilities assessment, and robustness evaluation.

As illustrated in Figure 3, our benchmark construction process employs a structured pipeline that begins with data collection from diverse sources. This raw data undergoes filtering and systematic metric design processes to provide a rigorous and comprehensive evaluation for each task. The TREAT benchmark encompasses three key components. The General Coding Tasks Evaluation (Section 3.1) assesses fundamental software development capabilities across seven core areas including code generation, code summarization, code translation, code reasoning, code review, test generation, and vulnerability detection. The Multi-Modality Tasks Evaluation (Section 3.2) extends beyond traditional text-based programming to evaluate capabilities in UI-based code generation, edit and repair tasks. Finally, the Robustness Evaluation Tasks (Section 3.3) assesses various models' reliability under various code transformation methods such as program structure transformation and providing misleading comments. We present the core workflow in building the benchmark in this section, and the detailed construction process for each task can be found in the Appendix A

### 3.1  GENERAL CODING TASKS EVALUATION

#### 3.1.1  DATA COLLECTION AND SELECTION

For general coding tasks, we construct our evaluation benchmark across seven important software engineering activities from software development to quality assurance. Our dataset spans multiple programming languages, with primary focus on Python and Java for most tasks, while extending to ten popular languages (C, C++, C#, Go, Java, JavaScript, TypeScript, PHP, Python, Ruby) for code summarization and code review to provide broad applicability.

To provide a comprehensive evaluation while reducing annotation costs, we employ a hybrid data crawling strategy that combines automated crawling from GitHub repositories and coding platforms with resampling from established public datasets. For tasks that can be automatically crawled and annotated, we actively crawl from public and continuously updated coding platforms and GitHub repositories, enabling to capture the most recent and comprehensive evaluation data; while for datasets that require manual checks or annotations, we sample data from recent representative benchmarks. For data collection, we collect high-quality data from two primary categories of reliable sources: continuously updated competitive coding platforms (e.g., GEEKSFORGEEKS, HACKER-RANK) that provide a large volume of algorithmic problems for evaluating model performance on algorithmic reasoning tasks, and GitHub repositories meeting strict quality criteria (≥100 stars and permissive open-source licenses such as Apache-2.0 and MIT) to gather real-world code samples

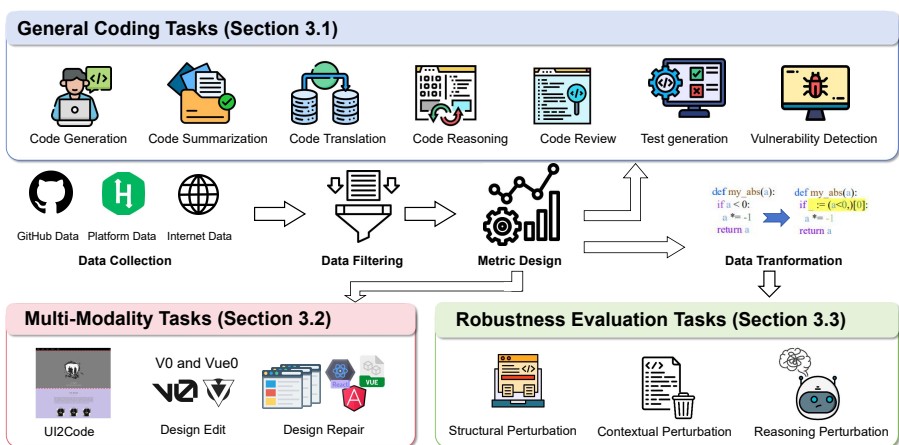

Figure 3: Evaluation methods of TREAT.

reflecting practical software development scenarios. To guarantee data integrity and eliminate redundant or low-quality samples, we first eliminate exact duplicates through string matching to avoid data redundancy, then follow established data cleaning method in each task to apply task-adaptive cleaning pipelines for filtering out irrelevant or invalid data (e.g., syntax errors, incomplete code snippets), and for selected aggregated benchmarks, we verify that they have undergone rigorous filtering and deduplication in their original publications to ensure reliability.

### 3.1.2 SCENARIO-SPECIFIC DATA COLLECTION METHODS

**Code Generation (CG):** For code generation, we utilize the algorithmic problems from GEEKS-FORGEEKS and HACKERRANK, with data up to 2025 and spanning easy, medium, and hard difficulty levels. We augment their existing test cases following the EvalPlus (Liu et al., 2023) methodology, with all generated test cases validated against ground-truth solutions to ensure accuracy.

**Code Summarization (CS):** For code summarization, we leverage the crawled GitHub repositories and use the Tree-sitter (Tree-sitter, 2025) parser to extract function-docstring pairs from each file. Then we apply Shi et al. (Shi et al., 2022)'s data cleaning methods to remove noisy samples.

**Code Translation (CT):** We focus on Python-Java bidirectional translation using our collected GEEKSFORGEEKS problems and the existing PolyHumanEval datasets. We also augment the test suites following the EvalPlus (Liu et al., 2023) to ensure rigorous evaluation.

**Code Reasoning (CR):** For code reasoning, we follow previous work (Gu et al., 2024) and create two sub-tasks *input prediction* and *output prediction*. We leverage the crawled problems from GEEKSFORGEEKS and HACKERRANK and employ Tree-sitter to mask function names and generate candidate input-output-assertion triples for both sub-tasks.

**Code Review (CRv):** To construct a real-world code review dataset, we use the crawled GitHub repositories and extract diff hunk and review comment pairs from each pull request. We follow the filtering criteria in previous work (Li et al., 2022) to remove noisy review comments and construct evaluation data.

**Test Generation (TG):** For test generation evaluation, we follow SYMPROMPT (Ryan et al., 2024) and leverage its context augmentation technique on 24 projects used in CODAMOSA (Lemieux et al., 2023) to construct our dataset.

**Vulnerability Detection (VD):** For vulnerability detection, we adopt the PRIMEVUL benchmark (Ding et al., 2024) containing 6,968 expert-verified vulnerable functions and 228,800 benign functions across 140 CWEs.

### 3.2 MULTI-MODALITY BENCHMARK CONSTRUCTION

Evaluating code models on multi-modality tasks is crucial for understanding their ability to interpret and generate code from diverse input formats such as images or layouts. We evaluate code mod-

els on multi-modality tasks using the data from the DESIGNBENCH datasets (Xiao et al., 2025). It encompasses three tasks collected through GitHub repository mining of framework-based websites and analysis of top global websites, combined with real-world user modification requests from development platforms like Vercel's V0. Based on this, we construct the multi-modality benchmark containing three core tasks: UI code generation, UI code edit, and UI code repair.

### 3.3 ROBUSTNESS BENCHMARK CONSTRUCTION

Robustness is crucial for evaluating models' reliability and performance in real-world programming scenarios, especially in code reasoning, where models must follow program logic rather than pattern matching. Hence, we adopt the perturbation strategies from CODECRASH (Lam et al., 2025), which include structural and semantic perturbations, to stress-test code reasoning under extreme and non-ideal programs using output prediction (Gu et al., 2024). We use an aggregated program structure-consistent perturbation (PSC-ALL) that integrates identifier renaming, conditional reformatting, and garbage code insertion, reconstructing the program structure while preserving functionality. Beyond structure, we adopt two levels of NL-embedded perturbations: contextual-level, where we inject manifestly misleading cues to the program context through code comments (MCC) or print statements (MPS), and reasoning-level, where it injects plausible but incorrect hints (MHC) to trigger rationalization. In our work, we use data from the CR collection (Section 3.1.2) and apply the above perturbation strategies to evaluate models' robustness.

## 4 EVALUATION SETUP

In this section, we provide the overall experimental setup. The detailed setup, such as the used prompt and metrics for each scenario, could be found in the Appendix B.

### 4.1 MODEL SELECTION

To provide a comprehensive evaluation across various LLMs, we evaluate over 26 state-of-the-art models of varying sizes and versions, including both open-source and closed-source LLMs: GPT family (Hurst et al., 2024; OpenAI, 2025a;b;c), Anthropic Claude series (Anthropic, 2024), Google Gemini (Google AI, 2024), DeepSeek family (DeepSeek-AI et al., 2025b;a), Alibaba Qwen (Yang et al., 2025b; Hui et al., 2024; Yang et al., 2025a), Meta LLaMA (Meta, 2024), and xAI Grok (Grok-3-Mini) (xAI, 2025). For multi-modality evaluation, we exclude models that cannot accept visual inputs and replace models that have multi-modality versions with their corresponding multi-modal variants (e.g., replacing Qwen2.5-72B-Instruct with Qwen2.5-72B-VL-Instruct (Qwen, 2025)). The detailed model list and their configuration are presented in the Appendix B.

### 4.2 ENHANCED EVALUATION METHOD

To avoid potential evaluation bias caused by using only one prompt, we employ the multi-prompt evaluation strategies tailored to each task's requirements for a more comprehensive and fair evaluation. For all tasks, we first adopt established prompt templates from recent benchmarks such as BIGCODEBENCH (Zhuo et al., 2024) and OCTOPACK (Muennighoff et al., 2023) as the seed prompt. To enhance prompt diversity and reduce potential bias, we use GPT-4o (Hurst et al., 2024) to generate two paraphrased variants of each base template and check their validity manually. Besides, we employ an adaptive solution extraction method that uses LLMs to extract solutions from LLM responses when Markdown parsing is ambiguous or fails (details in Appendix B.2).

### 4.3 EVALUATION METRICS

We select the most popular evaluation metrics for each task. For code generation, translation, and reasoning tasks, we adopt pass@1 accuracy (Chen et al., 2021). For code summarization and review tasks, we follow (Jiang et al., 2025; Sun et al., 2025) and employ LLM-as-judge evaluation using GPT-4o (Hurst et al., 2024) to assess quality on a 1-5 scale, which we convert to percentages for consistency. The test generation task is evaluated using compilation success rate and coverage metrics (line and branch coverage). Vulnerability detection employs standard classification metrics including accuracy, precision, recall, and F1-score. For multi-modality tasks, apart from code

Table 2: Overall model performance (%) on general coding tasks. The top three results on each task are highlighted in green ($1^{st}$), orange ($2^{nd}$), and blue ($3^{rd}$) backgrounds, respectively.

| Model Name | CG | CS | CT | CR | CRv | TG | VD | Avg. Rank |
|---|---|---|---|---|---|---|---|---|
| **GPT-5** | **89.9** | 65.7 | **97.9** | **97.8** | 33.1 | **82.6** | **67.3** | 1 |
| **Claude-Sonnet-4** | 74.0 | **65.9** | 86.0 | 87.9 | **35.0** | 77.0 | **69.5** | 2 |
| **Claude-3.7-Sonnet** | 70.0 | 63.7 | 85.1 | 57.6 | 34.8 | 75.3 | 61.8 | 3 |
| DeepSeek-R1 (0528) | 68.8 | 63.8 | 87.0 | 96.7 | 34.9 | 67.4 | 56.0 | 4 |
| o3-mini | 79.9 | 60.4 | 92.8 | 97.0 | 34.6 | 69.7 | 50.5 | 5 |
| GPT-4.1 | 76.8 | 60.0 | 87.6 | 63.5 | 34.4 | 75.4 | 59.8 | 6 |
| Qwen3-235B-A22B | 63.2 | 64.3 | 87.1 | 94.1 | 34.5 | 66.7 | 55.5 | 7 |
| o4-mini | 74.2 | 61.1 | 81.0 | **98.1** | 33.5 | 81.1 | 56.3 | 8 |
| Grok-3-Mini | 73.4 | 62.5 | 87.7 | 96.4 | **35.3** | 65.9 | 51.2 | 9 |
| DeepSeek-R1 | 59.9 | 63.8 | 89.2 | 95.1 | 33.4 | 69.0 | 56.5 | 10 |
| GPT-4o | 66.4 | 62.8 | 82.0 | 57.7 | 33.8 | 69.3 | 60.3 | 11 |
| Claude-3.5-Sonnet | 59.5 | **66.2** | 81.7 | 60.1 | 34.6 | 73.2 | 47.7 | 12 |
| DeepSeek-V3 | 65.2 | 64.3 | 82.1 | 57.7 | 34.2 | 68.6 | 51.5 | 13 |
| Gemini-2.5-Pro | 61.1 | 60.3 | **90.3** | 97.2 | 34.8 | 32.6 | 54.5 | 14 |
| Qwen3-32B | 63.1 | 63.1 | 86.0 | 94.0 | 34.2 | 65.2 | 53.5 | 15 |
| Qwen3-30B-A3B | 69.0 | 59.7 | 80.1 | 92.3 | 34.6 | 64.9 | 54.0 | 16 |
| GPT-4-turbo | 59.5 | 63.2 | 80.1 | 53.6 | 33.8 | 67.7 | 59.8 | 17 |
| LLaMA-3.3-70B | 40.7 | **65.9** | 70.0 | 47.2 | 33.9 | 66.7 | **62.3** | 18 |
| Gemma-3-27B | 51.3 | 61.3 | 65.9 | 41.6 | **35.0** | 64.7 | 62.0 | 19 |
| Qwen2.5-72B | 63.8 | 62.6 | 72.5 | 48.2 | 34.4 | 64.8 | 52.3 | 20 |
| Qwen2.5-Coder-32B | 62.5 | 62.6 | 74.6 | 56.2 | 34.2 | 65.0 | 51.7 | 21 |
| Claude-3.5-Haiku | 50.9 | 61.6 | 75.0 | 46.1 | 34.1 | 44.6 | 61.2 | 22 |
| LLaMA-4-Scout | 51.2 | 59.6 | 64.4 | 48.4 | 34.1 | 68.7 | 49.0 | 23 |
| LLaMA-3.1-70B | 48.7 | 58.6 | 67.7 | 41.5 | 33.4 | 66.3 | 57.2 | 24 |
| GPT-3.5-turbo | 50.6 | 56.3 | 66.5 | 34.8 | 31.3 | 67.5 | 45.8 | 25 |
| LLaMA-3.1-8B | 31.8 | 54.3 | 49.6 | 28.8 | 32.7 | 46.0 | 54.5 | 26 |

complication rate and code modification similarity (CMS), we also utilize visual specialized metrics including CLIP score and MLLM-as-Judge score (Xiao et al., 2025).

## 5 EXPERIMENT RESULTS

### 5.1 MULTI-TASK PERFORMANCE COMPARISON

Table 2 presents the performance comparison across different general coding tasks. Due to space limitation, we report only the most popular metric for each task, with full results provided in the Appendix C. The results show that current state-of-the-art models achieve strong performance on some tasks such as code summarization and code reasoning, but exhibit notable weaknesses in others like code review and test generation. Many models show large performance gaps across different task categories, suggesting that existing LLMs have not achieved consistent proficiency across all coding capabilities. Specifically, we could observe that:

**Models exhibit substantial performance variation across different tasks.** Current models tend to specialize in specific domains rather than achieving uniform capabilities, with no single model performing optimally across all evaluated tasks. For example, GPT-5 achieves exceptional performance in code generation with 89.9% accuracy and excels in test generation with 82.6% coverage rate, yet performs poorly on code review tasks with only 33.1% score. Similarly, o3-mini demonstrates strong reasoning capabilities, achieving 79.9% and 92.8% pass rate in code generation and code reasoning, but struggles with vulnerability detection, reaching only 50.5% accuracy.

**Different models lead different tasks.** For example, o4-mini achieves the best results in code reasoning at 98.1%, while Claude-Sonnet-4 performs best in vulnerability detection at 69.5%. Other models also show distinct areas of expertise. These results indicate that different models have developed specialized strengths in specific programming domains. This specialization reflects the diverse nature of coding tasks, which require different skills from logical reasoning to code understanding.

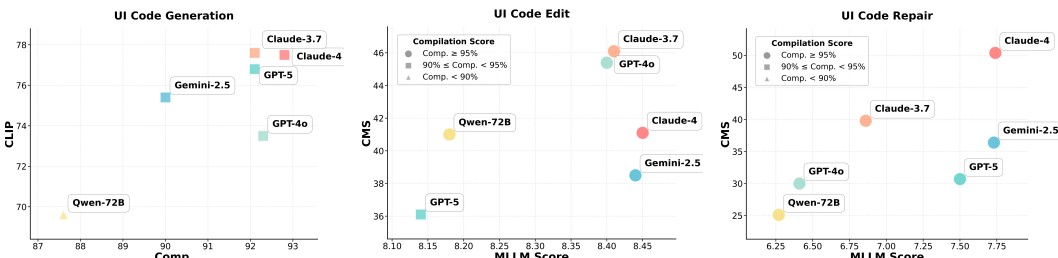

Figure 4: Multi-modality evaluation results.

## 5.2 MULTI-MODALITY EVALUATION

Figure 4 presents the multi-modality evaluation results of leading MLLMs. The detailed results of more models and each framework are shown in the Appendix C.8. We observe substantial performance variations and task-specific limitations across different tasks.

**Models show different performance bottlenecks on different tasks.** In UI code generation tasks, models are hindered by syntactic errors, facing the challenge of compilation errors. Claude-Sonnet-3.7 achieves the highest CLIP score of 77.6, demonstrating superior visual-semantic alignment, yet its compilation success rate of 92.1% falls slightly behind Claude-Sonnet-4's 92.8%. In contrast, UI code edit and repair tasks are primarily constrained by inadequate visual understanding and modification capabilities. We can find that the compilation rate of almost all models is higher than 95%, while both MLLM scores and CMS scores remain relatively modest, particularly in design repair tasks where CMS scores consistently fall below 50% across all evaluated models. The high compilation rates and lower functional accuracy scores indicate that while models can generate syntactically correct code given the code to modify, they struggle with precise code localization and targeted modifications.

## 5.3 ROBUSTNESS EVALUATION

Table 3: Robustness evaluation results. Darker red highlights represent more severe degradation under robustness testing.

| Model | Vanilla | PSC-ALL | MCC | MPS | MHC | Avg $\Delta\%$ |
|---|---|---|---|---|---|---|
| *Large Reasoning Models (enable thinking)* | | | | | | |
| GPT-5 | 99.5 | +0.5% | +0.0% | +0.0% | -0.5% | +0.0% |
| Gemini-2.5-Pro | 100.0 | -1.0% | -0.5% | -0.5% | -1.9% | -1.0% |
| DeepSeek-R1 | 98.1 | -1.5% | -2.5% | -1.5% | -5.9% | -2.8% |
| Qwen3-32B | 98.6 | -4.4% | -4.9% | -3.4% | -3.4% | -4.0% |
| o4-mini | 99.0 | -0.5% | -13.6% | -1.5% | -6.8% | -5.6% |
| Claude-Sonnet-4 | 94.7 | -8.6% | -2.5% | -7.1% | -7.6% | -6.5% |
| Qwen3-235B-A22B | 97.6 | -2.5% | -27.6% | -10.8% | -8.4% | -12.3% |
| *Large Language Models (under direct inference)* | | | | | | |
| Claude-3.7-Sonnet | 85.6 | -7.9% | -7.9% | -7.3% | -3.9% | -6.7% |
| Claude-3.5-Sonnet | 66.3 | -4.3% | -10.9% | -12.3% | -22.5% | -12.5% |
| GPT-4o | 73.1 | -12.5% | -21.1% | -28.3% | -21.1% | -20.7% |
| LLaMA-3.3-70B | 58.7 | -20.5% | -22.1% | -32.8% | -13.1% | -22.1% |
| GPT-4.1 | 78.8 | -12.8% | -30.5% | -27.4% | -20.7% | -22.9% |
| LLaMA-3.1-70B | 56.7 | -23.7% | -16.1% | -33.9% | -17.8% | -22.9% |
| Qwen2.5-32B-Coder | 61.5 | -12.5% | -39.8% | -32.8% | -20.3% | -26.4% |
| DeepSeek-V3 | 72.6 | -21.2% | -31.1% | -27.2% | -33.8% | -28.3% |
| Qwen2.5-72B | 63.5 | -18.9% | -25.0% | -37.1% | -42.4% | -30.9% |
| **Average** | 81.5 | -9.5% | -16.0% | -16.5% | -14.4% | -14.1% |

Table 3 presents the robustness evaluation results of different models under various perturbations. The experimental results reveal severe robustness issues in current LLMs on coding tasks. Based on the results, we have the following findings:

**All models exhibit substantial performance degradation under code perturbations.** With semantically-preserving code perturbations, all tested models show varying degrees of performance decline. On average, models experience performance drops of 9.5%, 16.0%, 16.5%, and 14.4% under PSC-ALL, MCC, MPS, and MHC, respectively, resulting in an overall average performance

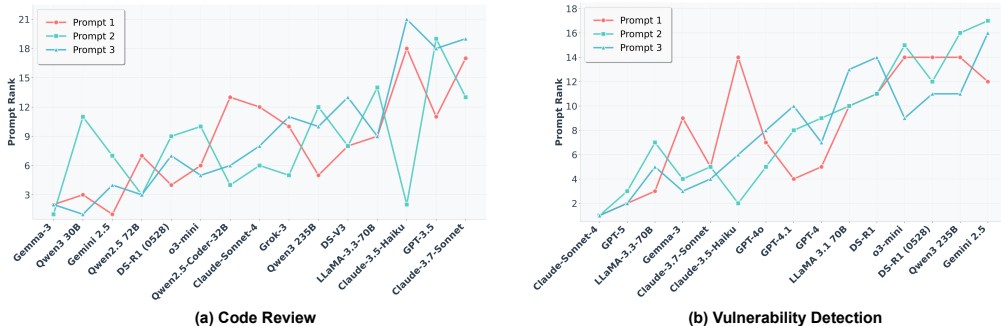

Figure 5: The performance variation of top-10 models using different prompts.

decline of 14.1%. This widespread problem suggests that current LLMs lack a robust understanding of code semantics and are easily misled by surface modifications.

**Large reasoning models demonstrate better robustness.** Models with advanced thinking capabilities, such as GPT-5 and Gemini-2.5-Pro, exhibit markedly stronger robustness compared to non-reasoning LLMs. These models show only minor performance fluctuations across most perturbation scenarios, with average performance drops controlled within 3%. In contrast, traditional models without reasoning exhibit considerable performance degradation, with models like DeepSeek-V3 and GPT-4o showing average performance decreases exceeding 20%.

**Contextual-level perturbations cause the most severe impact.** Models are more sensitive to contextual-level perturbations (MCC and MPS), indicating that LLMs are easily influenced by misleading natural language cues embedded in code. Misleading comments cause the largest performance drop at 20.0%, suggesting that models overly rely on comment information for code understanding rather than analyzing the actual code logic.

## 5.4 EFFECT OF THE MULTI-PROMPT EVALUATION

Figure 5 demonstrates the substantial performance variation across different prompts for code review and vulnerability detection tasks. The full results of all models on other tasks can be found in the Appendix C.9. Our analysis indicates that model performance exhibits significant sensitivity to prompt variations in some tasks. For example, Claude-3.5-Haiku shows remarkable fluctuations, with performance ranks ranging from as high as 3 to as low as 18 depending on the specific prompt used. These findings highlight the importance of employing multiple prompts to provide a more comprehensive and reliable evaluation of model capabilities, especially for tasks where prompt sensitivity is particularly evident.

## 6 CONCLUSION

This paper presents TREAT, a comprehensive evaluation framework that assesses the ability of LLMs in code intelligence tasks. Through multi-task, multi-language, and multi-modality evaluation of 26 state-of-the-art models, our framework reveals both their strengths and limitations, yielding several key insights into current models' ability to handle diverse coding scenarios and maintain robustness under code transformations. TREAT provides researchers and practitioners with a standardized approach for model comparison across real-world software development contexts.

## 7 LIMITATION AND FUTURE WORK

While TREAT provides a comprehensive evaluation framework, several limitations should be acknowledged. Our current evaluation mainly focus on the function level, which may not fully capture the complexity of real-world software engineering that requires repository-level understanding. This evaluation framework does not contains some aspects of code quality such as the security. Additionally, TREAT faces the persistent challenge of potential data contamination. In the future, we will continuously enhance the benchmark by expanding evaluation tasks, incorporating more evaluation aspects, and regularly updating evaluation datasets to prevent data contamination.

ETHICS STATEMENT

This work adheres to the ICLR Code of Ethics. We are dedicated to ensuring that TREAT serves exclusively for academic research. Our plan includes the launch of a leaderboard website and the provision of data and code access. During our data crawling process, we adhered to the regulations of each website, and all the GitHub data we crawled has permissive open-source licenses (Apache-2.0, MIT). TREAT does not contain any personal data or offensive content. No human subjects or animal experimentation was involved in this work.

REPRODUCIBILITY STATEMENT

To encourage reproducibility, we release our code and benchmark data at `https://code-treat.vercel.app/`. We describe the details of the benchmark construction in Section 3 and the experimental setup in Section 4. Finally, we elaborate further details in Appendix including the detailed data collection process for each task (Appendix A), the used LLMs and experimental setup (Appendix B) and further details of experiment results (Appendix C).

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

APPENDIX

CONTENTS

# A    DETAILED BENCHMARK CONSTRUCTION METHODS

## A.1    CODE GENERATION

**Language Scope.** In this paper, we concentrate our evaluation of code generation on Python and Java—two languages that together span a wide range of programming paradigms, from scripting and rapid prototyping to strongly-typed, object-oriented development.

Table 4: Dataset difficulty distribution

| Dataset | Total # | # Easy | # Medium | # Hard |
|---|---|---|---|---|
| GEEKSFORGEEKS | 1536 | 818 | 603 | 115 |
| HACKERRANK | 455 | 142 | 157 | 156 |

**Problem Definition.** Each problem is represented as a tuple $(P, T, S)$ where $P$ denotes the natural-language problem statement, $T$ the set of available test cases, and $S = \{S_{\text{python}}, S_{\text{java}}, S_{\text{cpp}}\}$ the set of ground-truth solutions in three languages.

**Problem Collection.** We construct benchmarks by scraping GEEKSFORGEEKS and HACKERRANK using Python-based HTML scrapers. For each problem, we extracted the title, natural-language description, difficulty level, release date, human-verified solutions in Python, Java, and C++, and the sample test cases provided on the platform. We retain only those problems for which at least one language-specific solution compiles and executes successfully, ensuring that every problem in the dataset has a valid implementation and avoiding ambiguous or unsolvable tasks.

**Test Collection.** When official test suites are available, we adopt them in full, as they typically exercise common edge cases. For problems with insufficient coverage, we follow the spirit of EVALPLUS (Liu et al., 2023) and LIVECODEBENCH (Jain et al., 2025) by using a large language model (LLM; GPT-4o in our implementation) to synthesize additional random and adversarial inputs. No auxiliary type signatures or annotations beyond the original problem description are provided. To standardize stdin/stdout evaluation across tasks and languages, we use a lightweight *Driver Code* harness that parses inputs from standard input and emits outputs to standard output; this harness is provided as part of our augmentation pipeline (Step 1).

**Three-stage augmentation.**

1. **Constraint elicitation & Driver Code provisioning.** Prompt the LLM to infer and state preconditions, invariants, input domains, and corner cases implied solely by the problem description (without external type information). In this stage, we also provide the *Driver Code* that specifies the STDIN format and expected STDOUT schema.

2. **Generator synthesis.** Prompt the LLM to produce an input-constructor function that samples random and adversarial test cases consistent with the elicited constraints and compatible with the provided *Driver Code*.

3. **Validation and iteration.** Use a second LLM to check whether the constructor violates the elicited constraints or the I/O contract implied by the *Driver Code*; on violation, refine and retry for up to three rounds. If validated, execute the ground-truth reference solution via the same driver to derive expected outputs, and retain only synthesized tests that either confirm correct behavior or expose faults, from which we compute TPR.

## A.2    CODE SUMMARIZATION

**Language Scope.** Our summarization dataset comprises function–docstring pairs from ten widely used languages on GitHub, including C, C++, C#, Go, Java, JavaScript, TypeScript, PHP, Python, and Ruby, enabling evaluation of cross-language generalization in code summarization.

**Project Selection and Data Collection.** We assembled our corpus from publicly available GitHub repositories created in 2023 and restricted to projects with permissive licenses (e.g., Apache-2.0, MIT) and at least 100 stars.

**Function Extraction and Cleaning.** Using the Tree-sitter library (Brunsfeld & GitHub, 2018), we parsed each repository to extract all function definitions along with their docstrings. We then applied

the cleaning methods proposed by Shi et al. (Shi et al., 2022) to isolate only the first sentence of each docstring, producing concise description–function pairs $(f, D)$.

**Dataset Statistics.** Table 5 summarizes the number of function–docstring pairs collected for each language after filtering and cleaning.

Table 5: Function–Docstring Pair Counts by Language

| Language | Count |
|---|---|
| C | 347,480 |
| C++ | 212,319 |
| C# | 28,862 |
| Java | 177,268 |
| Go | 680,785 |
| JavaScript | 188,309 |
| TypeScript | 70,917 |
| PHP | 90,312 |
| Python | 743,201 |
| Ruby | 2,140 |
| Total | 2,541,581 |

## A.3 CODE TRANSLATION

**Language Scope.** We focus exclusively on translations between Python and Java, enabling direct cross-language comparisons within two of the most widely used programming ecosystems.

**Dataset Composition.** Our Code Translation dataset is built atop the GEEKSFORGEEKS corpus, chosen for its extensive problem coverage and community-verified solutions. To introduce greater linguistic diversity and complexity, we integrate POLYHUMANEVAL (Tao et al., 2024a), a 14-language extension of HumanEval that has been rigorously validated across all target languages.

**Test Collection.** For GEEKSFORGEEKS code translation tasks, we reuse the generated test suites described in our code generation evaluation to ensure comparability across settings. For POLYHUMANEVAL, whose native tests are limited in scope, we augment coverage using the comprehensive LLM- and mutation-based test sets from EvalPlus's HUMANEVALPLUS pipeline (Liu et al., 2023). Before inclusion, we executed all EvalPlus-provided test cases and observed that some exhibit type incompatibilities or overflow-related issues for Java; leveraging the reference Java solutions from POLYHUMANEVAL, we detected and excluded such cases. The resulting corpus contains only executable, type-consistent cases suitable for cross-language evaluation, while exercising both nominal and corner-case behaviors of the translated programs.

## A.4 CODE REASONING

**Task Splitting.** We follow previous work (Gu et al., 2024) and divide code reasoning into two complementary tasks: input prediction and output prediction. In input prediction, the model must infer the missing inputs that produce a given expected output, while in output prediction, it must compute the correct output for supplied inputs. This dual setup probes both backward (from output to input) and forward (from input to output) program comprehension.

**Language Scope.** Consistent with our code generation and translation evaluations, we evaluate reasoning in both Python and Java, ensuring comparable cross-language insights.

**Dataset Construction.** From our HACKERRANK and GEEKSFORGEEKS corpora, we use Tree-sitter to identify each focal function and normalize its name to $f$, preserving any helper routines that $f$ invokes. For every masked function, an LLM (o3-mini) generates five candidate triples $\langle$inputs, expected_output, assertion$\rangle$. The prompt indicates the prediction target by replacing the corresponding element with the placeholder "??": for *input prediction* we mask expected_output, and for *output prediction* we mask inputs. We exclude trivial or ill-posed instances by remov-

ing functions with no parameters and non-informative returns (e.g., *None*/*void*), and by discarding triples that are inconsistent with the function interface or control-flow preconditions.

**Assertion Statements.** Each example is packaged with a language-appropriate executable check that binds the predicted quantity to a verifiable oracle. In Python, we `assert` that invoking $f$ on the predicted `inputs` equals `expected_output`. In Java, we use `assertEquals` to compare the method's return value against the expected result (with appropriate boxing and tolerance where needed). These assertions serve both as a standardized harness for evaluation and as a safeguard against malformed instances; any example that fails to execute or violates the assertion under the reference implementation is discarded.

## A.5   CODE REVIEW

**Language Scope.** Our code review dataset covers the same ten widely used programming languages—C, C++, C#, Go, Java, JavaScript, TypeScript, PHP, Python, and Ruby—to ensure broad applicability of models in generating reviews.

**Project Selection and Data Collection.** Repositories created in 2023 were selected if they carried a permissive license (e.g., Apache-2.0, MIT) and met a threshold of at least 100 stars at the 2025 crawl. Pull-request metadata and associated discussion threads for PRs opened in 2023 were then harvested and filtered according to the CodeReviewer (Li et al., 2022) protocol; forks, and archived projects were excluded.

**Review pair extraction.** For each pull request, we parsed the unified diff and decomposed it into individual diff hunks $D$ using GitHub's default context length, grouping changes by file. For every hunk we selected the earliest human-authored review comment $C$ that was paired with that hunk; comments authored by the commit author were excluded. If multiple comments addressed the same hunk, only the earliest was retained. The resulting collection of $(D, C)$ pairs—diff hunks annotated with human reviewer feedback—was used to train and evaluate automated code-review systems.

**Dataset Statistics.** Table 6 summarizes the number of diff-review pairs collected for each language after filtering and cleaning.

Table 6: Diff–review Pair Counts by Language

| Language | Count |
| --- | --- |
| C | 4,138 |
| C++ | 43,648 |
| C# | 18,055 |
| Java | 23,493 |
| Go | 44,191 |
| JavaScript | 11,634 |
| TypeScript | 109,813 |
| PHP | 2,186 |
| Python | 145,981 |
| Ruby | 3,597 |
| Total | 406,736 |

## A.6   TEST GENERATION

**Dataset Selection.** We relied on a publicly available pipeline that augments focal-method context to improve test-synthesis precision. For Python, we followed SYMPROMPT's methodology (Ryan et al., 2024), applying its context augmentation to 24 projects drawn from CODAMOSA (Lemieux et al., 2023) to enrich each target function with module-level context. Because SYMPROMPT does not provide explicit branch-coverage metadata, and because relying on users or LLMs to infer branching can lead to insufficient coverage, we augmented the pipeline with lightweight branch annotations (e.g., `has_branches` and `expected_branches`) to assist test construction.

### A.7 VULNERABILITY DETECTION

**Dataset Selection.** We adopted the PRIMEVUL benchmark (Ding et al., 2024), which comprises 6,968 expert-verified vulnerable functions and 228,800 benign functions spanning 140 Common Weakness Enumerations (CWEs). PRIMEVUL provides expert-guided labels, rigorous deduplication to eliminate near-duplicate fragments, and chronological splits designed to prevent temporal leakage between training and test sets. We also used the PRIMEVUL-PAIRED dataset, which pairs each vulnerable function with its patched counterpart, enabling pairwise evaluation of a model's sensitivity to semantic changes introduced by security fixes.

### A.8 MULTI-MODALITY TASKS

To evaluate the model's capability in handling multi-modality programming requirements, we adopt the DESIGNBENCH (Xiao et al., 2025), which encompasses three distinct tasks: UI code generation, UI code edit and UI code repair, defined as follows:

**UI Code Generation** ($\mathcal{T}_G$). The objective of UI code generation is to generate expected code based on the UI Mockups. Formally, given a UI design image $I$, the task aims to generate corresponding UI code $C$ such that $\mathcal{T}_G : I \rightarrow C$. The input contains the UI design image $I$, and the output is the UI code $C$ that accurately reproduces the visual layout and styling.

**UI Code Edit** ($\mathcal{T}_E$). The goal of the UI code edit is to generate front-end code that complies with user modification instructions. Given the original UI design image $I_o$, original UI code $C_o$, and user instruction $T$ described in natural language, the task produces modified code $C_{new}$ such that $\mathcal{T}_E : (I_o, C_o, T) \rightarrow C_{new}$. The input contains the original UI design image $I_o$, original UI code $C_o$, and user instruction $T$, while the output is the updated code $C_{new}$ incorporating the requested modifications.

**UI Code Repair** ($\mathcal{T}_R$). The goal of the UI code repair is to repair the UI code with display issues. Given the problematic UI code $C_p$, the problematic UI image $I_p$, the task generates repaired UI code $C_r$ such that $\mathcal{T}_R : (C_p, I_p) \rightarrow C_r$. The input contains the problematic UI code $C_p$ and image $I_p$, the output is the repaired code $C_r$ that resolves visual design issues.

### A.9 CODE ROBUSTNESS

To evaluate LLM robustness, we adopt the CodeCrash (Lam et al., 2025), a unified stress-testing benchmark, to systematically evaluate model robustness in code reasoning under semantically preserved perturbations using output prediction tasks (Gu et al., 2024). Specifically, CodeCrash designs four types of perturbations:

**Aggregated Structural Perturbations (PSC-ALL)**. Combine variable renaming, expression reformatting, and garbage code injection to construct functionally equivalent but complex programs, representing traditional transformations that expose whether LLMs rely on pattern matching.

**Contextual-level Misleading Perturbations**. **(1) Misleading Code Comments (MCC)**: Insert natural language comments that explicitly contradict the actual code logic, testing whether LLMs can filter out shallow misleading cues. **(2) Misleading Print Statements (MPS)**: Embed misleading messages as print statements, probing whether the effect is tied to a specific injection format.

**Reasoning-level Misleading Perturbations (MHC)**. Provide plausible but incorrect high-level hints about the expected outputs, directly challenging model reasoning and highlighting potential rationalization issues.

### A.10 CODE-TREAT-LITE

We provide the complete benchmark dataset Code-TREAT as well as the sampled Code-TREAT-lite (as described above) in an anonymous Hugging Face repository (`https://huggingface.co/Code-TREAT/datasets`). All experimental results in this paper are based on Code-TREAT-lite.

Table 7: List of Evaluated LLMs

| Model Name | Abbreviation | Size | Open-source |
|---|---|---|---|
| GPT-3.5-Turbo-0125 | GPT-3.5 | Unknown | × |
| GPT-4-Turbo-2024-04-09 | GPT-4 | Unknown | × |
| GPT-4o-2024-11-20 | GPT-4o | Unknown | × |
| GPT-4.1-2025-04-14 | GPT-4.1 | Unknown | × |
| o3-Mini (Med) | o3-mini | Unknown | × |
| o4-Mini (Med) | o4-mini | Unknown | × |
| GPT-5 | GPT-5 | Unknown | × |
| Claude-3.5-Haiku | Claude-3.5-Haiku | Unknown | × |
| Claude-3.5-Sonnet | Claude-3.5-Sonnet | Unknown | × |
| Claude-3.7-Sonnet | Claude-3.7-Sonnet | Unknown | × |
| Claude-Sonnet-4 | Claude-Sonnet-4 | Unknown | × |
| Gemini-2.5-Pro-05-06 | Gemini-2.5-Pro | Unknown | × |
| Grok-3-Mini (High) | Grok-3-Mini | Unknown | × |
| DeepSeek-V3 | DeepSeek-V3 | 671B (37B active) | ✓ |
| DeepSeek-R1 | DeepSeek-R1 | 671B (37B active) | ✓ |
| DeepSeek-R1 (0528) | DeepSeek-R1 (0528) | 671B (37B active) | ✓ |
| Qwen2.5-72B-Instruct | Qwen2.5-72B | 72B | ✓ |
| Qwen2.5-Coder-32B-Instruct | Qwen2.5-Coder-32B | 32B | ✓ |
| Qwen3-32B | Qwen3-32B | 32B | ✓ |
| Qwen3-30B-A3B | Qwen3-30B | 30B (3B active) | ✓ |
| Qwen3-235B-A22B | Qwen3-235B | 235B (22B active) | ✓ |
| LLaMA-3.1-8B-Instruct | LLaMA-3.1-8B | 8B | ✓ |
| LLaMA-3.1-70B-Instruct | LLaMA-3.1-70B | 70B | ✓ |
| LLaMA-3.3-70B-Instruct | LLaMA-3.3-70B | 70B | ✓ |
| LLaMA-4-Scout-17B-16E-Instruct | LLaMA-4-Scout | 109B (17B active) | ✓ |
| Gemma-3-27B-Instruct | Gemma-3-27B | 27B | ✓ |

## B DETAILED EXPERIMENTAL SETUP

### B.1 EVALUATED MODELS

As shown in Table 7, to provide a comprehensive evaluation across various LLMs, we evaluate 26 models of varying sizes and versions for general coding tasks, including both open-source and closed-source LLMs: GPT family (GPT-3.5-Turbo-0125, GPT-4-Turbo-2024-04-09, GPT-4o-2024-11-20, GPT-4.1-2025-04-14, o3-mini, o4-mini, GPT-5) (Hurst et al., 2024), Anthropic Claude (Claude-3.5-Haiku, Claude-3.5-Sonnet, Claude-3.7-Sonnet, Claude-Sonnet-4) (Anthropic, 2024), Google Gemini & Gemma (Gemini-2.5-Pro-05-06, Gemma-3-27B-Instruct) (Google AI, 2024), DeepSeek family (DeepSeek-V3, R1, R1-0528) (DeepSeek-AI et al., 2025b;a), Alibaba Qwen (Qwen2.5-72B-Instruct, Qwen-32B-Coder-Instruct, Qwen3-32B, Qwen3-30B-A3B, Qwen3-235B-A22B) (Yang et al., 2025b; Hui et al., 2024; Yang et al., 2025a), Meta LLaMA (LLaMA-3.1-8B-Instruct, LLaMA-3.1-70B-Instruct, LLaMA-3.3-70B-Instruct, LLaMA-4-Scout-17B-16E-Ins) (Meta, 2024), and xAI Grok (Grok-3-Mini) (xAI, 2025).

### B.2 CODE GENERATION

**Model Configuration.** Following BIGCODEBENCH (Zhuo et al., 2024), we set the temperature to 0.8 and, where supported, use a top-$p$ of 0.95. To accommodate both models limited to 8,192 tokens and those with larger context windows, we cap the maximum output length at $\min(\text{Token}_{max}, 16{,}384)$, where $\text{Token}_{max}$ denotes the maximum token allowance of each individual model.

**Prompt Design.** We employ three zero-shot prompt templates from recent benchmarks: BIG-CODEBENCH (Zhuo et al., 2024), OCTOPACK (Muennighoff et al., 2023), and LIVE-CODEBENCH (Jain et al., 2025), and retain the models' default system prompt settings. The detailed system and user prompts are provided in Appendix G.1.

**Data Sampling & Testing.** Owing to the large size of our Code Generation dataset corpus, we constructed a balanced yet tractable evaluation suite by randomly sampling problems from two sources, GEEKSFORGEEKS and HACKERRANK. For each language (Python and Java), we selected the same set of problems, with approximately half drawn from each source, to ensure a representative mix of difficulty levels and problem types. For every problem, the model receives only the natural-language (NL) description and is prompted to produce a complete solution in Markdown.

**Evaluation Process.** Model outputs were parsed from Markdown. If a response contained exactly one fenced code block, we extracted that block as the implementation; otherwise we invoked a secondary LLM-based extraction step to identify the intended implementation. The resulting code was passed to an automated pipeline that compiles/interprets and runs it against the reference test suite; syntax errors, runtime errors, and timeouts were recorded as failures.

**Evaluation Metrics.** We adopt PASS@1 accuracy (Chen et al., 2021) as the primary evaluation metric and scale all scores in $[0, 1]$ to percentages by multiplying by 100 for readability.

### B.3 CODE SUMMARIZATION

**Model Configuration.** Following BIGCODEBENCH (Zhuo et al., 2024), we set the temperature to 0.8 and, where supported, use a top-$p$ of 0.95. To accommodate both models limited to 8,192 tokens and those with larger context windows, we cap the maximum output length at $\min(\text{Token}_{max}, 16{,}384)$, where $\text{Token}_{max}$ denotes the maximum token allowance of each individual model.

**Prompt Design.** We adopt the zero-shot direct prompt template from Sun et al. (Sun et al., 2025). To increase prompt diversity, we then ask GPT-4o (Hurst et al., 2024) to generate two paraphrased variants of this template, yielding three distinct prompts for each test example. In our system prompt, we require models to output their answers in JSON format, in addition to the default helpful assistant instructions. Detailed system and user prompts, including those for the LLM-as-Judge setting, are provided in Appendix G.2.

**Data Sample & Testing.** We randomly sample 200 function–docstring pairs to form a balanced evaluation set. For each sample, the model was given only the function implementation and prompted to produce a concise summary.

**Evaluation Process.** We parsed the model's response and extracted the first sentence, mirroring the procedure used to isolate human docstrings in Shi et al. (Shi et al., 2022). The extracted sentences (both model-generated and human-written) were then passed through the same cleaning pipeline described in Shi et al. to normalize formatting and remove spurious tokens. Finally, the cleaned summaries were evaluated in batch using an LLM-based judging pipeline that assigns quality scores (e.g., correctness, completeness, relevance) which we aggregate into the reported metrics.

**Evaluation Metrics** Recognizing BLEU's inability to capture nuanced summaries and the variability of human annotations, we follow recent work (Sun et al., 2025) in using an LLM judge. Specifically, we prompt GPT-4o (Hurst et al., 2024) to assign each generated summary a quality score from 1 to 5, where higher values denote better accuracy, conciseness, and informativeness. We include the human reference summaries in the judging pool to establish a baseline. Finally, we scale all scores in $[1, 5]$ to percentages by multiplying by 20 for readability.

### B.4 CODE TRANSLATION

**Model Configuration.** Following BIGCODEBENCH (Zhuo et al., 2024), we set the temperature to 0.8 and, where supported, use a top-$p$ of 0.95. To accommodate both models limited

to 8,192 tokens and those with larger context windows, we cap the maximum output length at $\min(\text{Token}_{max}, 16{,}384)$, where $\text{Token}_{max}$ denotes the maximum token allowance of each individual model.

**Prompt Design.**    We employ the zero-shot direct prompt template from POLYHUMANEVAL (Sun et al., 2025) and then use GPT-4o (Hurst et al., 2024) to generate two paraphrased variants, resulting in three prompts per example. In our system prompt, we instruct the models to act as a code translation system. The detailed system and user prompts are provided in Appendix G.3.

**Data Sampling & Testing.**    To ensure a fair evaluation of model coding capabilities, we use the same sample data for HACKERRANK as in the CODE GENERATION task, and conduct comprehensive testing on the POLYHUMANEVAL benchmark. For each translation task, models receive only the source-language implementation and are prompted to generate the corresponding target-language implementation, which must be returned as a fenced code block in Markdown.

**Evaluation Process.**    Model outputs were parsed from Markdown. If a response contained exactly one fenced code block, we extracted that block as the implementation; otherwise we invoked a secondary LLM-based extraction step to identify the intended implementation. The resulting code was passed to an automated pipeline that compiles/interprets and runs it against the reference test suite; syntax errors, runtime errors, and timeouts were recorded as failures.

**Evaluation Metrics.**    We adopt PASS@1 accuracy (Chen et al., 2021) as the primary evaluation metric and scale all scores in $[0, 1]$ to percentages by multiplying by 100 for readability.

### B.5   CODE REVIEW

**Model Configuration.**    Following BIGCODEBENCH (Zhuo et al., 2024), we set the temperature to 0.8 and, where supported, use a top-$p$ of 0.95. To accommodate both models limited to 8,192 tokens and those with larger context windows, we cap the maximum output length at $\min(\text{Token}_{max}, 16{,}384)$, where $\text{Token}_{max}$ denotes the maximum token allowance of each individual model.

**Prompt Design.**    We adopt the zero-shot prompt template from LLAMA-REVIEWER (Lu et al., 2023) and use GPT-4o-2024-11-20 (Hurst et al., 2024) to generate two paraphrased variants, yielding three prompts per example. In our system prompt, we instruct the models to act as specialized code reviewers and to produce comments in JSON format. The detailed system and user prompts—including those used for LLM-as-Judge—are provided in Appendix G.4.

**Data Sampling & Testing.**    For each language we randomly sampled 200 diff–review pairs from the union dataset, maintaining diversity by stratifying on change size and file type. Each example consists of a single diff hunk; models were provided only the hunk and asked to generate a review comment in the prescribed JSON format.

**Evaluation Process.**    We parse the model's JSON response to extract the `"comments"` field. Parsed comments are then scored using an LLM-as-judge procedure (GPT-4o), which rates lexical similarity to the human reference review comment.

**Evaluation Metrics.**    Because BLEU scores are low and uninformative when comparing detailed LLM reviews against human review comments, we follow Jiang et al. (Jiang et al., 2025) in using an LLM as judge. We use an GPT-4o  (Hurst et al., 2024) as the judge to rate each generated review's lexical similarity to the human reference on a 1–5 scale according to the following detailed setup:

- **Judge Messages.** Judge model is prompted with a *system message* that instructs it to "grade a generated code review," mimic grading ten times internally, and then output only the final JSON grade:

  ```
  {"grade":<integer 1-5>}
  ```

- **Grading Criteria.** The *judge prompt* presents both the generated and reference reviews and specifies:
    1. Grade = 5 if the review is identical to the reference.
    2. Grade = 4 if it is semantically equivalent despite wording differences.
    3. Grade = 3 if it correctly covers some reference comments.
    4. Grade = 2 if only loosely related in content.
    5. Grade = 1 if completely unrelated.
- **Aggregation.** We collect the JSON grades for all 200 samples per language and report (1) the mean grade, and (2) the distribution of grades 1 through 5 to analyze model performance and error modes. we scale all scores in $[1, 5]$ to percentages by multiplying by 20 for readability.

### B.6 CODE REASONING

**Model Configuration.** Following BIGCODEBENCH (Zhuo et al., 2024), we set the temperature to 0.8 and, where supported, use a top-$p$ of 0.95. To accommodate both models limited to 8,192 tokens and those with larger context windows, we cap the maximum output length at $\min(\text{Token}_{max}, 16{,}384)$, where $\text{Token}_{max}$ denotes the maximum token allowance of each individual model.

**Prompt Design.** We adopt the zero-shot direct prompt template from CRUX (Gu et al., 2024) and use GPT-4o (Hurst et al., 2024) to generate two paraphrased variants, yielding three prompts per example. In our system prompt, we require models to output their answers in JSON format, in addition to the default helpful assistant instructions. The detailed system and user prompts are provided in Appendix G.5.

**Data sampling & Testing.** We randomly sampled 200 problems from the union of HACKERRANK and GEEKSFORGEEKS. For each problem we constructed two task variants: (1) input prediction — the models receive the function and a masked input placeholder and are asked to produce concrete input values; and (2) output prediction — the models receive the function and specific input(s) and are asked to produce the expected output. Models were instructed to return answers in a compact, programmatically parsable form (e.g., Python/Java literals or comma-separated values).

**Evaluation process.** We use a simple, regex-first parsing pipeline: when a model reply clearly contains the needed values we extract them with lightweight patterns and substitute them into the masked assertion (e.g., `assert f(*inputs) == expected_output`). If the regex extraction fails or is ambiguous, we fall back to a secondary LLM (GPT-4o-mini) to produce a canonical representation for the assertion. The resulting assertions are executed; compilation errors, runtime exceptions, and timeouts are recorded as failures.

**Evaluation Metrics.** We use pass@1 accuracy (Chen et al., 2021) as our primary metric, where each example is scored as 1 if the model's prediction satisfies the assertion and 0 otherwise. We report the average pass@1 over the evaluation set and and scale all score in $[0, 1]$ to percentage by multiplying 100 for improved readability.

### B.7 TEST GENERATION

**Model Configuration.** Following BIGCODEBENCH (Zhuo et al., 2024), we set the temperature to 0.8 and, where supported, use a top-$p$ of 0.95. To accommodate both models limited to 8,192 tokens and those with larger context windows, we cap the maximum output length at $\min(\text{Token}_{max}, 16{,}384)$, where $\text{Token}_{max}$ denotes the maximum token allowance of each individual model.

**Prompt Design.** We adopt the zero-shot direct prompt templates from SymPrompt (Ryan et al., 2024) for Python. To enhance diversity, we ask GPT-4o (Hurst et al., 2024) to paraphrase each template into two additional variants, yielding three distinct prompts. In our system prompt, we instruct the models to act as a professional unit test writer. The detailed system prompt and user prompts are provided in Appendix G.6.

**Data Sampling & Testing.** We randomly sample 200 functions from the CODAMOSA dataset (Lemieux et al., 2023) with the context-assistant annotations provided by SYM-PROMPT (Ryan et al., 2024).

**Evaluation Process.** Model outputs were parsed from Markdown. The extracted test suite is executed with `pytest` (using `pytest-cov`) in a sandboxed environment; we record syntax errors, runtime failures, test outcomes, timeouts, and per-example coverage.

**Evaluation Metrics.** Following prior works (Yuan et al., 2023; Xie et al., 2023; Yang et al., 2024d), we assess test quality using three metrics:

- **Compilation Success Rate (CSR):** code executing successfully or not.
- **Line Coverage ($\text{Cov}_L$):** the percentage of source-code lines exercised by the test suite.
- **Branch Coverage ($\text{Cov}_B$):** the percentage of control-flow branches executed by the test suite.

### B.8 VULNERABILITY DETECTION

**Model Configuration.** Following BigCodeBench (Zhuo et al., 2024), we set the temperature to 0.8 and, where supported, use a top-$p$ of 0.95. To accommodate both models limited to 8,192 tokens and those with larger context windows, we cap the maximum output length at $\min(\text{Token}_{max}, 16{,}384)$, where $\text{Token}_{max}$ denotes the maximum token allowance of each individual model.

**Prompt Design.** We adopt the zero-shot direct prompt templates from Ding et al. (Ding et al., 2024) for both PRIMEVUL and PRIMEVUL-PAIRED. To increase prompt diversity, we ask GPT-4o (Hurst et al., 2024) to generate two paraphrased variants of each template, yielding three prompts per example. In our system prompt, we instruct the models to act as a security expert in analyzing code for vulnerabliity. The detailed system prompt and user prompts are provided in Appendix G.7.

**Data Sampling & Tesing.** We randomly sampled 200 single-function examples from PRIMEVUL and 200 function pairs from PRIMEVUL-PAIRED. For the single-function set we enforced a class-balance constraint so that the absolute difference between the number of *vulnerable* and *benign* examples is < 10 to avoid skewed metrics and ensure stable comparisons.

**Evaluation Process.** For PRIMEVUL, each model receives a single function and predicts either *vulnerable* or *benign*. For PRIMEVUL-PAIRED, the model is shown both the vulnerable and patched versions of a function and returns a pair of labels. Predictions are compared against ground-truth annotations to produce per-example outcomes; we aggregate these outcomes to compute the reported metrics.

**Evaluation Metrics.** We evaluate the model performance using the following metrics:

- **PRIMEVUL *Metrics.***
    - *Accuracy*: the fraction of correct predictions over all examples
    - *Precision*: the proportion of predicted vulnerabilities that are true vulnerabilities
    - *Recall*: the proportion proportion of actual vulnerabilities correctly identified
    - *F1-Score*: the harmonic mean of precision and recall
- **PRIMEVUL-PAIRED *Metrics.*** We treat each vulnerable–patched pair as a single instance, classifying the model's joint prediction into one of four categories (Ding et al., 2024):
    - *Pair-wise Correct (P-C)*: both functions labeled correctly.
    - *Pair-wise Vulnerable (P-V)*: both functions (incorrectly) labeled vulnerable.
    - *Pair-wise Benign (P-B)*: both functions (incorrectly) labeled benign.
    - *Pair-wise Reversed (P-R)*: labels swapped between vulnerable and patched versions.

### B.9 MULTI-MODALITY TASKS

**Model Configuration.** We evaluate eight MLLMs that have been widely explored in multi-modal tasks, namely GPT-4o-2024-11-20 (Hurst et al., 2024), GPT-5 (OpenAI, 2025b), Claude-3.7-Sonnet (Anthropic, 2024), Claude-Sonnet-4 (Anthropic, 2025b), Gemini-2.5, Gemini-2.0 (Doshi, 2025), Qwen2.5-VL-72B-Instruct (Qwen, 2025), LLaMA-3.2-90B-Vision (Meta, 2024).

In configuring the MLLMs, we set the temperature to 0 and the maximum number of tokens output to 16,384.

**Evaluation Metrics.** We evaluate the model performance using the following metrics:

- *Visual Metrics*. *CLIP* (Radford et al., 2021) is applied to measure the semantic similarity between the generated and original webpages.
- *Code Metrics*. (1) *Compilation Success Rate (CSR)* represents the percentage of generated code that compiles successfully without errors. Assume that the total number of samples is N and the number of samples compiled successfully is S, then $CSR = \frac{S}{N}$. (2) *Code Modification Similarity (CMS)*. We employ the Jaccard similarity (Thada & Jaglan, 2013) to quantify the precision of code modifications on design edit and design repair tasks by comparing the sets of modified line numbers between the ground truth and generated code. Let $A$ represent the set of line numbers modified in the ground truth code and $B$ represent the set of line numbers modified in the generated code. The CMS is formally defined as: $CMS(A, B) = \frac{|A \cap B|}{|A \cup B|}$.
- *MLLM-as-Judge Metrics*. MLLMs have shown great performance in assisting judges across diverse modalities (Chen et al., 2024b; Wang et al., 2025b). Therefore, we prompt GPT-4o (Hurst et al., 2024) to determine whether the model meets the user's requirements on the design edit task and resolve the design issues on the design repair task, and output an **MLLM score** between 0 and 10 with detailed explanations (0-3 denotes the poor edit/repair, 4-6 denotes partial edit/repair, 7-8 denotes good edit/repair and 9-10 denotes excellent edit/repair).

### B.10 CODE ROBUSTNESS

**Model Configuration.** We evaluate multiple models of varying sizes and versions, including both open-source and closed-source LLMs: GPT family (GPT-4o, GPT-4o-mini, GPT-4.1, 5, o4-mini) (Hurst et al., 2024; OpenAI, 2025a;b), Anthropic Claude (Claude-3.5-Sonnet, 3.7-Sonnet, Claude-Sonnet-4) (Anthropic, 2024; 2025a;b), Google Gemini (Gemini-2.5-Pro) (Doshi, 2025), DeepSeek (DeepSeek-V3, R1) (DeepSeek-AI et al., 2025b), Alibaba Qwen (Qwen2.5-32B-Coder-Instruct, Qwen2.5-72B-Instruct, Qwen3-32B, 235B-A22B) (Hui et al., 2024; Yang et al., 2025b;a), and Meta LLaMA (LLaMA-3.1-70B-Instruct, LLaMA-3.3-70B-Instruct) (Grattafiori et al., 2024).

**Evaluation Metrics.** We adopt PASS@1 accuracy (Chen et al., 2021) as the primary evaluation metric and scale all scores in $[0, 1]$ to percentages by multiplying by 100 for readability. All perturbed results are reported as relative ($\Delta_\% = \frac{\text{Perturbed} - \text{VAN}}{\text{VAN}} \times 100\%$) differences from the corresponding vanilla baseline.

## C DETAILED EXPERIMENT RESULTS AND ANALYSIS

### C.1 CODE GENERATION

**Model Performance and Language Effects.** Table 8 summarizes Pass@1 accuracy on code generation tasks, split by model, language (Python, Java), and dataset (GeeksforGeeks, HackerRank). GPT-5 establishes itself as the clear leader, achieving the highest accuracy across all splits, including an overall Pass@1 of 89.9%, with 91.5% on GeeksforGeeks and 85.3% on HackerRank. Its performance is robust across both Python (89.1%) and Java (90.8%), suggesting strong cross-language capability and minimal bias between these languages at the frontier of model capabilities. Second-tier models, such as o3-mini at 79.9% and GPT-4.1 at 76.8%, lag behind GPT-5 by a significant margin—over 10 percentage points in most splits. Among all evaluated models, there is a rapid drop-off after the top performers, with accuracy for the majority of models clustering in the 50–70% range, indicating a clear stratification in current code generation capabilities.

Table 8: Model Performance on Code Generation. The top three results on each task are highlighted in green ($1^{st}$) , orange ($2^{nd}$) , and blue ($3^{rd}$) backgrounds, respectively.

| Model | Overall | | | Python | | | Java | | |
|---|---|---|---|---|---|---|---|---|---|
| | Overall | GeeksforGeeks | HackerRank | Overall | GeeksforGeeks | HackerRank | Overall | GeeksforGeeks | HackerRank |
| GPT-5 | 89.9 | 91.5 | 85.3 | 89.1 | 90.4 | 84.9 | 90.8 | 92.5 | 85.7 |
| o3-mini (Med) | 79.9 | 81.4 | 75.6 | 82.2 | 84.0 | 76.9 | 77.6 | 78.7 | 74.4 |
| GPT-4.1-2025-04-14 | 76.8 | 79.4 | 68.8 | 77.8 | 81.2 | 67.5 | 75.7 | 77.6 | 70.0 |
| o4-mini (Med) | 74.2 | 76.9 | 65.9 | 79.5 | 81.7 | 72.8 | 68.8 | 72.1 | 59.0 |
| Claude-Sonnet-4 | 74.0 | 75.4 | 69.7 | 75.0 | 76.9 | 69.2 | 73.0 | 73.9 | 70.2 |
| Grok-3-Mini (High) | 73.4 | 73.9 | 71.8 | 70.6 | 71.8 | 67.1 | 76.1 | 76.0 | 76.4 |
| Claude-3.7-Sonnet | 70.0 | 70.1 | 69.6 | 68.2 | 67.3 | 71.0 | 71.7 | 72.9 | 68.3 |
| Qwen3-30B-A3B | 69.0 | 74.3 | 53.0 | 70.6 | 77.7 | 49.0 | 67.4 | 70.8 | 57.1 |
| DeepSeek-R1 (0528) | 68.8 | 68.0 | 71.0 | 70.3 | 70.8 | 68.8 | 67.2 | 65.2 | 73.2 |
| GPT-4o-2024-11-20 | 66.4 | 68.4 | 60.4 | 71.0 | 73.3 | 63.8 | 61.8 | 63.4 | 57.1 |
| DeepSeek-V3 | 65.2 | 66.2 | 62.5 | 75.3 | 77.7 | 68.3 | 55.2 | 54.6 | 56.7 |
| Qwen2.5-72B-Instruct | 63.8 | 65.3 | 59.2 | 65.2 | 66.1 | 62.5 | 62.3 | 64.4 | 55.9 |
| Qwen3-235B-A22B | 63.2 | 63.1 | 63.8 | 64.7 | 65.4 | 62.7 | 61.8 | 60.7 | 64.9 |
| Qwen3-32B | 63.1 | 64.5 | 58.8 | 66.3 | 69.6 | 56.4 | 59.9 | 59.5 | 61.2 |
| Qwen2.5-Coder-32B-Instruct | 62.5 | 64.4 | 57.0 | 64.4 | 66.9 | 56.9 | 60.7 | 61.9 | 57.1 |
| Gemini-2.5-Pro-05-06 | 61.1 | 60.7 | 62.3 | 68.1 | 65.4 | 76.3 | 54.1 | 56.0 | 48.4 |
| DeepSeek-R1 | 59.9 | 55.6 | 72.7 | 61.0 | 57.9 | 70.5 | 58.8 | 53.4 | 74.8 |
| Claude-3.5-Sonnet | 59.5 | 62.4 | 50.6 | 60.0 | 62.4 | 52.9 | 58.9 | 62.5 | 48.2 |
| GPT-4-turbo-2024-04-09 | 59.5 | 60.4 | 56.6 | 65.1 | 66.4 | 61.1 | 53.8 | 54.4 | 52.1 |
| Gemma-3-27B-Instruct | 51.3 | 52.1 | 48.7 | 57.7 | 59.3 | 52.9 | 44.9 | 45.0 | 44.6 |
| Llama-4-Scout-17B-16E-Instruct | 51.2 | 51.3 | 51.0 | 52.8 | 53.8 | 49.7 | 49.6 | 48.7 | 52.4 |
| Claude-3.5-Haiku | 50.9 | 55.8 | 36.0 | 60.6 | 65.7 | 45.4 | 41.1 | 46.0 | 26.6 |
| GPT-3.5-turbo-0125 | 50.6 | 51.7 | 47.0 | 53.8 | 55.7 | 48.1 | 47.4 | 47.8 | 46.0 |
| Llama-3.1-70B-Instruct | 48.7 | 50.3 | 43.9 | 49.8 | 51.7 | 43.9 | 47.6 | 48.9 | 43.9 |
| Llama-3.3-70B-Instruct | 40.7 | 37.9 | 49.1 | 39.7 | 37.3 | 46.8 | 41.7 | 38.5 | 51.4 |
| Llama-3.1-8B-Instruct | 31.8 | 33.1 | 27.7 | 33.1 | 34.6 | 28.8 | 30.4 | 31.6 | 26.6 |

One notable observation is the prevalence of instruction drift: models sometimes generate unsolicited usage examples or disregard required code templates. This behavior results in outputs that are incompatible with automated evaluation harnesses, leading to an underestimation of their actual coding ability in certain cases. Despite such issues, the overall rankings remain consistent and robust across different benchmarks and codebases.

**Task-Specific Limitations and Performance Bottlenecks.** The analysis highlights several persistent challenges in current model performance. Most prominently, there is a systematic bias toward Python across almost all models except GPT-5, as evidenced by a consistent 10–20 percentage point performance gap in favor of Python over Java. This bias likely stems from imbalances in pretraining and fine-tuning datasets, which tend to heavily favor Python, thus equipping models with stronger priors for Python syntax, idioms, and library usage. Additionally, prompt misinterpretation emerges as a recurring bottleneck, particularly for Java. When prompts use phrasings such as "write a {lang} script" and lang is set to Java, several models mistakenly generate JavaScript code. This systematic evaluation artifact results in unconditional failures for affected Java test cases and reveals a vulnerability in current prompt understanding, especially when language names overlap with other widely-used programming languages. Together, these findings underscore the need for more balanced and robust instruction-following, as well as improved prompt disambiguation and better handling of language-specific conventions in next-generation code models.

## C.2 CODE SUMMARIZATION

**Model Performance and Language Effects.** Table 9 summarizes model performance on code summarization tasks, as measured by GPT-4o judge scores across a variety of programming languages. GPT-5 leads with an exceptional overall quality score of 98.4%, consistently outperforming all competitors across nearly every language, including C, C++, Java, Python, and JavaScript, where scores frequently exceed 98%. Claude-3.5-Sonnet and LLaMA-3.3-70B also demonstrate strong capabilities, achieving overall scores of 96.5% and 96.0%, respectively, with their best results clustering closely to the top performer. Across the board, most large and instruction-tuned models maintain remarkably high summarization quality, often in the 95–99% range for mainstream languages, and all substantially exceed the human baseline of 44.6%. This wide margin highlights the brevity and sparsity typical of organic docstrings, which the LLMs' outputs decisively surpass in both completeness and style. Notably, model scale and instruction quality are primary drivers of performance, as reasoning-oriented models such as DeepSeek-R1 and Gemini-2.5-Pro do not exhibit any consistent advantage in this task. Instead, their results underscore the importance of fine-tuning and

Table 9: Model Performance on Code Summarization (%). The top three results on each task are highlighted in green (1st), orange (2nd), and blue (3rd) backgrounds, respectively.

*JS = JavaScript, TS = TypeScript

| Model | Overall | C | C++ | C# | Go | Java | JS | TS | PHP | Python | Ruby |
|---|---|---|---|---|---|---|---|---|---|---|---|
| GPT-5 | **98.4** | **99.0** | **99.0** | **99.0** | **99.6** | **99.2** | **98.3** | **99.2** | **98.4** | **98.6** | 93.8 |
| Claude-3.5-Sonnet | **96.5** | **97.2** | **95.8** | **96.0** | **96.0** | 95.5 | **97.4** | **97.5** | **96.9** | 95.9 | **97.2** |
| LLaMA-3.3-70B-Instruct | **96.0** | 94.6 | **96.1** | **95.8** | **96.7** | **96.5** | 95.7 | **96.6** | **97.1** | 95.6 | **95.4** |
| Qwen3-235B-A22B | 95.3 | **95.6** | 94.7 | 95.1 | 95.5 | **95.7** | **95.8** | 94.7 | 95.6 | **96.1** | **94.3** |
| Claude-Sonnet-4 | 93.8 | 91.8 | 93.8 | 94.1 | 93.1 | 94.4 | 94.2 | 93.7 | 95.1 | 93.2 | 94.3 |
| DeepSeek-V3 | 92.8 | 94.3 | 93.4 | 92.6 | 92.0 | 91.6 | 92.5 | 92.4 | 91.8 | 94.4 | 93.2 |
| DeepSeek-R1 (0528) | 90.6 | 92.3 | 91.5 | 90.8 | 88.8 | 89.5 | 90.3 | 91.0 | 88.7 | 94.5 | 88.9 |
| Qwen3-32B | 90.2 | 92.6 | 90.3 | 90.0 | 88.0 | 86.7 | 91.4 | 90.0 | 90.1 | 93.6 | 89.2 |
| GPT-4-turbo-2024-04-09 | 90.0 | 91.4 | 90.3 | 89.8 | 91.3 | 87.8 | 89.7 | 89.3 | 87.6 | 91.2 | 91.1 |
| Claude-3.7-Sonnet | 88.1 | 87.4 | 88.0 | 86.4 | 87.7 | 83.6 | 88.9 | 87.8 | 88.9 | 90.4 | 91.9 |
| GPT-4o-2024-11-20 | 87.7 | 86.7 | 86.9 | 90.0 | 86.9 | 86.5 | 87.1 | 87.2 | 86.8 | 87.8 | 91.2 |
| Qwen2.5-Coder-32B-Instruct | 86.8 | 87.0 | 86.0 | 85.3 | 88.9 | 85.2 | 89.0 | 87.3 | 86.0 | 87.6 | 85.6 |
| Qwen2.5-72B-Instruct | 86.5 | 87.4 | 86.8 | 87.7 | 86.0 | 84.4 | 86.6 | 86.8 | 84.7 | 87.7 | 87.3 |
| Claude-3.5-Haiku | 85.2 | 87.0 | 85.5 | 82.9 | 88.1 | 86.8 | 87.0 | 86.5 | 85.2 | 87.3 | 76.0 |
| Grok-3-Mini (High) | 85.1 | 85.8 | 85.4 | 83.9 | 85.6 | 84.1 | 85.8 | 86.6 | 84.7 | 85.4 | 83.7 |
| o4-mini (Med) | 84.6 | 84.3 | 85.3 | 83.7 | 87.7 | 83.3 | 85.6 | 86.0 | 83.7 | 87.9 | 78.5 |
| Gemma-3-27B-Instruct | 83.0 | 80.6 | 81.5 | 83.1 | 82.6 | 82.3 | 83.5 | 82.3 | 84.3 | 84.7 | 84.6 |
| Qwen3-30B-A3B | 81.4 | 82.5 | 80.1 | 81.3 | 81.7 | 77.5 | 81.8 | 82.0 | 80.3 | 84.1 | 83.1 |
| GPT-4.1-2025-04-14 | 80.2 | 79.0 | 79.8 | 78.8 | 82.1 | 80.5 | 80.9 | 80.3 | 80.5 | 80.0 | 80.4 |
| o3-mini (Med) | 79.5 | 86.7 | 87.0 | 86.1 | 87.2 | 82.9 | 86.7 | 84.8 | 85.5 | 23.0 | 85.4 |
| Gemini-2.5-Pro-05-06 | 78.7 | 78.1 | 77.4 | 79.7 | 79.9 | 74.2 | 82.0 | 80.2 | 80.3 | 77.1 | 78.3 |
| LLaMA-3.1-70B-Instruct | 74.5 | 74.2 | 75.4 | 78.5 | 73.3 | 74.9 | 75.9 | 75.2 | 79.0 | 67.8 | 71.1 |
| LLaMA-4-Scout-17B-16E-Instruct | 74.4 | 70.6 | 71.7 | 79.6 | 75.9 | 77.2 | 73.7 | 72.5 | 77.0 | 70.3 | 75.2 |
| GPT-3.5-turbo-0125 | 71.2 | 71.9 | 70.4 | 72.9 | 70.3 | 70.2 | 69.2 | 71.3 | 72.2 | 71.3 | 72.2 |
| LLaMA-3.1-8B-Instruct | 64.2 | 59.9 | 64.0 | 64.9 | 66.7 | 63.8 | 64.5 | 64.2 | 64.6 | 61.7 | 67.6 |
| Human Baseline | 44.6 | 44.1 | 38.2 | 41.8 | 54.3 | 34.8 | 45.4 | 40.1 | 48.3 | 48.8 | 50.7 |

high-quality, language-specific training data for code summarization. There is minimal variation across programming languages, with even lower-resource languages such as Ruby and PHP receiving high-quality summaries from the top models, further confirming the strong generalization of frontier LLMs.

**Task-Specific Limitations and Performance Bottlenecks.** Despite near-ceiling performance from leading models, while our evaluation setup follows the most widely adopted practices in the field (Sun et al., 2025), our experimental observations reveal that this methodology may still present several issues for further resolution. The use of a single LLM judge, such as GPT-4o, may introduce bias and style sensitivity, as its preferences for certain phrasings or lengths can influence scores independently of semantic correctness. Minor formatting differences or stylistic choices may therefore yield notable score shifts, even in cases where the underlying summary remains unchanged. Furthermore, although efforts are made to ensure the judge differs from the evaluated models, there remains a risk of cross-family self-preference, potentially inflating the scores for some model families. Pairwise comparisons mitigate but do not eliminate this concern. Another artifact of the evaluation pipeline is the uniform truncation of outputs to the first sentence, which can inadvertently penalize models that prepend reasoning or place their core summary at the end, this is observed, for instance, in o3-mini's Python summaries. This truncation policy is applied to all models without model-specific adjustment, ensuring fairness but possibly underestimating some models' true summarization ability. Taken together, these factors highlight that while current models display extraordinary summarization accuracy, subtle evaluation artifacts and judge-related biases represent the main bottlenecks to further performance gains in this setting.

## C.3 CODE TRANSLATION

Table 10: Model Performance on Code Translation (%). The top three results on each task are highlighted in green ($1^{st}$), orange ($2^{nd}$), and blue ($3^{rd}$) backgrounds, respectively.

| Model | Overall | | | HackerRank | | | PolyHumanEval | | |
|---|---|---|---|---|---|---|---|---|---|
| | Overall | Python→Java | Java→Python | Overall | Python→Java | Java→Python | Overall | Python→Java | Java→Python |
| GPT-5 | 97.9 | 98.1 | 97.8 | 97.1 | 97.9 | 96.3 | 99.0 | 98.4 | 99.6 |
| o3-mini (Med) | 92.8 | 90.9 | 94.8 | 89.3 | 87.3 | 91.3 | 97.3 | 95.3 | 99.2 |
| Gemini-2.5-Pro-05-06 | 90.3 | 92.6 | 88.0 | 84.9 | 88.9 | 80.9 | 97.1 | 97.2 | 97.0 |
| DeepSeek-R1 | 89.2 | 87.0 | 91.4 | 84.2 | 82.5 | 85.9 | 95.5 | 92.7 | 98.4 |
| Grok-3-Mini (High) | 87.7 | 86.5 | 88.8 | 81.8 | 81.8 | 81.3 | 95.2 | 92.3 | 98.2 |
| GPT-4.1-2025-04-14 | 87.6 | 86.6 | 88.6 | 80.0 | 79.6 | 80.4 | 97.2 | 95.3 | 99.0 |
| Qwen3-235B-A22B | 87.1 | 82.1 | 92.1 | 80.8 | 74.5 | 87.2 | 95.0 | 91.7 | 98.4 |
| DeepSeek-R1 (0528) | 87.0 | 84.7 | 89.2 | 83.5 | 83.5 | 83.5 | 91.4 | 86.2 | 96.5 |
| Qwen3-32B | 86.0 | 82.3 | 89.7 | 79.7 | 76.3 | 83.2 | 93.9 | 89.8 | 98.0 |
| Claude-Sonnet-4 | 86.0 | 86.9 | 85.0 | 76.6 | 79.3 | 73.9 | 97.9 | 96.5 | 99.2 |
| Claude-3.7-Sonnet | 85.1 | 84.1 | 86.1 | 78.0 | 80.6 | 75.5 | 94.0 | 88.4 | 99.6 |
| DeepSeek-V3 | 82.1 | 80.4 | 83.9 | 71.6 | 70.4 | 72.8 | 95.5 | 93.1 | 98.0 |
| GPT-4o-2024-11-20 | 82.0 | 80.8 | 83.2 | 70.7 | 70.8 | 70.5 | 96.3 | 93.5 | 99.2 |
| Claude-3.5-Sonnet | 81.7 | 82.9 | 80.5 | 74.2 | 79.5 | 68.9 | 91.2 | 87.2 | 95.1 |
| o4-mini (Med) | 81.0 | 79.5 | 82.5 | 70.0 | 67.9 | 72.0 | 95.0 | 94.1 | 95.9 |
| GPT-4-turbo-2024-04-09 | 80.1 | 78.0 | 82.2 | 69.4 | 68.3 | 70.5 | 93.7 | 90.4 | 97.0 |
| Qwen3-30B-A3B | 80.1 | 75.2 | 85.0 | 69.5 | 64.4 | 74.5 | 93.6 | 88.8 | 98.4 |
| Claude-3.5-Haiku | 75.0 | 72.7 | 77.2 | 61.5 | 61.1 | 61.9 | 92.1 | 87.4 | 96.7 |
| Qwen2.5-Coder-32B-Instruct | 74.6 | 74.7 | 74.5 | 58.4 | 59.9 | 56.9 | 95.1 | 93.5 | 96.7 |
| Qwen2.5-72B-Instruct | 72.5 | 75.9 | 69.1 | 56.2 | 59.8 | 52.6 | 93.2 | 96.3 | 90.0 |
| Llama-3.3-70B-Instruct | 70.0 | 68.8 | 71.1 | 57.1 | 58.2 | 55.9 | 86.4 | 82.3 | 90.4 |
| Llama-3.1-70B-Instruct | 67.7 | 68.2 | 67.2 | 51.8 | 54.3 | 49.2 | 87.9 | 85.8 | 90.0 |
| GPT-3.5-turbo-0125 | 66.5 | 66.8 | 66.2 | 47.6 | 49.5 | 45.7 | 90.5 | 88.8 | 92.3 |
| Gemma-3-27B-Instruct | 65.9 | 65.4 | 66.3 | 46.6 | 49.8 | 43.4 | 90.2 | 85.2 | 95.3 |
| Llama-4-Scout-17B-16E-Instruct | 64.4 | 63.1 | 65.8 | 49.4 | 50.6 | 48.2 | 83.4 | 78.9 | 88.0 |
| Llama-3.1-8B-Instruct | 49.6 | 47.2 | 52.1 | 29.1 | 31.4 | 26.8 | 75.7 | 67.3 | 84.1 |

**Model Performance and Dataset Effects.** Table 10 presents a comprehensive comparison of LLMs on Python↔Java code translation, reporting Pass@1 accuracy across both HACKERRANK and POLYHUMANEVAL benchmarks. GPT-5 establishes a new state-of-the-art with 97.9% overall accuracy, maintaining exceptional results in both directions and across datasets, including up to 99.0% on PolyHumanEval. The next tier—o3-mini and Gemini-2.5-Pro—remains highly competitive ($\geq$90%), while most leading models cluster above 85%. Among the strongest models, translation is nearly symmetric in both directions, confirming balanced competence. Notably, Pass@1 scores are systematically higher on POLYHUMANEVAL than on HACKERRANK for all model tiers and translation directions, indicating that POLYHUMANEVAL is an relative easier benchmark, likely due to models' greater exposure to HumanEval-style problems in pretraining. There is a sharp performance drop beyond the frontier models, with accuracy falling into the 65–75% range for lower tiers. Overall, the results reveal a clear capacity-performance scaling effect, with newer and larger models outperforming smaller or earlier versions by a substantial margin.

**Task-Specific Limitations and Performance Bottlenecks.** Despite these advances, several challenges remain prominent, particularly among mid- and lower-tier models. While leading models exhibit robust and symmetric performance, many smaller models show a tendency for higher accuracy in the Java→Python direction, benefiting from Python's more permissive syntax and forgiving I/O; however, this advantage is not universal, with some exceptions observed. The significant and consistent gap between results on POLYHUMANEVAL and HACKERRANK underscores a broader limitation in model generalization: most models achieve high performance on familiar, benchmark-like problems but are less reliable on the stricter or more diverse scenarios found in HackerRank. For less capable models, accuracy drops sharply, reflecting both a lack of robustness to new evaluation harnesses and a persistent gap between surface-level correctness and deeper semantic understanding. These findings highlight that, although state-of-the-art models now translate code between Python and Java with near-perfect fidelity on established benchmarks, substantial room for improvement remains in achieving robust and generalizable code translation across diverse datasets and real-world settings.

## C.4 CODE REVIEW

**Model Performance and Language Effects.** Table 11 presents the lexical similarity ratings of contemporary large language models (LLMs) on code review generation, as evaluated by GPT-4o, which assesses how closely model-generated reviews resemble human-written references in terms of

Table 11: Model Performance on Code Review Generation (%). The top three results on each task are highlighted in green ($1^{st}$) , orange ($2^{nd}$) , and blue ($3^{rd}$) backgrounds, respectively.

| Model | Overall | C | Cpp | Csharp | Go | Java | Javascript | Php | Python | Ruby | Typescript |
|---|---|---|---|---|---|---|---|---|---|---|---|
| Gemma-3-27B-Instruct | 31.7 | 28.9 | 30.6 | 32.3 | 32.1 | 30.9 | 34.3 | 30.5 | 33.0 | 33.0 | 31.7 |
| Qwen3-30B-A3B | 31.6 | 29.9 | 31.9 | 31.1 | 33.0 | 30.9 | 32.9 | 31.1 | 32.7 | 31.4 | 30.8 |
| Gemini-2.5-Pro-05-06 | 31.5 | 29.3 | 31.5 | 31.5 | 31.2 | 30.1 | 35.0 | 29.9 | 32.4 | 32.6 | 31.4 |
| Qwen2.5-72B-Instruct | 31.3 | 29.5 | 31.0 | 31.9 | 32.5 | 30.1 | 35.0 | 29.6 | 31.7 | 31.0 | 30.5 |
| DeepSeek-R1 (0528) | 31.1 | 28.5 | 30.9 | 31.4 | 31.9 | 30.9 | 34.4 | 29.3 | 31.5 | 31.4 | 31.3 |
| o3-mini (Med) | 31.1 | 28.6 | 31.5 | 31.0 | 31.7 | 30.2 | 34.8 | 29.9 | 31.6 | 31.3 | 30.7 |
| Qwen2.5-Coder-32B-Instruct | 31.1 | 29.0 | 31.1 | 31.5 | 31.1 | 30.3 | 34.7 | 29.1 | 31.9 | 31.1 | 31.0 |
| Grok-3-Mini (High) | 30.9 | 28.7 | 30.4 | 31.2 | 31.9 | 30.9 | 33.6 | 29.4 | 31.2 | 30.8 | 31.5 |
| Qwen3-235B-A22B | 30.9 | 28.9 | 30.3 | 30.9 | 32.0 | 29.4 | 34.4 | 29.3 | 31.6 | 31.2 | 31.5 |
| Claude-Sonnet-4 | 30.9 | 28.7 | 30.6 | 31.3 | 31.4 | 29.6 | 34.1 | 30.1 | 31.2 | 31.4 | 31.0 |
| DeepSeek-V3 | 30.9 | 28.1 | 29.9 | 31.1 | 31.9 | 30.4 | 33.5 | 30.4 | 32.0 | 30.2 | 31.4 |
| LLaMA-3.3-70B-Instruct | 30.7 | 28.4 | 29.4 | 31.9 | 31.2 | 29.8 | 32.9 | 29.3 | 31.8 | 31.6 | 30.7 |
| Claude-3.5-Haiku | 30.6 | 28.9 | 31.6 | 30.5 | 31.3 | 30.2 | 30.7 | 31.3 | 30.1 | 30.1 | 31.0 |
| Claude-3.7-Sonnet | 30.4 | 28.6 | 30.6 | 31.1 | 31.1 | 30.3 | 32.6 | 29.6 | 30.1 | 30.2 | 30.1 |
| GPT-3.5-turbo-0125 | 30.4 | 30.5 | 31.6 | 29.7 | 31.6 | 30.7 | 29.6 | 29.0 | 32.4 | 29.2 | 29.8 |
| Qwen3-32B | 30.4 | 29.0 | 29.9 | 30.1 | 31.3 | 30.2 | 32.5 | 29.5 | 30.5 | 29.9 | 30.7 |
| GPT-4o-2024-11-20 | 30.3 | 28.3 | 30.5 | 29.8 | 30.8 | 29.5 | 34.1 | 28.9 | 30.4 | 30.7 | 30.3 |
| LLaMA-3.1-8B-Instruct | 30.2 | 28.4 | 29.2 | 29.0 | 31.3 | 30.2 | 32.0 | 28.8 | 31.5 | 31.0 | 30.7 |
| LLaMA-3.1-70B-Instruct | 30.2 | 28.4 | 29.9 | 30.4 | 31.1 | 29.3 | 32.3 | 29.4 | 30.8 | 30.3 | 29.7 |
| LLaMA-4-Scout-17B-16E-Instruct | 30.1 | 28.3 | 29.7 | 30.2 | 30.7 | 29.5 | 32.3 | 29.1 | 30.7 | 30.2 | 30.4 |
| Claude-3.5-Sonnet | 30.0 | 28.7 | 29.0 | 30.0 | 30.8 | 29.4 | 33.2 | 28.3 | 30.0 | 30.1 | 30.1 |
| GPT-4-turbo-2024-04-09 | 29.7 | 27.3 | 29.1 | 30.1 | 30.7 | 29.3 | 32.3 | 29.6 | 29.1 | 29.4 | 29.9 |
| GPT-4.1-2025-04-14 | 29.4 | 27.3 | 28.5 | 29.0 | 30.2 | 29.2 | 32.6 | 28.7 | 29.8 | 28.8 | 30.4 |
| o4-mini (Med) | 29.0 | 26.9 | 28.5 | 28.8 | 29.6 | 28.3 | 32.5 | 28.0 | 28.8 | 29.3 | 29.4 |
| DeepSeek-R1 | 27.3 | 24.9 | 27.0 | 26.4 | 27.9 | 27.2 | 30.6 | 25.7 | 28.0 | 26.6 | 28.2 |
| GPT-5 | 26.9 | 24.3 | 26.6 | 26.9 | 26.7 | 25.8 | 30.5 | 25.7 | 26.9 | 26.9 | 28.4 |

wording, structure, and focus. In this evaluation, higher ratings indicate that a model's review is lexically and stylistically closer to the human reference, while lower ratings reflect greater divergence. Gemma-3-27B achieves the highest overall similarity rating at 31.7%, closely followed by Qwen3-30B (31.6%) and Gemini-2.5-Pro (31.5%), with leading models demonstrating robust performance across a diverse set of programming languages. For instance, Gemma-3-27B obtains the top similarity ratings in languages such as Java, Python, Ruby, and TypeScript, while Qwen3-30B and Gemini-2.5-Pro excel in Cpp, Go, and JavaScript. Notably, in JavaScript, both Gemini-2.5-Pro and Qwen2.5-72B attain the highest similarity rating (35.0%), underscoring the competitive landscape. Despite these achievements, the overall ratings remain modest and tightly clustered, reflecting the inherent challenge of matching human reviewer style and phrasing under this evaluation protocol.

**Task-Specific Limitations and Performance Bottlenecks.** The main limitation of this task lies in the evaluation method. Although we follow the popular LLM-as-a-judge evaluation method in this field, the dependence on a single human-written reference and lexical similarity as judged by GPT-4o may pose some limitations. The metric inherently favors model outputs that closely mimic the specific language and focus of the human review, rather than those that offer unique, alternative, or equally valid critiques. Consequently, a higher similarity rating signals a closer match to the reference in terms of phrasing and content, while a lower rating often indicates linguistic or stylistic divergence, not necessarily a deficiency in review quality. Furthermore, even state-of-the-art models that generate comprehensive or insightful comments may receive limited credit when the reference review is incomplete, uninformative, or fails to address key issues in the code. This phenomenon is particularly evident with models such as GPT-5, which perform strongly across most code-related tasks and frequently generate high-quality, detailed review comments. Despite this, GPT-5 may still obtain relatively modest similarity ratings if its suggestions differ from or go beyond those present in the human reference, especially in cases where the reference itself is shallow or lacks substance. This reliance on potentially limited human reviews as ground truth can obscure genuine advances in model capability, and may penalize models that identify subtle bugs or offer substantive suggestions overlooked by the reference. The constrained spread of similarity ratings among leading models thus suggests that current progress is bounded by the ability to imitate the human reference rather than provide substantively better reviews.

## C.5 CODE REASONING

Table 12: Model Performance (%) on Code Reasoning. The top three results on each task are highlighted in green ($1^{st}$), orange ($2^{nd}$), and blue ($3^{rd}$) backgrounds, respectively.

| Model | Overall | | | Input | | | Output | | |
|---|---|---|---|---|---|---|---|---|---|
| | Overall | Python | Java | Overall | Python | Java | Overall | Python | Java |
| o4-mini (Med) | 98.1 | 96.6 | 99.5 | 97.7 | 96.1 | 99.3 | 98.4 | 97.1 | 99.7 |
| GPT-5 | 97.8 | 95.7 | 100.0 | 98.2 | 96.4 | 100.0 | 97.5 | 94.9 | 100.0 |
| Gemini-2.5-Pro-05-06 | 97.2 | 95.4 | 99.0 | 98.2 | 97.7 | 98.8 | 96.2 | 93.2 | 99.1 |
| o3-mini (Med) | 97.0 | 94.6 | 99.5 | 96.9 | 94.6 | 99.3 | 97.2 | 94.6 | 99.7 |
| DeepSeek-R1 (0528) | 96.7 | 94.7 | 98.7 | 97.0 | 95.3 | 98.6 | 96.3 | 94.0 | 98.7 |
| Grok-3-Mini (High) | 96.4 | 93.3 | 99.5 | 97.0 | 94.5 | 99.4 | 95.8 | 92.1 | 99.5 |
| DeepSeek-R1 | 95.1 | 93.0 | 97.2 | 95.4 | 94.7 | 96.0 | 94.8 | 91.3 | 98.3 |
| Qwen3-235B-A22B | 94.1 | 90.5 | 97.6 | 93.4 | 89.9 | 96.9 | 94.8 | 91.2 | 98.3 |
| Qwen3-32B | 94.0 | 91.5 | 96.5 | 93.6 | 91.3 | 96.0 | 94.4 | 91.7 | 97.1 |
| Qwen3-30B-A3B | 92.3 | 89.6 | 95.0 | 91.5 | 89.0 | 93.9 | 93.2 | 90.2 | 96.1 |
| Claude-Sonnet-4 | 87.8 | 85.7 | 90.0 | 85.2 | 83.8 | 86.7 | 90.5 | 87.6 | 93.3 |
| GPT-4.1-2025-04-14 | 63.5 | 61.8 | 65.2 | 59.9 | 57.5 | 62.2 | 67.1 | 66.0 | 68.2 |
| Claude-3.5-Sonnet | 60.1 | 58.9 | 61.3 | 56.3 | 53.4 | 59.3 | 63.8 | 64.4 | 63.2 |
| DeepSeek-V3 | 57.7 | 56.8 | 58.5 | 52.8 | 51.9 | 53.7 | 62.6 | 61.8 | 63.4 |
| GPT-4o-2024-11-20 | 57.7 | 55.2 | 60.1 | 54.2 | 52.7 | 55.7 | 61.1 | 57.7 | 64.6 |
| Claude-3.7-Sonnet | 57.6 | 55.0 | 60.1 | 54.0 | 51.1 | 57.0 | 61.1 | 59.0 | 63.1 |
| Qwen2.5-Coder-32B | 56.2 | 52.6 | 59.7 | 50.8 | 45.3 | 56.3 | 61.5 | 59.9 | 63.2 |
| GPT-4-turbo-2024-04-09 | 53.6 | 52.4 | 54.8 | 51.1 | 49.2 | 53.0 | 56.1 | 55.7 | 56.6 |
| LLaMA-4-Scout | 48.4 | 47.5 | 49.2 | 40.9 | 35.4 | 46.4 | 55.8 | 59.7 | 52.0 |
| Qwen2.5-72B | 48.2 | 48.2 | 48.3 | 43.5 | 41.9 | 45.1 | 53.0 | 54.5 | 51.4 |
| LLaMA-3.3-70B | 47.2 | 43.8 | 50.7 | 45.5 | 39.5 | 51.5 | 49.0 | 48.0 | 49.9 |
| Claude-3.5-Haiku | 46.1 | 45.4 | 46.7 | 42.7 | 40.0 | 45.3 | 49.5 | 50.7 | 48.2 |
| Gemma-3-27B-Instruct | 41.6 | 39.0 | 44.3 | 37.3 | 30.4 | 44.1 | 46.0 | 47.5 | 44.5 |
| LLaMA-3.1-70B | 41.5 | 38.1 | 45.0 | 38.7 | 33.5 | 43.9 | 44.4 | 42.6 | 46.1 |
| GPT-3.5-turbo-0125 | 34.8 | 35.1 | 34.4 | 32.5 | 30.9 | 34.1 | 37.0 | 39.3 | 34.7 |
| LLaMA-3.1-8B | 28.8 | 32.6 | 25.0 | 26.7 | 29.9 | 23.6 | 30.8 | 35.2 | 26.4 |

**Model Performance and Reasoning Effects.** Table 12 provides a comprehensive overview of model capabilities on code reasoning tasks, measured through input and output prediction accuracy in both Python and Java. The results indicate a marked stratification among model families, with GPT-5 and Gemini-2.5-Pro setting the state of the art. o4-mini achieves the highest overall Pass@1 accuracy of 98.1%, maintaining balanced strength across both Python and Java. GPT-5 excels particularly on Java, reaching perfect accuracy in both overall and output prediction, and maintaining a strong position on Python. Gemini-2.5-Pro stands out for its superior input prediction in Python and competitive results elsewhere. Other models such as o3-mini, DeepSeek-R1, and Grok-3-Mini also demonstrate consistently high accuracy, illustrating that advances in architecture and scaling correlate directly with improved reasoning performance. Notably, this capacity-reasoning relationship becomes increasingly evident in more complex settings; larger and more recent models consistently outperform earlier or smaller counterparts, particularly in Python where the task demands more sophisticated reasoning. In contrast, models like Claude-Sonnet-4, which perform well in web-based evaluations, do not transfer this advantage fully to code reasoning, as evidenced by a lower overall accuracy of 87.8%. The trailing group of models, including GPT-3.5, LLaMA-3.1-8B, and compact Qwen or Gemma variants, remain limited in their reasoning capabilities, frequently falling below 50% overall accuracy. This sharp divide underscores the importance of both model scale and design in supporting complex reasoning tasks across programming languages.

**Task-Specific Limitations and Performance Bottlenecks.** Further examination of the results highlights persistent bottlenecks that inhibit optimal model performance, particularly in input reasoning

for Python. Even among top-performing models, there is a clear and recurring gap between Python and Java, with input prediction in Python proving more challenging and less consistent. A key factor underlying this discrepancy appears to be the inherent flexibility and less rigid syntax of Python, which increases the potential for subtle formatting and representation errors in predicted inputs or outputs. Models frequently struggle with faithfully preserving the expected structure of string literals and variable representations in Python, leading to a measurable drop in accuracy, whereas Java's stricter and more explicit syntax mitigates such issues and enables higher reliability in both input and output prediction. This trend is further accentuated among mid- and lower-tier models, where input reasoning accuracy for Python can fall below 60% or even lower, in stark contrast to the consistently higher performance observed in Java. These results suggest that despite recent progress, current architectures still face significant obstacles in capturing and generalizing language-specific conventions, particularly in the more flexible and variable Python setting. Addressing these bottlenecks will require not only continued scaling but also more targeted innovations in code understanding and syntactic reasoning across diverse programming paradigms.

## C.6 TEST GENERATION

Table 13: Model Performance (%) on Test Generation. The top three results on each task are highlighted in green ($1^{st}$), orange ($2^{nd}$), and blue ($3^{rd}$) backgrounds, respectively.

| Model | SymPrompt | | |
|---|---|---|---|
| | CSR | Cov$_L$ | Cov$_{Br}$ |
| Claude-3.5-Sonnet | **99.8** | 73.2 | 70.3 |
| o4-mini (Med) | **99.8** | **81.1** | **77.3** |
| Claude-3.5-Haiku | **99.7** | 44.6 | 38.2 |
| Qwen3-235B-A22B | **99.7** | 66.7 | 58.9 |
| Claude-3.7-Sonnet | 99.3 | 75.3 | 71.0 |
| GPT-4-turbo-2024-04-09 | 99.3 | 67.7 | 60.3 |
| Qwen2.5-Coder-32B-Instruct | 99.3 | 65.0 | 58.1 |
| Qwen3-30B-A3B | 99.3 | 64.9 | 59.4 |
| o3-mini (Med) | 99.3 | 69.7 | 66.7 |
| Claude-Sonnet-4 | 99.2 | **77.0** | **73.5** |
| GPT-4.1-2025-04-14 | 99.2 | 75.4 | 72.3 |
| GPT-5 | 99.2 | **82.6** | **81.8** |
| Gemini-2.5-Pro-05-06 | 99.0 | 32.6 | 25.1 |
| Qwen2.5-72B-Instruct | 99.0 | 64.8 | 56.0 |
| Qwen3-32B | 99.0 | 65.2 | 58.2 |
| DeepSeek-V3 | 98.8 | 68.6 | 63.5 |
| GPT-3.5-turbo-0125 | 98.8 | 67.5 | 55.4 |
| DeepSeek-R1 (0528) | 98.7 | 67.4 | 58.8 |
| DeepSeek-R1 | 98.5 | 69.0 | 62.8 |
| GPT-4o-2024-11-20 | 98.5 | 69.3 | 63.6 |
| LLaMA-3.1-70B-Instruct | 98.5 | 66.3 | 56.2 |
| Grok-3-Mini (High) | 98.3 | 65.9 | 62.5 |
| LLaMA-3.3-70B-Instruct | 98.3 | 66.7 | 58.0 |
| LLaMA-4-Scout-17B-16E-Instruct | 97.7 | 68.7 | 58.3 |
| Gemma-3-27B-Instruct | 97.5 | 64.7 | 56.3 |
| LLaMA-3.1-8B-Instruct | 96.0 | 46.0 | 33.7 |

**Model Performance and Capacity Effects.** Table 13 summarizes model performance on the SymPrompt-Python unit test generation benchmark, reporting comprehensive success rate (CSR), line coverage (Cov$_L$), and branch coverage (Cov$_{Br}$). Both Claude-3.5-Sonnet and achieve the highest CSR of 99.8%, establishing a clear upper bound in reliability for input/output prediction. However, when considering code coverage metrics, GPT-5 distinguishes itself with leading results in both line coverage (82.6%) and branch coverage (81.8%), closely followed by and Claude-Sonnet-4. Notably, Claude-3.5-Sonnet, while excelling in CSR, demonstrates moderate coverage (73.2% and 70.3% for Cov$_L$ and Cov$_{Br}$, respectively), suggesting some limitation in generating tests that comprehensively explore program logic.

Performance varies substantially across model families and sizes. The latest Claude, GPT, and Qwen variants consistently surpass earlier versions and smaller-scale models, underscoring a strong capacity-performance relationship in unit test generation. Larger models such as Claude-3.7-Sonnet, Qwen3-235B, and GPT-4.1 approach top-tier results in coverage, while smaller or prior-generation models like Gemini-2.5-Pro and LLaMA-3.1-8B lag considerably, particularly in coverage metrics. This performance stratification reinforces that scaling and architectural improvements yield measurable gains, especially on the more demanding aspects of code analysis and test completeness.

**Task-Specific Limitations and Performance Bottlenecks.** Despite high comprehensive success rates across most frontier models, coverage remains a persistent bottleneck. Many models maintain near-ceiling CSR yet fall short in coverage, revealing a discrepancy between producing minimal passing tests and generating diverse cases that robustly validate program behavior. For instance, and Claude-3.7-Sonnet, while highly reliable, are still outperformed by GPT-5 in both line and branch coverage, highlighting a gap in the ability to exercise complex code paths. Lower coverage by models such as Gemini-2.5-Pro and Claude-3.5-Haiku further underscores challenges in code reasoning and exploration, likely attributable to limited contextual understanding or training focus.

A particularly striking phenomenon is observed in Gemini-2.5-Pro, which, despite achieving a competitive CSR of 99.0%, exhibits extremely low coverage rates for both line (32.6%) and branch (25.1%) metrics. This suggests a fundamental shortcoming in the model's ability to generate tests that adequately explore program execution paths. Qualitative inspection reveals that Gemini-2.5-Pro frequently produces tests that either redundantly mock dependencies or even reimplement the focal method itself within the test suite, behaviors which are inconsistent with standard unit testing practice. This pattern likely reflects a lack of exposure to unit test generation tasks during model training, resulting in overgeneralized or misaligned output that fails to capture the intended testing objectives. Such findings highlight the importance of task-specific fine-tuning and exposure for robust coding capabilities in automated test generation.

## C.7 VULNERABILITY DETECTION

**Model Performance and Comparative Effects.**

Table 14 reports the performance of LLMs on vulnerability detection across both single-function and paired-function scenarios, highlighting significant contrasts between models and task setups. In the single-function PRIMEVUL setting, Claude-Sonnet-4 achieves the highest accuracy (69.5%) and F1 score (73.7%), setting a new state of the art for this benchmark. GPT-5 and GPT-4-turbo closely follow, with F1 scores of 69.2% and 69.9% respectively, underscoring consistent improvements from recent GPT-family advances. Gemini-2.5-Pro and GPT-4o also demonstrate robust recall, with Gemini-2.5-Pro achieving the highest recall (92.9%) yet comparatively lower precision, resulting in moderate overall F1. Notably, models like Qwen2.5-72B and Qwen2.5-Coder-32B demonstrate unusually high precision (73.2% and 70.4%), but this comes at the cost of extremely low recall, indicating a tendency toward conservative positive predictions while missing many actual vulnerabilities.

In the more challenging PRIMEVUL-PAIRED task, model performance diverges sharply. GPT-4.1 attains the highest P-C score (90.8%), evidencing an exceptional ability to simultaneously label both vulnerable and patched variants correctly. In contrast, Gemini-2.5-Pro leads in P-V (72.4%), indicating a strong bias toward labeling both functions as vulnerable, which maximizes recall but inflates false positives. Certain models, including Qwen2.5-72B and LLaMA-3.3-70B, stand out with strong P-B scores (81.3% and 79.3%, respectively), reflecting a pronounced preference for benign classification. Across most models, however, the P-R metric remains relatively low, suggesting that catastrophic reversals—where patched code is labeled vulnerable and vice versa—are still infrequent but not eliminated. These results reinforce that while LLMs have made strides in detecting vulnerabilities in isolation, comparative reasoning between functionally similar but semantically divergent code remains a significant obstacle.

**Task-Specific Limitations and Performance Bottlenecks.**

Analysis of the results reveals persistent task-specific bottlenecks that constrain model effectiveness on vulnerability detection. In the single-function scenario, several models achieve respectable accuracy and F1 scores by leveraging recognizable vulnerability patterns or established coding anti-

Table 14: Model Performance on Vulnerability Detection. The top three results on each task are highlighted in green ($1^{st}$), orange ($2^{nd}$), and blue ($3^{rd}$) backgrounds, respectively.

| Model | PrimeVul | | | | PrimeVul-Paired | | | |
|---|---|---|---|---|---|---|---|---|
| | Acc | Prec | Recall | F1 | P-C | P-V | P-B | P-R |
| Claude-Sonnet-4 | 69.5 | 66.8 | 82.1 | 73.7 | 73.3 | 18.0 | 2.8 | 5.8 |
| GPT-4-turbo-2024-04-09 | 59.8 | 57.3 | 89.7 | 69.9 | 49.5 | 10.7 | 33.0 | 6.8 |
| GPT-5 | 67.3 | 67.9 | 70.5 | 69.2 | 80.3 | 13.5 | 2.5 | 3.7 |
| GPT-4o | 60.3 | 58.3 | 83.3 | 68.6 | 41.5 | 28.2 | 14.2 | 16.2 |
| LLaMA-3.1-70B-Instruct | 57.2 | 55.5 | 89.1 | 68.4 | 18.3 | 0.3 | 59.7 | 21.7 |
| Claude-3.5-Haiku | 61.2 | 59.3 | 80.8 | 68.4 | 32.8 | 3.8 | 35.0 | 28.3 |
| Gemini-2.5-Pro-05-06 | 54.5 | 53.7 | 92.9 | 68.1 | 25.6 | 72.4 | 0.5 | 1.5 |
| Gemma-3-27B-Instruct | 62.0 | 60.6 | 76.9 | 67.8 | 35.8 | 8.2 | 30.2 | 25.8 |
| LLaMA-3.3-70B-Instruct | 62.3 | 61.6 | 73.4 | 67.0 | 12.8 | 0.5 | 79.3 | 7.4 |
| GPT-4.1-2025-04-14 | 59.8 | 61.2 | 62.2 | 61.7 | 90.8 | 2.2 | 0.0 | 7.0 |
| DeepSeek-R1 | 56.5 | 56.9 | 67.0 | 61.6 | 81.7 | 12.0 | 2.2 | 4.2 |
| Qwen3-235B-A22B | 55.5 | 56.9 | 59.6 | 58.2 | 85.2 | 6.8 | 2.9 | 5.1 |
| Claude-3.5-Sonnet | 47.7 | 49.9 | 68.9 | 57.9 | 77.7 | 9.2 | 5.2 | 8.0 |
| Grok-3-Mini (High) | 51.2 | 52.6 | 62.5 | 57.1 | 78.3 | 12.3 | 3.0 | 6.3 |
| Claude-3.7-Sonnet | 61.8 | 69.1 | 48.1 | 56.7 | 80.6 | 5.7 | 7.7 | 6.0 |
| DeepSeek-R1 (0528) | 56.0 | 58.1 | 55.1 | 56.6 | 72.5 | 19.7 | 2.2 | 5.7 |
| Qwen3-32B | 53.5 | 56.5 | 46.2 | 50.8 | 69.0 | 10.2 | 14.6 | 6.1 |
| o4-mini (Med) | 56.3 | 64.6 | 36.2 | 46.4 | 75.3 | 3.7 | 8.3 | 12.7 |
| LLaMA-3.1-8B-Instruct | 54.5 | 61.5 | 33.3 | 43.2 | 9.3 | 6.5 | 50.7 | 33.5 |
| Qwen3-30B-A3B | 54.0 | 61.5 | 30.8 | 41.0 | 60.9 | 9.9 | 20.4 | 8.8 |
| DeepSeek-V3 | 51.5 | 63.6 | 15.7 | 25.2 | 39.8 | 0.2 | 52.0 | 8.0 |
| Qwen2.5-72B-Instruct | 52.3 | 73.2 | 13.1 | 22.3 | 14.5 | 1.5 | 81.3 | 2.7 |
| o3-mini (Med) | 50.5 | 61.5 | 12.8 | 21.2 | 54.7 | 3.5 | 35.8 | 6.0 |
| Qwen2.5-Coder-32B-Instruct | 51.7 | 70.4 | 12.2 | 20.8 | 24.0 | 8.0 | 49.0 | 19.0 |
| LLaMA-4-Scout-17B-16E-Instruct | 49.0 | 55.1 | 12.2 | 19.9 | 19.8 | 2.2 | 58.5 | 19.5 |
| GPT-3.5-turbo-0125 | 45.8 | 40.8 | 9.3 | 15.1 | 13.0 | 1.8 | 37.4 | 47.9 |

patterns; however, this approach is often brittle and susceptible to overfitting, as evidenced by the trade-off between high precision and low recall in several models. The paired-function setting, by contrast, exposes the models' limited capacity for nuanced semantic reasoning. Here, even top models show a marked drop in balanced accuracy and struggle to consistently distinguish patched from vulnerable functions when differences are subtle and syntactic cues are minimal. This performance gap highlights that the comparative nature of the paired task demands deeper understanding of code semantics, intent of changes, and implications for program security.

Underlying these limitations are two interrelated challenges. First, models that excel in isolated detection frequently rely on surface-level cues, which do not transfer to the more complex comparative setting where semantic intent is crucial. Second, minor syntactic edits in code pairs often correspond to major shifts in vulnerability status, requiring the model to move beyond superficial pattern matching toward genuine comprehension of control flow, data dependencies, and defensive programming practices. The generally low P-C scores across the board reinforce the difficulty of this task, suggesting that even state-of-the-art models have not yet closed the gap between local vulnerability recognition and robust, context-aware reasoning about code security. Addressing these challenges will require the development of models with stronger program analysis capabilities and targeted training on semantically-rich vulnerability patterns.

## C.8 MULTI-MODALITY TASKS

**Model Performance and Capacity Effects.** Table 15 presents the performance of MLLMs on web development tasks across front-end frameworks, including React, Vue, Angular, and vanilla HTML/CSS. Claude-Sonnet-4, Claude-3.7-Sonnet, GPT-5 and Gemini-2.5 emerge as top performers, with Claude-Sonnet-4 achieving the highest overall performance, including superior CLIP scores (0.6907-0.8385) for Design Generation and exceptional MLLM scores for Design Edit (7.69-9.43) and Design Repair (7.37-8.14). Claude-3.7-Sonnet demonstrates strong compilation rates (0.6867-

0.9746) alongside competitive performance across all tasks, while Gemini-2.5-Flash exhibits robust performance with reliable compilation success rates consistently exceeding 0.68. A clear capacity-performance relationship emerges across model families, with larger variants consistently outperforming their smaller counterparts, particularly on complex tasks requiring code localization and visual understanding capabilities.

**Task-Specific Limitations and Performance Bottlenecks.** Our analysis reveals distinct task-specific bottlenecks that constrain MLLM effectiveness in web development scenarios. For Design Generation tasks, models encounter dual challenges: compilation errors and visual inaccuracies. Angular exhibits the lowest compilation success rates (0.6747-0.7590) compared to React and Vue (>0.83), while moderate CLIP scores (around 0.6) indicate substantial opportunities for improvement in visual fidelity. Conversely, Design Edit and Design Repair tasks are primarily limited by code localization deficiencies, as evidenced by CMS scores significantly below compilation rates. Even top-performing models like Claude-Sonnet-4 achieve CMS scores of only 0.2992-0.6588 for Design Edit and 0.3795-0.6772 for Design Repair, despite maintaining compilation rates above 0.9. These findings underscore the critical need for enhanced code understanding and precise localization capabilities in MLLMs to enable more effective web development assistance.

Table 15: Model Performance on Multi-modality Tasks under different tasks and frameworks. The top two performing results are highlighted in green ($1^{st}$) and orange ($2^{nd}$) .

| Metric | Framework | Claude | | GPT | | Gemini | | LLaMA | Qwen |
|---|---|---|---|---|---|---|---|---|---|
| | | Claude-4 | Claude-3.7 | GPT-5 | GPT-4o | Gemini-2.5 | Gemini-2.0 | LLaMA-90B | Qwen-72B |
| *Design Generation* | | | | | | | | | |
| **CLIP (%)** | React | 83.9 | 80.8 | 83.7 | 76.4 | 79.4 | 76.1 | 70.4 | 77.9 |
| | Vue | 81.2 | 83.2 | 79.0 | 77.3 | 77.8 | 69.0 | 53.2 | 68.4 |
| | Angular | 59.1 | 60.2 | 59.6 | 59.6 | 60.0 | 60.1 | 53.3 | 51.5 |
| | Vanilla | 81.2 | 81.3 | 80.6 | 76.8 | 80.2 | 75.9 | 64.0 | 76.0 |
| **Compilation (%)** | React | 99.1 | 95.4 | 97.2 | 97.2 | 91.7 | 90.8 | 94.5 | 95.4 |
| | Vue | 97.5 | 97.5 | 96.6 | 94.9 | 93.2 | 83.9 | 74.6 | 85.6 |
| | Angular | 67.5 | 68.7 | 67.5 | 71.1 | 68.7 | 71.1 | 73.5 | 62.6 |
| *Design Edit* | | | | | | | | | |
| **MLLM Score** | React | 7.7 | 8.2 | 8.3 | 8.0 | 8.4 | 7.8 | 6.2 | 8.1 |
| | Vue | 8.0 | 8.4 | 7.5 | 8.2 | 8.1 | 8.1 | 6.3 | 7.6 |
| | Angular | 8.3 | 8.0 | 8.6 | 8.3 | 8.2 | 9.1 | 5.7 | 8.2 |
| | Vanilla | 9.4 | 9.2 | 9.3 | 9.2 | 9.2 | 9.0 | 7.7 | 9.1 |
| **CMS (%)** | React | 42.1 | 46.6 | 35.1 | 52.5 | 36.6 | 37.1 | 26.4 | 44.0 |
| | Vue | 29.9 | 40.5 | 26.9 | 37.0 | 30.3 | 32.8 | 21.0 | 32.8 |
| | Angular | 65.9 | 68.3 | 59.6 | 61.0 | 58.2 | 63.9 | 47.0 | 60.2 |
| | Vanilla | 34.0 | 34.4 | 30.2 | 33.9 | 35.8 | 29.1 | 19.5 | 32.1 |
| **Compilation (%)** | React | 97.2 | 100.0 | 91.7 | 98.1 | 97.2 | 100.0 | 91.7 | 99.1 |
| | Vue | 98.1 | 98.1 | 88.6 | 94.3 | 97.1 | 95.2 | 91.4 | 93.3 |
| | Angular | 92.4 | 90.9 | 97.0 | 90.9 | 90.9 | 100.0 | 86.4 | 90.9 |
| *Design Repair* | | | | | | | | | |
| **MLLM Score** | React | 7.6 | 6.8 | 7.4 | 6.4 | 7.7 | 6.3 | 4.2 | 5.6 |
| | Vue | 7.4 | 6.6 | 7.0 | 6.3 | 7.4 | 6.1 | 4.8 | 6.0 |
| | Angular | 8.1 | 6.9 | 7.8 | 5.9 | 8.0 | 5.3 | 4.6 | 6.5 |
| | Vanilla | 7.8 | 7.2 | 7.8 | 7.1 | 7.7 | 7.3 | 5.7 | 6.9 |
| **CMS (%)** | React | 55.7 | 48.3 | 29.7 | 27.5 | 33.7 | 17.6 | 4.5 | 18.7 |
| | Vue | 40.0 | 30.7 | 31.6 | 25.2 | 36.2 | 17.8 | 5.0 | 11.3 |
| | Angular | 67.7 | 57.2 | 51.0 | 50.7 | 56.7 | 39.7 | 31.0 | 55.6 |
| | Vanilla | 38.0 | 22.9 | 10.6 | 16.4 | 19.0 | 16.3 | 3.7 | 14.5 |
| **Compilation (%)** | React | 100.0 | 100.0 | 100.0 | 100.0 | 100.0 | 100.0 | 92.9 | 92.9 |
| | Vue | 100.0 | 100.0 | 96.3 | 100.0 | 100.0 | 96.3 | 100.0 | 100.0 |
| | Angular | 100.0 | 92.9 | 100.0 | 100.0 | 100.0 | 100.0 | 78.6 | 92.9 |

## C.9 EFFECT OF MULTI-PROMPT EVALUATION

We present the evaluation of using different prompts and their average performance in Figure 6 to Figure 13. We observe substantial prompt sensitivity with varying degrees of impact across different task categories. Specifically, we can achieve the following findings:

**Prompt sensitivity exhibits task-specific patterns with varying magnitudes of impact.** Tasks such as vulnerability detection, test generation, and code review demonstrate observational performance fluctuations across different prompts. For instance, in vulnerability detection tasks, when using prompt 1, GPT-4.1 achieve much higher performance than using prompt 2 and prompt 3; while for Claude-3.5-Haiku, the performance of prompt 1 is 10 ranks lower than prompt 2. In contrast, tasks like code reasoning and code translation exhibit relatively stable performance across different prompting approaches. The vast majority of models maintain the same ranking across dif-

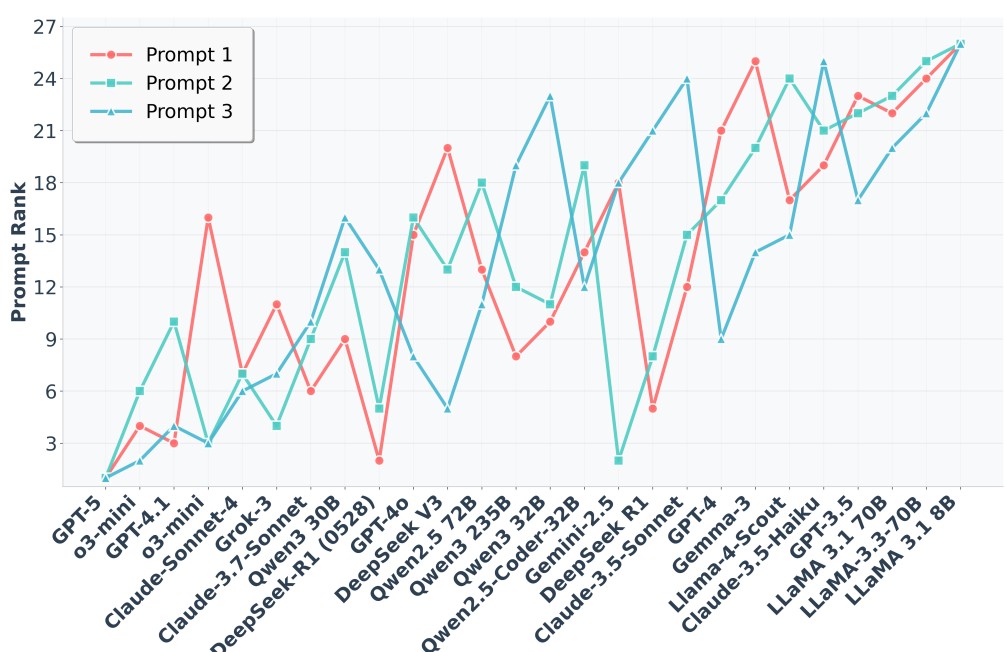

Figure 6: The performance variation of different prompt on code generation.

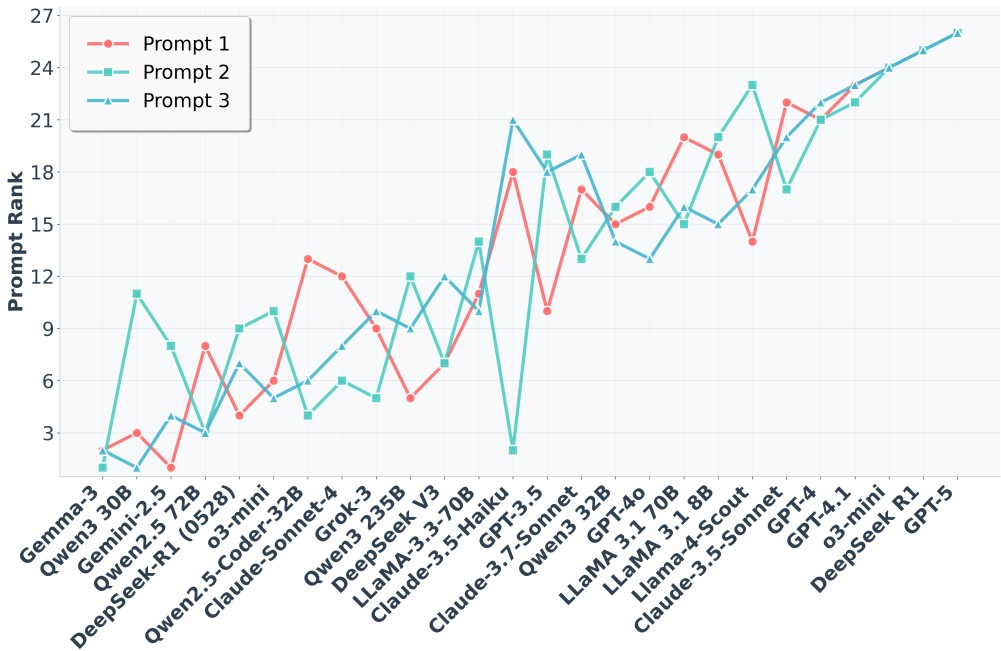

Figure 7: The performance variation of different prompt on code review.

ferent prompts, and even for the few inconsistencies, the maximum difference does not exceed 5. This suggests that different tasks are affected by evaluation prompts to varying degrees. For some tasks such as vulnerability detection, test generation, and code review, using a single prompt may introduce evaluation bias.

**Multi-prompt evaluation provides more reliable and robust assessment results.** Given the considerable performance disparities observed across prompts, our multi-prompt evaluation approach offers enhanced reliability compared to single-prompt assessments. To obtain comprehensive and

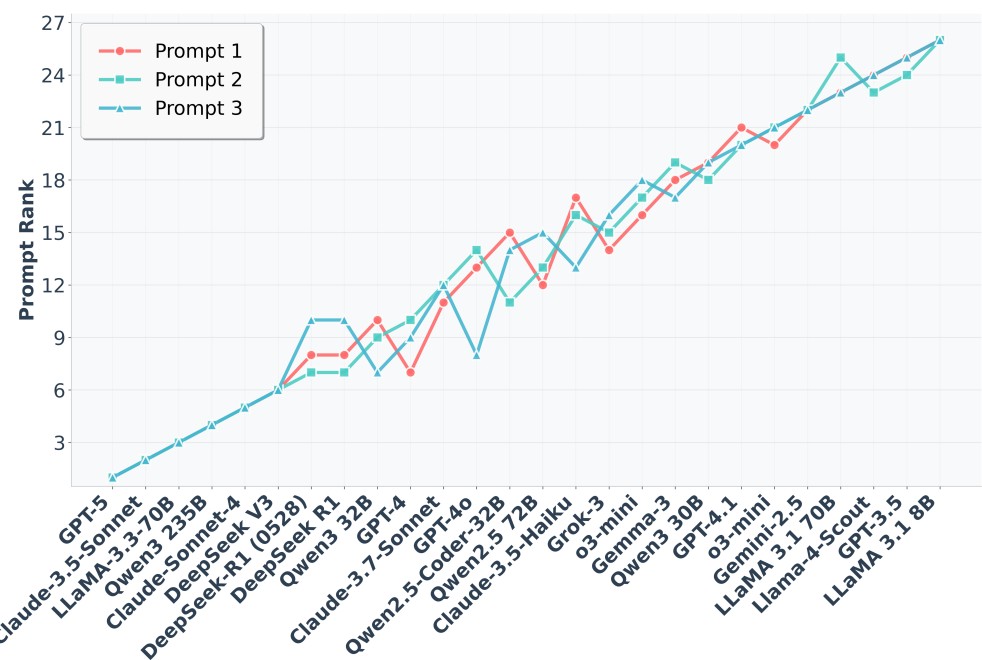

Figure 8: The performance variation of different prompt on code summarization.

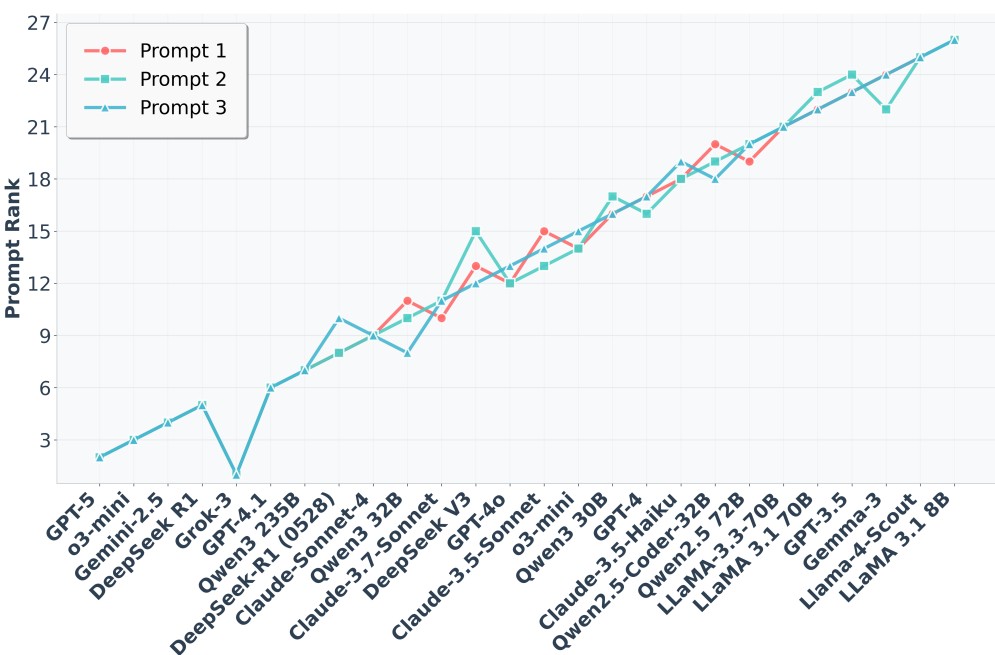

Figure 9: The performance variation of different prompt on code translation.

unbiased evaluation results, we employ multiple diverse prompts and report the averaged performance scores across all prompt variations. This methodology mitigates the potential bias introduced by any individual prompt design and provides a more accurate assessment of the models' capabilities across different programming tasks.

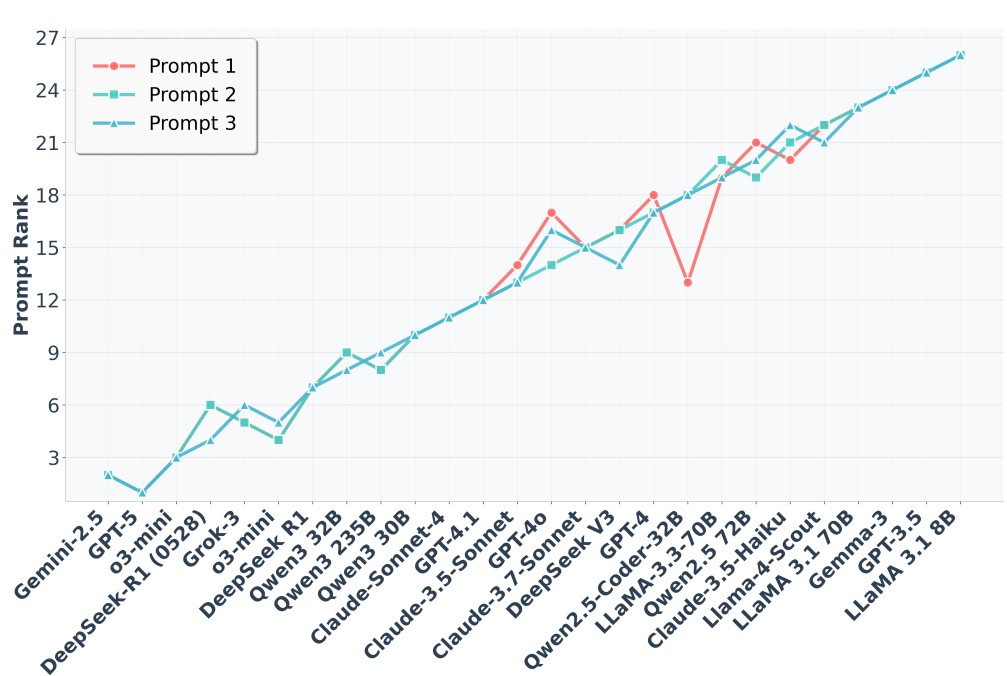

Figure 10: The performance variation of different prompt on input prediction.

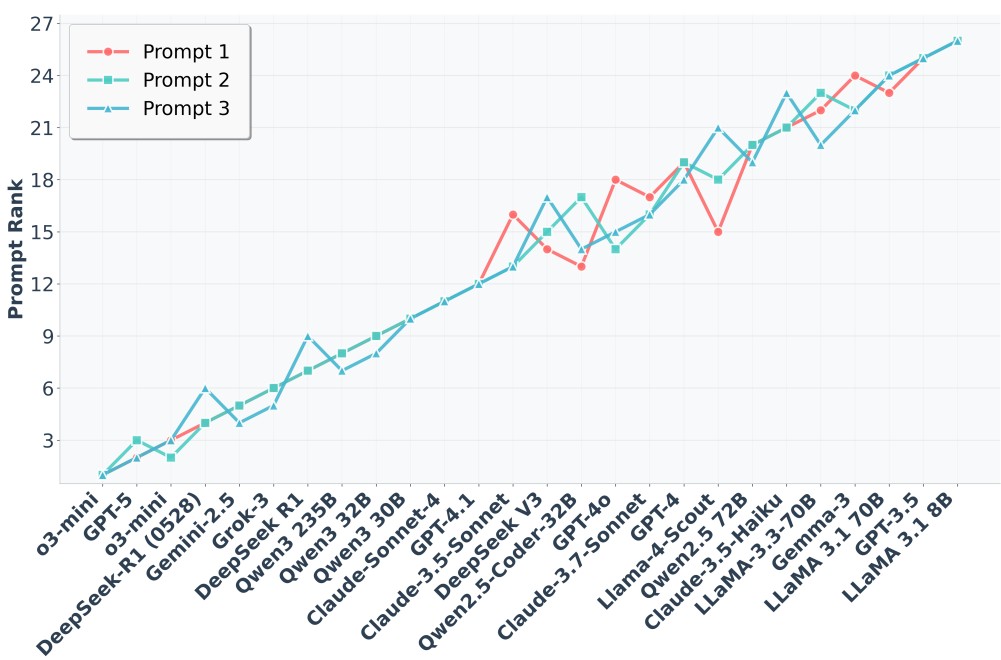

Figure 11: The performance variation of different prompt on output prediction.

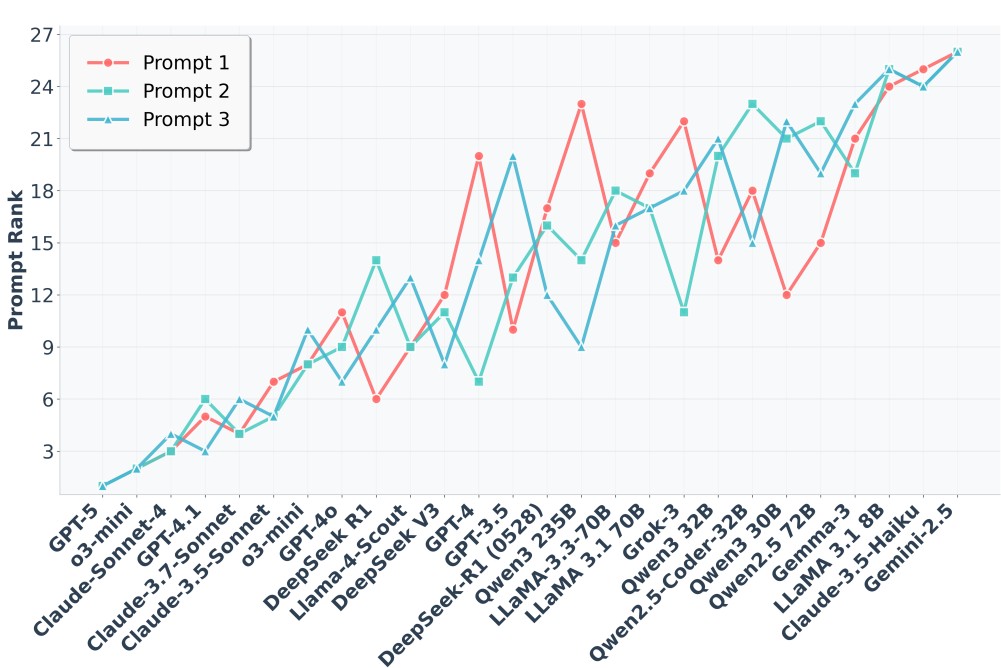

Figure 12: The performance variation of different prompt on unit test generation.

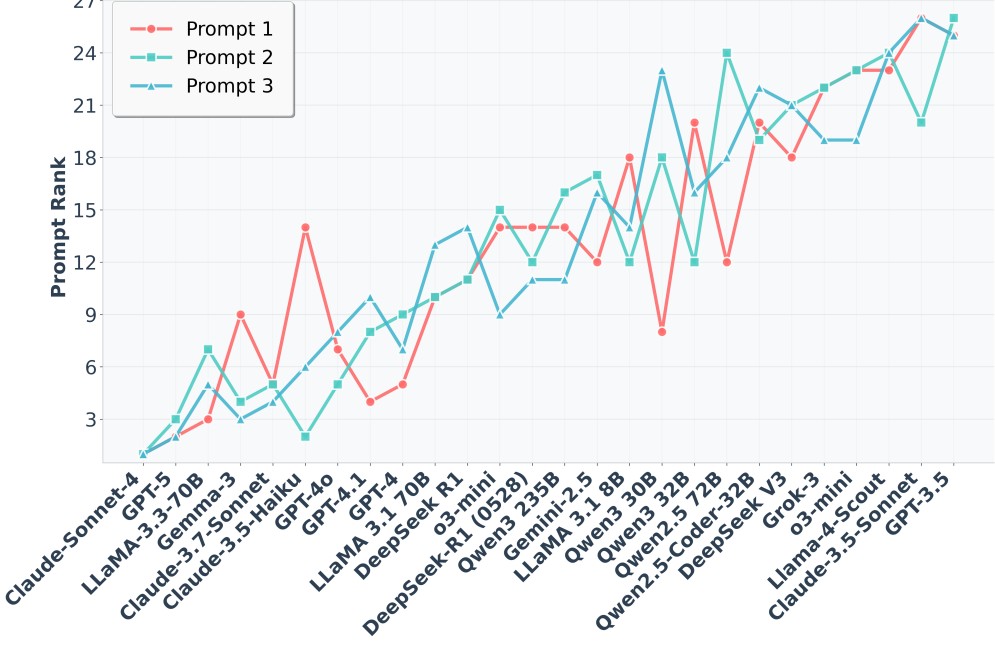

Figure 13: The performance variation of different prompt on vulnerability detection.

## D  EXTENDED RELATED WORK

### D.1  LARGE LANGUAGE MODELS FOR CODE

Large language models (LLMs) for code have rapidly advanced tasks such as code generation, completion, and reasoning. Incoder (Fried et al., 2022) unify code synthesis and editing by training on masked code segments that are moved to the end of files, enabling zero-shot code infilling and improved performance on tasks like type inference, comment generation, and variable renaming. CodeGen (Nijkamp et al., 2023) open-sources a family of models up to 16.1B parameters together with the JAXFORMER training library and shows that multi-turn prompts substantially improve program synthesis. The StarCoder model (Li et al., 2023a) with 15.5B parameters models trained on one trillion tokens and fine-tuned on Python yields StarCoder, which outperforms many predecessors. CodeT5+ (Wang et al., 2023b) improves upon existing architectures by combining encoder and decoder modules and employs a mixture of pretraining objectives. CodeLLaMA (Rozière et al., 2023) extends LLaMA2 to code and emphasizes open foundational models for code. OCTOPACK (Muennighoff et al., 2024) highlights instruction tuning in large code models, utilizing a vast dataset of Git commits, which pair code changes with human instructions, for comprehensive code understanding. WizardCoder (Luo et al., 2024) proposed the Evol-Instruct method, which rewrites simple instructions into more complex instructions, pushing performance beyond both open and closed models. DeepSeek Coder (Guo et al., 2024) introduces a family of open models trained on a 2 trillion-token corpus using a fill-in-the-blank objective and reports state-of-the-art performance among open models, even surpassing some closed models. OpenCoder (Huang et al., 2025) releases a top-tier open code LLM together with transparent training data, a complete processing pipeline, and ablation experiments.

### D.2  LARGE LANGUAGE MODELS FOR SOFTWARE ENGINEERING

Large language models play an important role in various processes of software engineering, from requirement engineering to software development, testing and maintenance.

**Requirement Engineering**. LLMs enhance requirements engineering by automating elicitation, analysis, specification, and verification processes through multi-agent frameworks. Elicitation(Ataei et al., 2025) uses multiple persona-based agents to simulate user interactions and mine requirements comprehensively, reducing costs compared to traditional user studies. SpecGen(Ma et al., 2024a) automates specification generation through conversation-driven and mutation-based approaches. Multi-phase systems like Arora et al.(Arora et al., 2024a) cover all four RE phases using specialized agents: stakeholder/engineer agents for elicitation, formatting agents for specification, evaluator agents for analysis, and validator agents for final validation. MARE(Jin et al., 2024) similarly employs stakeholder agents for elicitation, modeler agents for requirement modeling, checker agents for verification, and documenter agents for specification writing, all communicating within a shared workspace for seamless information exchange.

**Software Development**. In the realm of front-end development, MLLMs have revolutionized creative designand web development practices.DCGen (Wan et al., 2024) proposes a divide-and-conquer strategy that generates submodule code separately before assembling complete webpages. DeclarUI (Zhou et al., 2024) combines element segmentation with page transition graphs to prompt MLLMs for mobile app UI generation with navigation logic. UICopilot (Gui et al., 2025) adopts a hierarchical approach by first generating HTML tree structures, then progressively generating UI components. LayoutCoder (Wu et al., 2025) introduces a layout-aware MLLM framework specifically designed to comprehend complex UI layouts and preserve layout fidelity in generated code. DesignRepair (Yuan et al., 2025) presents a dual-stream, knowledge-driven approach that leverages LLMs to detect and repair design quality issues in front-end code. Interaction2Code (Xiao et al., 2024) and DESIGNBENCH (Xiao et al., 2025) add interaction-aware generation and repair. LLM-based agents for end-to-end software development adopt classic software process models to standardize development workflows: **(A) Waterfall Process Model**:Most existing agents (e.g., AISD (Zhang et al., 2024a), LCG (Lin et al., 2024), ChatDev (Qian et al., 2023), CTC (Du et al., 2024), Self-Collaboration (Dong et al., 2024)) follow the linear waterfall model (Royce, 1987) with sequential phases (requirements engineering, design, implementation, testing, deployment, maintenance), while some extend it with iterative feedback loops for quality assurance and MetaGPT (Hong et al., 2023) integrates human-like Standardized Operating Procedures (SOPs) for role-based collab-

oration; **(B) Agile Development**: Some agents explore agile methodologies including Test-Driven Development (TDD) (Lin et al., 2024), which prioritizes writing tests before coding through test-implement-refine cycles, and Scrum (Lin et al., 2024; Nguyen et al., 2025), which breaks development into iterative sprints, with experiments showing Scrum achieves the best and most stable performance on function-level code generation benchmarks, followed by TDD (Lin et al., 2024).

**Software Testing**. Software testing checks isolated software units (e.g., methods or classes) to quickly identify and localize bugs (Yang et al., 2024a). While LLMs like ChatGPT can generate unit tests with decent readability and usability (Yuan et al., 2023), they still exhibit compilation/execution errors and limited coverage. Recent LLM-based agents address these issues through iterative refinement: (A) **Fixing Compilation/Execution Errors**. ChatTester (Yuan et al., 2023) and ChatUniTest (Xie et al., 2023) iteratively collect error messages and refine buggy test code; (B) **Increasing Coverage**. TELPA (Yang et al., 2024a) employs backward/forward program analysis and counter-example sampling with CoT strategy to enhance coverage of hard-to-reach branches; (C) **Enhancing Fault Detection**. MuTAP (Dakhel et al., 2024) uses mutation testing feedback, where surviving mutants guide LLM refinement to improve test cases' bug detection capabilities.

**Software Operation and Maintenance**. LLM-based agents for end-to-end software maintenance follow a common pipeline to automatically resolve real-world GitHub issues through multiple phases: **(A) Preprocessing** – agents prepare repository knowledge (e.g., RepoUnderstander (Ma et al., 2024b) builds knowledge graphs, Agentless (Xia et al., 2024) creates hierarchical structures); **(B) Issue Reproduction** – agents generate test scripts to trigger unexpected behaviors when reproduction tests are unavailable (e.g., SWE-agent (Yang et al., 2024b), MASAI (Arora et al., 2024b) with two-stage template-based approach); **(C) Issue Localization** – agents identify relevant code elements using: (C.1) retrieval-based strategies via BM25 similarity (Tao et al., 2024b), (C.2) navigation-based approaches with search interfaces (Yang et al., 2024b; Arora et al., 2024b; Zhang et al., 2024b; Xia et al., 2024), (C.3) spectrum-based fault localization calculating suspiciousness scores from test coverage (Zhang et al., 2024b; Chen et al., 2024a), and (C.4) simulation using Monte Carlo Tree Search (Ma et al., 2024b); **(D) Task Decomposition** – breaking issues into fine-grained sub-tasks (Tao et al., 2024b; Ma et al., 2024b); **(E) Patch Generation** – creating fixes for localized suspicious code elements (Xia et al., 2024); **(F) Patch Verification** – validating correctness through code review (Tao et al., 2024b), static checking for syntax (Zhang et al., 2024b; Ma et al., 2024b; Arora et al., 2024b; Xia et al., 2024; Yang et al., 2024b), and dynamic checking via test execution (Chen et al., 2024a; Arora et al., 2024b; Xia et al., 2024); **(G) Patch Ranking** – identifying highest-probability correct patches using ranker agents (Arora et al., 2024b) or majority voting (Xia et al., 2024). These approaches are evaluated on benchmarks like SWE-bench (Jimenez et al., 2023) containing real-world GitHub issues across popular Python repositories.

### D.3 LARGE LANGUAGE MODELS EVALUATION

Recent years have witnessed substantial efforts in building benchmarks to evaluate the capabilities of LLMs on code-related tasks. Early benchmarks such as HUMANEVAL (Chen et al., 2021), MBPP (Austin et al., 2021), and APPS (Hendrycks et al., 2021), as well as extensions like HUMANEVAL+ (Liu et al., 2023), focused on evaluating function-level code generation performance. Due to the rapid advancement of code-oriented LLMs, more challenging and realistic benchmarks have been proposed. LIVECODEBENCH (Jain et al., 2025) continuously collects new contest problems from LEETCODE, ATCODER, AND CODEFORCES, offering a contamination-free setting for evaluating code generation. CCTEST (Li et al., 2023b) focuses on real-world code completion tasks, efficiently testing and fixing inconsistency bugs in real products including Github copilot. BIGCODEBENCH (Zhuo et al., 2024) focuses on library-aware code generation, assessing models' ability to handle diverse libraries across multiple domains. INFIBENCH (Li et al., 2024d) provides the first large-scale QA benchmark curated from Stack Overflow questions, challenging LLM capability in realistic software engineering contexts. SWE-BENCH (Jimenez et al., 2023) evaluates models on practical software engineering tasks by requiring them to resolve GitHub issues through multi-file code modifications in realistic repositories. DYCODEEVAL (Chen et al.) introduces dynamic benchmarking that deliberately controls contamination to assess reasoning capabilities in code LLMs. DYNACODE (Hu et al., 2025) proposes a dynamic complexity-aware framework that automatically adjusts problem difficulty, enabling finer-grained and adaptive evaluation of code generation skills. EVOCODEBENCH (Li et al., 2024a) is an evolving benchmark tightly aligned with real-world GitHub repositories. It continuously incorporates new commits and domain-specific tasks

to prevent leakage and maintain relevance. MMCode (Li et al., 2024c) and SWE-BENCH MUL-TIMODAL (Yang et al., 2024c) extends the original SWE-bench by adding visual inputs such as screenshots, UI mockups, design files, showing that current multimodal code models suffer large performance drops without visual context and highlighting a generalization gap to visual software domains. CODEREVAL (Yu et al., 2024) and DevEval (Li et al., 2024b) emphasize repository-level code generation drawn from real open-source repositories. PPM (Chen et al., 2024d) presents an automated pipeline that uses LLMs themselves to synthesise diverse programming problems, facilitating scalable and varied benchmark creation. Recent efforts have also evaluated and proposed improvements for LLM-based competitive programming generation using real 2024 ICPC/CCPC contest problems (Wei et al., 2025).

### D.4 ROBUSTNESS OF CODE LLMs

Robustness in Code LLMs has become an important research question, as models that excel on standard benchmarks like HumanEval and MBPP often degrade sharply when exposed to real-world variations such as semantically-preserving code transformations and natural-language prompt perturbations (Mastropaolo et al., 2023; Zhuo et al., 2023). ReCode (Wang et al., 2023a) introduced the first robustness benchmark for code generation, applying 12 functionality-preserving perturbations including variable renaming, unused code insertion, and control-flow flattening. CCTest (Li et al., 2023b) focuses on real-world code completion tasks, efficiently testing and fixing inconsistency bugs in real products including Github copilot. NLPerturbator (Chen et al., 2024c) shifted focus to natural-language prompt variations, categorizing different real-world perturbation types derived from practitioner surveys and showing average pass@1 drops of 4.8–6.1% across StarCoder, CodeLlama, and DeepSeek-Coder. RobGen (Li et al., 2025) revealed that 35% of LLM-generated code is less robust than human references due to missing conditional checks and proposed a lightweight decoding-time framework that boosts robustness by 10% while preserving functional correctness. RobuNFR (Lin et al., 2025) extended evaluation to non-functional requirements including design, readability, reliability, performance, demonstrating that expressing the same NFR differently causes high output variability and up to 39% correctness loss. Recent work CodeCrash (Lam et al., 2025) comprehensively test LLMs in code reasoning under structural and NL-embedded perturbations. To mitigate this problem, many techniques such as structure-aware model training and robustness training are also introduced to improve code LLM's robustness (Tipirneni et al., 2024; Pei et al., 2022; Oh & Yoo, 2024).

## E LIMITATION AND FUTURE WORK

To further enhance the comprehensiveness and practical implication of this benchmark, we have planned several key directions for future work.

**Expanding Task Diversity:** While our current benchmark covers a range of fundamental tasks, we plan to introduce more complex and realistic challenges to better assess the advanced capabilities of LLMs (Jimenez et al., 2023; Wong et al., 2024; Peng et al., 2024). For example, code debugging which evaluate a model's ability to not only identify and locate errors but also to explain the underlying logic flaws in the code moves beyond simple code correction to test a model's deeper reasoning and diagnostic skills. Furthermore, tasks like issue resolution tasks (Jimenez et al., 2023; Yang et al., 2024b), require models to analyze entire problem contexts from sources like GitHub issues—including natural language descriptions, error logs, and user comments—and then propose and justify a complete code-based solution. This will measure a model's ability to handle repository-level software maintenance challenges that are common in real-world development.

**Introducing Multi-Level Granularity Evaluation:** Currently, our evaluation such as code generation and translation are primarily assessed at the function level. However, real-world software engineering operates on much larger scales. We plan to extend our evaluation to higher levels of abstraction to address this gap. This includes introducing repository-level tasks (Li et al., 2024b), which will require models to generate or translate complete source files containing multiple classes and functions. In future work, we aim to evaluate performance at multiple levels, challenging models to perform complex operations like implementing new features based on high-level requirements or executing large-scale refactoring across an entire codebase.

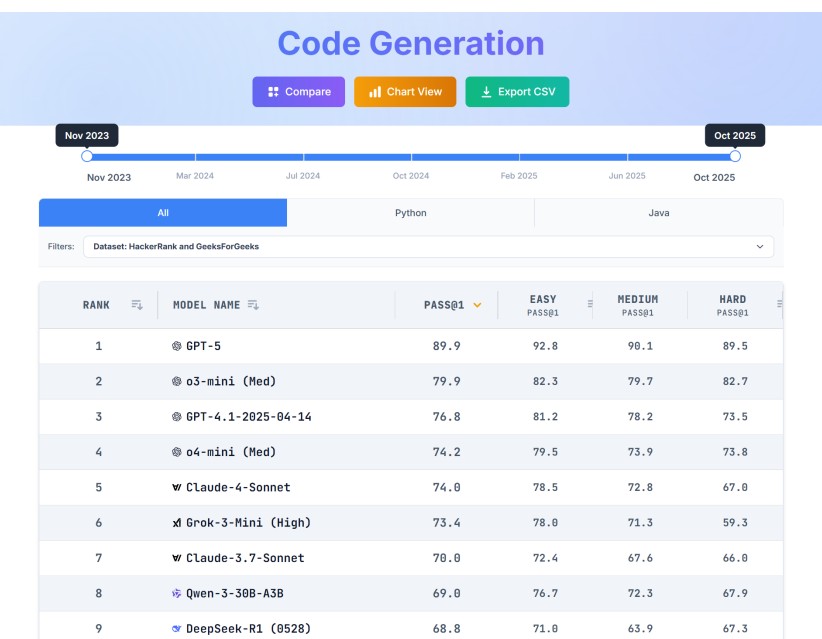

Figure 14: The leaderboard page for different tasks.

**Evaluating Diverse Prompting Strategies:** The effectiveness of a large language model is significantly influenced by the prompting strategy used. We will conduct a more systematic investigation into the impact of various prompting techniques—from straightforward zero-shot and few-shot methods (Gao et al., 2023) to more complex approaches like Chain-of-Thought (Wei et al., 2022) and agentic workflows (Xia et al., 2024). This will provide valuable, practical guidance on how to most effectively elicit high-quality outputs from models for different coding tasks, ultimately helping to define best practices for their application.

**Enhancing Security Evaluation:** Given the increasing deployment of LLMs in production environments, we plan to expand our security evaluation framework beyond current vulnerability detection tasks. Our assessment will cover various critical dimensions such as vulnerability assessment of security flaws in generated code, privacy protection evaluation to prevent sensitive data exposure and regulatory violations, bias detection and mitigation in generated algorithms, authorship and intellectual property compliance. This will establish essential safeguards for responsible LLM deployment in software engineering practices.

**Establishing a Regularly Update:** To combat the persistent issue of data contamination, where a model's training data may inadvertently include benchmark samples, we will implement a dynamic data collection and refreshment process (Jain et al., 2025; Zhang et al., 2025). By periodically sourcing new data from the latest open-source projects and programming platforms, we can ensure the benchmark remains fair and relevant. This regularly updating will help guarantee that we are assessing a model's true generalization capabilities on previously unseen code, thereby maintaining the long-term integrity and credibility of our evaluation.

## F ONLINE LEADERBOARD

Our online leaderboard is available at `https://code-treat.vercel.app/`.

In the leaderboard, we provide an interactive interface to view detailed results of each task, visualize the model performance with timeline and compare the ability of different models.

As shown in Figure 14, the leaderboard page displays a comprehensive ranking table of each task. Users can view model performance across multiple evaluation metrics for each task. The interface allows users to filter results by different time periods and switch between various tasks such as vulnerability detection. Each model entry shows detailed performance statistics.

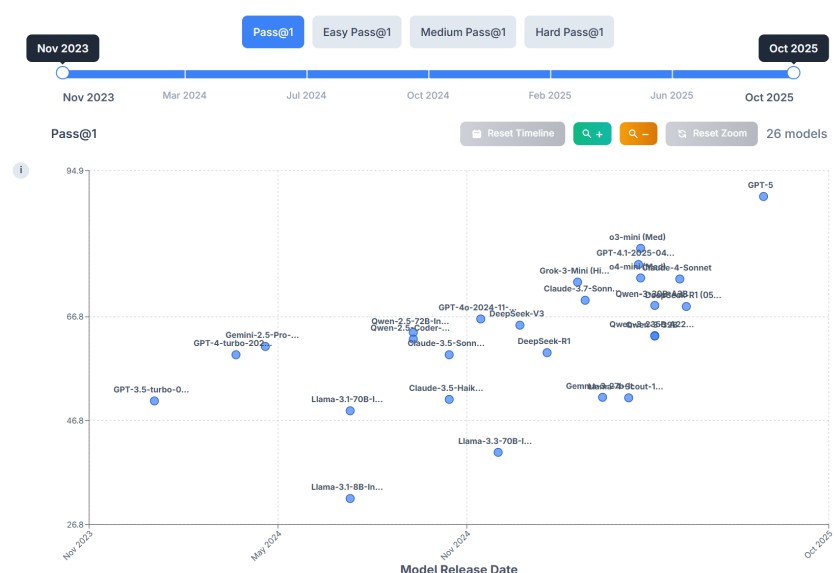

Figure 15: The model performance timeline page.

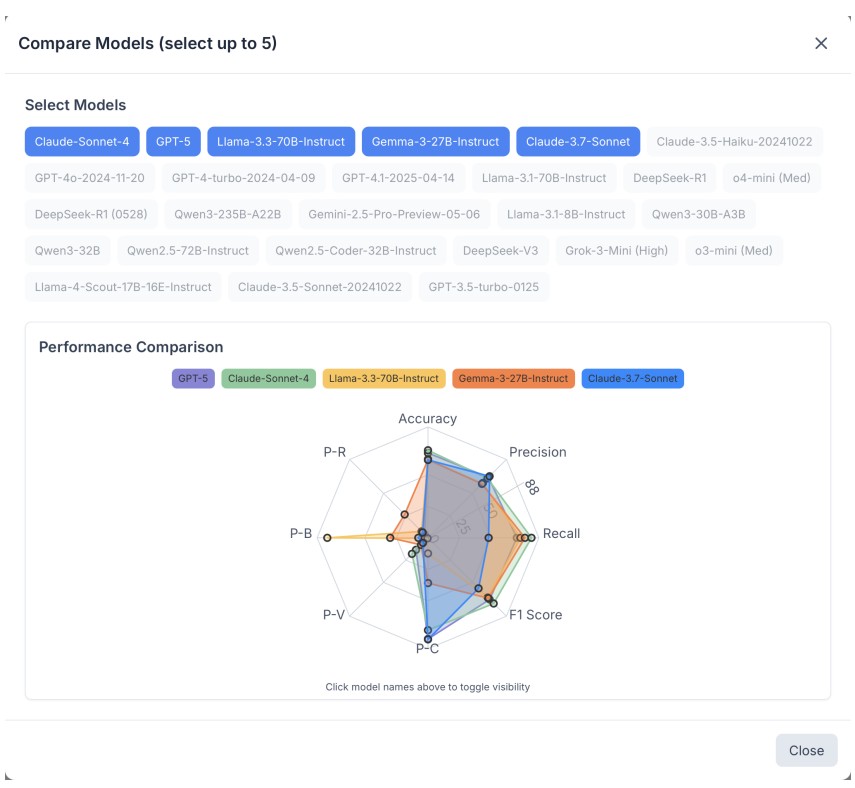

Figure 16: The leaderboard page for different tasks.

Figure 15 shows the model performance timeline comparison page, which provides a temporal view of how different models have evolved and improved over time. This scatter plot visualization plots model accuracy against release dates, with different colored points representing various model families. Users can interact with the timeline to explore historical trends and identify breakthrough moments in model development, making it easier to understand the progression of the field.

For more detailed analysis, Figure 16 shows our model comparison interface, which allows users to select different models for side-by-side comparison. The radar chart visualization displays multiple performance metrics simultaneously, including accuracy, precision, recall, F1 score, and other relevant measures. This enables researchers to conduct comprehensive comparative analysis and identify the strengths and weaknesses of different approaches across various evaluation dimensions.

## G  PROMPT DETAILS

### G.1  CODE GENERATION

**CODE GENERATION**

**SYSTEM PROMPT**

```
You are a helpful assistant.
```

**User Prompt Prompts 1**

```
Please provide a self-contained {PL} script that solves the
following problem in a markdown codeblock:{problem_description}
Your task is to complete the function {function_signatures} {
class_msg}
```

**USER PROMPT 2**

```
Write a {PL} function {function_signatures} {class_msg} to
solve the following problem:{problem_description}
```

**USER PROMPT 3**

```
You are an expert {PL} programmer. You will be given a question
 (problem_description) and will generate a correct {PL} program
 that matches the specification and passes all tests, You will
NOT return anything except for the program.
### Question
{problem_description}
{starter_code_msg}
### Answer:
(use the provided format with backticks)
```

### G.2  CODE SUMMARIZATION

## CODE SUMMARIZATION

### SYSTEM PROMPT

You are a helpful assistant.

### USER PROMPT 1

Please generate a short comment in one sentence for the
following function:
{code}

### USER PROMPT 2

Please write a brief comment in one sentence for the following
function:
{code}

### USER PROMPT 3

Kindly provide a concise comment in one sentence for the
following function:
{code}

## CODE SUMMARIZATION – LLM AS JUDGE

### SYSTEM PROMPT

You are a helpful assistant.

**USER PROMPT**

```
Here is a piece of code with corresponding comments. Please
rate each comment on a scale from 1 to 5, where a higher score
indicates better quality. A good comment should: 1) accurately
summarize the function of the code; 2) be expressed naturally
and concisely, without burdening the developer with reading; 3)
 help the developer understand the code quickly: Your answer
should be in the JSON format JSON: {"Comment 0": {your rating},
 "Comment 1": {your rating}, ..., "Comment n": {your rating}}.
Code:

Commnet 0: <human baseline summary>
Comment 1: <summary written by LLM_1>
Comment 2: <summary written by LLM_2>
...
Comment n: <summary written by LLM_3>
```

## G.3  CODE TRANSLATION

**CODE TRANSLATION**

**SYSTEM PROMPT**

```
You are a code translation system.
```

**USER PROMPT 1**

```
Translate {SL} To {TL}:
{SC}
```

**USER PROMPT 2**

```
Translating {SL} To {TL} ensures that {TL} code can be executed
:
{SC}
```

**USER PROMPT 3**

```
Please provide the {TL} translation for the following {SL} code
:
{SC}
```

### G.4 CODE REVIEW GENERATION

---

**CODE REVIEW GENERATION**

**SYSTEM PROMPT**

```
You are a code reviewer specializing in analyzing and providing
 feedback on code. Please provide your review comments in the
following JSON format: {"comments": "<your comments>"}.
```

**USER PROMPT 1**

```
Below is an instruction that describes a task, paired with an
input that provides further context. Write a response that
appropriately completes the request.
### Instruction:
Review the given code and provide a constructive code review
comment.
### Input:
The code/diff hunk is:
'{diff_hunk}'
### Response:
{{comment}}
```

**USER PROMPT 2**

```
Below is an instruction describing a task, along with
additional context. Your job is to generate a complete response
 based on the following request:
### Instruction:
Examine the provided code and offer constructive feedback.
### Input:
The code or diff hunk is:'{diff_hunk}'
### Response:
{{comment}}
```

**USER PROMPT 3**

```
Below is a task description along with additional context.
Provide an answer that fulfills the request.
### Instruction:
Examine the given code and deliver a helpful code review
comment.
### Input:
The code (or diff snippet) is:
'{diff_hunk}'
### Answer:
{{comment}}
```

**CODE REVIEW GENRATION – LLM AS JUDGE**

**SYSTEM PROMPT**

You are a smart code reviewer. You will be asked to grade a generated code review. You can mimic answering them in the background 10 times and provide me with the most frequently appearing answer. Furthermore, please strictly adhere to the output format specified in the question. There is no need to explain your answer. Please output your final answer in the following JSON format: {"grade": <your grade>}. The grade should be an integer between 1 and 5, inclusive.

**USER PROMPT**

I am going to give you a generated code review as well as its reference review. You should grade the generated review by comparing it to the reference review, and output a grade based on the following criteria:
1. If the generated review is identical to the reference review, Grade=5;
2. If the generated review is essential equivalent to the reference review although their expressions are not identical, Grade=4;
3. lf the generated review explicitly and correctly specifies some comments/suggestions presented in the reference review, Grade=3;
4. If the generated review is only loosely related to the reference review, Grade=2;
5. If the generated review is completely unrelated to the reference review in semantics, Grade=1.
Please NOTE that you should only output a grade without any explanation.
**Generated Code Review**:
<LLM generated-review>
**Reference Code Review**:
<human ground truth reference-review>

## G.5 CODE REASONING

### G.5.1 INPUT PREDICTION

**INPUT PREDICTION**

**SYSTEM PROMPT**

You are a helpful assistant. Please provide your input prediction in the following JSON format: {"input_prediction": "<your input prediction>"}.

**USER PROMPT 1 – JAVA**

You are given a piece of code containing Java method `f` (already defined elsewhere) and a masked `public static void main` template where all inputs are `??`. Your task is to identify suitable inputs for each `??` with concrete, valid values so that, when combined with the existing class that contains `f`, the program compiles and the assertion in `main` holds true. No extra information except the filled `public static void main` code should be included in your submission.
Code:
{function}
Masked main template:
{assertion_query}

**USER PROMPT 2 – JAVA**

You are provided with a Java method `f` (defined elsewhere) and a `public static void main` template with input placeholders marked as `??`. Your task is to replace each `??` with concrete, valid values so that the program compiles and the assertion in `main` passes when run together with the class containing `f`. Submit only the completed `public static void main` code no additional explanation or information.
Code:
{function}
Masked main template:
{assertion_query}

**USER PROMPT 3 – JAVA**

You are given a piece of code that includes a Java method `f` (defined elsewhere) and a `public static void main` template with masked inputs marked as `??`. Your task is to replace each `??` with concrete, valid input values such that, when combined with the existing class containing `f`, the program compiles successfully and the assertion in `main` passes. Your response must include **only** the completed `public static void main` code no additional explanation or information.
Code:
{function}
Masked main template:
{assertion_query}
Code:
{function}
Masked Main Template:
{assertion_query}

**USER PROMPT 1 – PYTHON**

You will be provided with a function `f` and a specified input
format `inputs = ??`. Your task is to identify a suitable input
for the function `f` that, when passed, results in the
specified output. The solution should complete the final line
of code to ensure the program executes error-free. Feel free to
use any correct input, and note that the function f may
incorporate predefined classes or data types. No extra
information should be included in your submission.
{function}
{assertion_query}

**USER PROMPT 2 – PYTHON**

You will be provided with a function `f` and a specified output
in the format `inputs = ??`. Your task is to complete the
final line of code so that the program executes error-free by
identifying an input that, when passed to `f`, results in the
specified output. There could be several correct inputs, and
you may choose any one of them to complete the line. Do not
include any extra information.
{function}
{assertion_query}

**USER PROMPT 3 – PYTHON**

You are provided with a function named `f` and an expression
formatted as `inputs = ??`. Complete the expression by
determining any possible input that, when passed to function `f
`, will produce the specified output. Ensure the final line of
code runs error-free. Note that there might be several valid
inputs; you only need to provide one. Avoid including any extra
information.
{function}
{assertion_query}

### G.5.2   OUTPUT PREDICTION

**OUTPUT PREDICTION**

**SYSTEM PROMPT**

You are a helpful assistant. Please provide your output
prediction in the following JSON format: {"output_prediction":
"<your output prediction>"}.

**USER PROMPT 1 – JAVA**

You are given a piece of code containing Java method `f` ( already defined elsewhere) and a masked `public static void main` template where the assertion's expected output(s) is/are `??`. Your task is to replace that `??` with concrete, valid value(s) so that, when combined with the existing class containing `f`, the program compiles and the assertion in `main ` holds true. No extra information except the filled `public static void main` code should be included in your submission.
Code:
{function}
Masked main template:
{assertion_query}

**USER PROMPT 2 – JAVA**

You are given a Java code snippet containing a method `f` ( defined elsewhere) and a `public static void main` template in which the expected output for an assertion is represented by `??`. Your task is to replace each `??` with specific, valid value(s) so that the program compiles successfully and the assertion in `main` passes. Your submission must include only the completed `public static void main` code    do not add any extra explanation or content.
Code:
{function}
Masked main template:
{assertion_query}

**USER PROMPT 3 – JAVA**

You are provided with a piece of code that includes a Java method `f` (already defined elsewhere) and a `public static void main` template where the expected output(s) in the assertion is/are marked as `??`. Your task is to replace each `??` with concrete, valid value(s) such that the program compiles and the assertion in `main` evaluates to true when combined with the given class containing `f`. Submit only the completed `public static void main` code    no additional information.
Code:
{function}
Masked main template:
{assertion_query}

**USER PROMPT 1 – PYTHON**

Based on the given code, which may contain errors, complete the
 assert statement with the output when executing the code on
the given test case. Do not output any extra information, even
if the function is incorrect or incomplete.
{function}
{assertion_query}

**USER PROMPT 2 – PYTHON**

Please analyze the provided code, which might have errors, and
finish the assert statement by specifying the expected result
when the code is run with the specified test case. Ensure your
response includes only the necessary information, without any
additional comments about possible inaccuracies or
incompleteness in the function.
{function}
{assertion_query}

**USER PROMPT 3 – PYTHON**

Given the function below, which may have errors, complete the
assert statement to reflect the output when running the
function with the provided test case. Only include the
necessary information in your response; do not add anything
extra, regardless of the function's correctness or completion
status.
{function}
{assertion_query}

## G.6 TEST GENERATION

**UNIT TEST GENERATION**

**SYSTEM PROMPT**

You are a professional Python unit test writer. Always output
valid pytest test code wrapped in
```python
{unit_test_code}
```
Cover normal cases, invalid ts, boundary conditions, and
exception paths.

**USER PROMPT 1**

```
{focal_method_code_with_context}
```
Write pytest unit tests for the focal method `{focal_method_name}` from the module `{module_name}`.

**USER PROMPT 2**

Here is the relevant code:
```
{focal_method_code_with_context}
```
Your task is to create pytest unit tests that thoroughly test the method `{focal_method_name}` defined in module `{module_name}`.

**USER PROMPT 3**

```
{focal_method_code_with_context}
```
Generate a set of pytest unit tests for `{focal_method_name}`. The method comes from the module `{module_name}`. Ensure coverage of normal cases, edge cases, and error handling.

## G.7 VULNERABILITY DETECTION

**VULNERABILITY DETECTION**

**SYSTEM PROMPT**

You are a security expert analyzing code for vulnerabilities. Be precise and accurate in your analysis.

### USER PROMPT 1 – SINGLE FUNCTION

```
Please analyze the following code:
```
{code}
```
Please indicate your analysis result with one of the options:
(1) YES: A security vulnerability detected.
(2) NO: No security vulnerability.

Only reply with one of the options above. Do not include any
further information.
```

### USER PROMPT 2 – SINGLE FUNCTION

```
Kindly review the code snippet below:
```
{code}
```
Then, respond exclusively with one of the following options:
(1) YES: A Security Vulnerability Detected.
(2) NO: No Security Vulnerability.

Please provide only one of the two responses, without any extra
 details.
```

### USER PROMPT 3 – SINGLE FUNCTION

```
Analyze the code snippet below:
```
{code}
```
After reviewing, choose exactly one of the following responses:
(1) YES: A security vulnerability detected.
(2) NO: No security vulnerability.

Respond only with one of these options and no additional
information.
```

### USER PROMPT 1 – PAIRED FUNCTIONS

Please analyze the following two code snippets separately. For each snippet, provide exactly one answer either (1) or (2) from the options below:

For `code1`:
- (1) YES: A security vulnerability detected.
- (2) NO: No security vulnerability.

For `code2`:
- (1) YES: A security vulnerability detected.
- (2) NO: No security vulnerability.

Ensure that your reply consists of two separate lines, one for `code1` and one for `code2`, with no additional information.

Code snippet for `code1`:
```
{code1}
```

Code snippet for `code2`:
```
{code2}
```

Please output in the following JSON format:
```json
{{"code1": "<YES / NO>",
 "code2": "<YES / NO>"}}
```

## USER PROMPT 2 – PAIRED FUNCTIONS

Analyze the two provided code snippets individually. For each
snippet, choose one answer from the following options:

**For `code1`:**
– (1) YES: A security vulnerability detected.
– (2) NO: No security vulnerability.
**For `code2`:**
– (1) YES: A security vulnerability detected.
– (2) NO: No security vulnerability.

Your response should consist of two separate lines one for `
code1` and one for `code2` with no extra details. Use the
following JSON format for your output:

```json
{{"code1": "<YES / NO>",
 "code2": "<YES / NO>"}}
```

Here are the code snippets:

**Code snippet for `code1`:**
```
{code1}
```

**Code snippet for `code2`:**
```
{code2}
```

**USER PROMPT 3 – PAIRED FUNCTIONS**

```
Examine the two provided code snippets independently. For each
snippet, select one answer either (1) or (2) from the choices
below:

For `code1`:
- (1) YES: A security vulnerability was detected.
- (2) NO: No security vulnerability found.

For `code2`:
- (1) YES: A security vulnerability was detected.
- (2) NO: No security vulnerability found.

Your response should include exactly two lines: one for `code1`
 and one for `code2`, without any extra details.

Here are the code snippets:

For `code1`:
```
{code1}
```

For `code2`:
```
{code2}
```

Return your answer in this JSON structure:
```json
{{"code1": "<YES / NO>",
 "code2": "<YES / NO>"}}
```
```

## H  BENCHMARK STATISTICS

The data statistics of our evaluation data is as follows:

| Task | Code Generation | Code Summarization | Code Translation | Code Reasoning | Code Review | Test Generation | Vulnerability Detection | Multi-modal |
|---|---|---|---|---|---|---|---|---|
| Data Volume | 1664 | 2000 | 744 | 2000 | 2000 | 200 | 400 | 900 |

Table 16: Task-wise data distribution

| Language | C | C++ | C# | Java | Go | JavaScript | TypeScript | PHP | Python | Ruby | Html | Css |
|---|---|---|---|---|---|---|---|---|---|---|---|---|
| Data Volume | 400 | 800 | 400 | 2604 | 400 | 1072 | 400 | 400 | 2804 | 400 | 228 | 228 |

Table 17: Language-wise data distribution

## I  ANALYSIS OF LLM-AS-A-JUDGE

One potential problem of using LLM-as-a-judge as the evaluation metric is that it may have model preference and introduce fairness problem. To mitigate this problem, we analyze the influence of different judge models (Gemini-2.5-Flash and Claude-4-Sonnet) on evaluation. We present the results in Table 18 and Table 19. From these tables, we can find that different judging models exert

a certain impact on the result rankings. But the three models generally maintain a certain degree of consistency. For example, on the code review dataset, the Kendall's W coefficient is 0.4936, indicating moderate consistency. Therefore, to provide a more reliable and convincing evaluation results, we use the average results of these three models as the evaluation results.

Table 18: Influence of different judge models on Code Summarization

| Model | Gemini-2.5-Flash | Rank | Claude-4 | Rank | GPT-4o | Rank |
|---|---|---|---|---|---|---|
| Claude-3.5-Haiku-20241022 | 41.82 | 19 | 57.82 | 16 | 85.24 | 15 |
| Claude-3.5-Sonnet-20241022 | 42.74 | 12 | 59.20 | 4 | 96.54 | 2 |
| Claude-3.7-Sonnet | 43.44 | 1 | 59.62 | 3 | 88.10 | 11 |
| Claude-Sonnet-4 | 43.44 | 1 | 60.38 | 1 | 93.76 | 5 |
| DeepSeek-R1 | 42.04 | 17 | 58.72 | 7 | 90.64 | 7 |
| DeepSeek-R1 (0528) | 41.98 | 18 | 58.72 | 7 | 90.64 | 7 |
| DeepSeek-V3 | 42.30 | 15 | 57.86 | 14 | 92.82 | 6 |
| GPT-3.5-turbo-0125 | 42.66 | 13 | 55.00 | 25 | 71.18 | 24 |
| GPT-4-turbo-2024-04-09 | 42.28 | 16 | 57.30 | 19 | 89.94 | 10 |
| GPT-4.1-2025-04-14 | 42.42 | 14 | 57.30 | 19 | 80.28 | 21 |
| GPT-4o-2024-11-20 | 42.82 | 9 | 57.84 | 15 | 87.88 | 12 |
| GPT-5 | 40.50 | 25 | 58.34 | 9 | 98.28 | 1 |
| Gemini-2.5-Pro-Preview-05-06 | 43.16 | 4 | 58.76 | 6 | 78.88 | 22 |
| Gemma-3-27B-Instruct | 43.12 | 5 | 57.96 | 13 | 82.96 | 19 |
| Grok-3-Mini (High) | 42.82 | 9 | 59.64 | 2 | 85.10 | 16 |
| Llama-3.1-70B-Instruct | 42.86 | 6 | 58.32 | 10 | 74.52 | 23 |
| Llama-3.1-8B-Instruct | 42.84 | 8 | 55.90 | 24 | 64.20 | 25 |
| Llama-3.3-70B-Instruct | 42.80 | 11 | 58.96 | 5 | 96.00 | 3 |
| Qwen2.5-72B-Instruct | 43.28 | 3 | 58.02 | 11 | 86.54 | 14 |
| Qwen2.5-Coder-32B-Instruct | 42.86 | 6 | 58.02 | 11 | 86.86 | 13 |
| Qwen3-235B-A22B | 40.88 | 23 | 56.84 | 21 | 95.12 | 4 |
| Qwen3-30B-A3B | 41.10 | 22 | 56.46 | 22 | 81.64 | 20 |
| Qwen3-32B | 41.40 | 20 | 57.74 | 17 | 90.10 | 9 |
| o3-mini (Med) | 40.60 | 24 | 56.24 | 23 | 84.26 | 18 |
| o4-mini (Med) | 41.14 | 21 | 57.70 | 18 | 84.38 | 17 |

## J  HUMAN STUDY

We conducted a human evaluation to investigate the reliability of LLM-as-judge used in our paper. We conduct this by randomly sampling 60 samples from the code summarization task, using predictions from the top 3 performing models on 20 same samples. Three developers with at least 5 years of experience participated, using the same criteria as the LLM judges. The results are shown in Table 20. We calculated the Pearson correlation coefficient between LLM judge scores and average human scores, resulting in a correlation of 0.99 with p-value 0.016 ($<0.05$) which indicates a high degree of consistency.

## K  ANALYSIS OF DIFFERENT METRICS

To comprehensively investigate the performance, we add additional smooth metrics to complement Pass@1 and LLM scores. Specifically, for code generation task, we follow previous work (Jiang et al., 2024) and add CodeBLEU to measure partial correctness and structural similarity; for code summarization task, we follow previous work and use BLEU (Sun et al., 2025) to evaluate the quality of generated summaries. The results are shown in Table 21 and 22. From these tables, we can observe that the model's performance under these metrics is relatively low. This is because even if the generated code or comments exhibit significant textual differences from the ground truth, they may still be consistent in terms of functionality and semantics.

Table 19: Influence of different judge models on Code Review

| Model | Gemini-2.5-Flash | Rank | Claude-4 | Rank | GPT-4o | Rank |
|---|---|---|---|---|---|---|
| Claude-3.5-Haiku-20241022 | 32.20 | 16 | 39.58 | 17 | 30.56 | 12 |
| Claude-3.5-Sonnet-20241022 | 34.16 | 2 | 39.54 | 18 | 29.98 | 20 |
| Claude-3.7-Sonnet | 33.46 | 3 | 40.38 | 8 | 30.48 | 14 |
| Claude-Sonnet-4 | 32.84 | 6 | 41.04 | 3 | 31.04 | 8 |
| DeepSeek-R1 | 32.26 | 15 | 40.48 | 7 | 27.38 | 25 |
| DeepSeek-R1 (0528) | 32.38 | 11 | 41.12 | 2 | 31.30 | 5 |
| DeepSeek-V3 | 32.56 | 9 | 39.60 | 16 | 30.52 | 13 |
| GPT-3.5-turbo-0125 | 29.60 | 26 | 34.72 | 26 | 29.64 | 22 |
| GPT-4-turbo-2024-04-09 | 32.30 | 13 | 39.36 | 22 | 29.70 | 21 |
| GPT-4.1-2025-04-14 | 32.92 | 5 | 40.80 | 4 | 29.38 | 23 |
| GPT-4o-2024-11-20 | 31.46 | 21 | 39.42 | 20 | 30.46 | 15 |
| GPT-5 | 29.98 | 25 | 42.78 | 1 | 26.62 | 26 |
| Gemini-2.5-Pro-05-06 | 32.68 | 8 | 40.30 | 11 | 31.46 | 3 |
| Gemma-3-27B-Instruct | 32.72 | 7 | 40.62 | 6 | 31.74 | 1 |
| Grok-3-Mini (High) | 34.54 | 1 | 40.34 | 10 | 30.90 | 10 |
| Llama-3.1-70B-Instruct | 31.32 | 22 | 38.74 | 24 | 30.14 | 19 |
| Llama-3.1-8B-Instruct | 30.42 | 24 | 37.36 | 25 | 30.20 | 18 |
| Llama-3.3-70B-Instruct | 32.34 | 12 | 38.80 | 23 | 30.70 | 11 |
| Llama-4-Scout-17B-16E-Instruct | 32.30 | 13 | 39.66 | 15 | 30.36 | 16 |
| Qwen2.5-72B-Instruct | 32.50 | 10 | 39.40 | 21 | 31.42 | 4 |
| Qwen2.5-Coder-32B-Instruct | 31.80 | 20 | 39.52 | 19 | 31.22 | 6 |
| Qwen3-235B-A22B | 32.14 | 17 | 40.38 | 8 | 31.10 | 7 |
| Qwen3-30B-A3B | 31.84 | 19 | 40.30 | 11 | 31.70 | 2 |
| Qwen3-32B | 32.10 | 18 | 40.28 | 13 | 30.36 | 16 |
| o3-mini (Med) | 33.10 | 4 | 39.68 | 14 | 31.00 | 9 |
| o4-mini (Med) | 30.60 | 23 | 40.76 | 5 | 29.02 | 24 |

| Model | LLM Judge Score | Human 1 Score | Human 2 Score | Human 3 Score |
|---|---|---|---|---|
| GPT-5 | 65 | 63 | 65 | 63 |
| Claude-3.5-Sonnet | 28 | 27 | 26 | 29 |
| LLaMA-3.3-70B | 38 | 39 | 38 | 37 |

Table 20: Model evaluation scores

# L    DATA CONTAMINATION ANALYSIS

Data contamination is a widespread challenge in benchmarking internet-scale LLMs, especially since most models do not disclose their training data for auditing. To mitigate this problem, for most tasks, we make the data collection and process stage an automated pipeline and plan to make the benchmark live, which could avoid the data contamination problem. Besides, to validate the contamination probability of current data, we adopted the widely-used min-k (Shi et al., 2023) method to detect potential contamination. We used Qwen3-30B-A3B for detection given its recent release (May 2025) and open-source nature, which enables access to the model's output probabilities. The results are shown in Table 23. The results indicate that even newer models did not exhibit data leakage on our dataset. We have added this analysis to the appendix to ensure research transparency.

# M    LARGE LANGUAGE MODELS USAGE STATEMENT

In this paper, we employed LLMs to support the polish and refinement of this manuscript. The LLM was utilized to enhance linguistic expression and boost text comprehensibility. The model's assistance encompassed activities including sentence restructuring and grammatical checks.

We emphasize that the LLMs are not used for conceptual development or experimental framework design. The authors assume complete accountability for all manuscript content, including portions

Table 21: Model Performance on Java and Python Tasks (CodeBLEU)

| Rank | Model | Java | Python | Average |
|---|---|---|---|---|
| 1 | Claude-3.7-Sonnet | 35.38 | 27.91 | 31.65 |
| 2 | GPT-4.1-2025-04-14 | 35.35 | 26.96 | 31.16 |
| 3 | Claude-Sonnet-4 | 34.76 | 26.92 | 30.84 |
| 4 | Qwen2.5-72B-Instruct | 34.17 | 27.01 | 30.59 |
| 5 | Claude-3.5-Sonnet-20241022 | 34.16 | 26.90 | 30.53 |
| 6 | Qwen2.5-Coder-32B-Instruct | 34.05 | 26.63 | 30.34 |
| 7 | o3-mini (Med) | 34.09 | 26.55 | 30.32 |
| 8 | GPT-5 | 34.03 | 26.15 | 30.09 |
| 9 | GPT-4o-2024-11-20 | 33.15 | 26.95 | 30.04 |
| 10 | DeepSeek-R1 (0528) | 34.42 | 25.59 | 30.01 |
| 11 | Qwen3-235B-A22B | 33.94 | 25.22 | 29.58 |
| 12 | DeepSeek-V3 | 30.98 | 27.72 | 29.35 |
| 13 | Qwen3-32B | 33.83 | 24.85 | 29.33 |
| 14 | DeepSeek-R1 | 34.03 | 24.66 | 29.33 |
| 15 | Qwen3-30B-A3B | 33.54 | 24.56 | 29.06 |
| 16 | GPT-4-turbo-2024-04-09 | 31.50 | 26.54 | 29.02 |
| 17 | Grok-3-Mini (High) | 32.52 | 25.51 | 29.01 |
| 18 | o4-mini (Med) | 31.92 | 25.84 | 28.88 |
| 19 | Llama-3.3-70B-Instruct | 32.15 | 25.31 | 28.73 |
| 20 | GPT-3.5-turbo-0125 | 31.90 | 25.49 | 28.69 |
| 21 | Claude-3.5-Haiku-20241022 | 30.96 | 26.03 | 28.49 |
| 22 | Llama-4-Scout-17B-16E-Instruct | 31.29 | 25.38 | 28.34 |
| 23 | Llama-3.1-70B-Instruct | 31.71 | 24.76 | 28.24 |
| 24 | Gemini-2.5-Pro-05-06 | 29.80 | 25.76 | 27.78 |
| 25 | Gemma-3-27B-Instruct | 28.93 | 25.65 | 27.29 |
| 26 | Llama-3.1-8B-Instruct | 30.33 | 23.29 | 26.81 |

Table 22: Code Summarization Task Results (BLEU)

| Rank | Model | BLEU Score |
|---|---|---|
| 1 | Llama-4-Scout-17B-16E-Instruct | 3.59 |
| 2 | Llama-3.1-8B-Instruct | 3.35 |
| 3 | GPT-4.1-2025-04-14 | 3.19 |
| 4 | Llama-3.1-70B-Instruct | 3.03 |
| 5 | Gemini-2.5-Pro-Preview-05-06 | 3.02 |
| 6 | GPT-3.5-turbo-0125 | 2.88 |
| 7 | Qwen2.5-72B-Instruct | 2.77 |
| 8 | GPT-4o-2024-11-20 | 2.76 |
| 9 | o4-mini(Med) | 2.72 |
| 10 | Gemma-3-27B-Instruct | 2.62 |
| 10 | o3-mini(Med) | 2.62 |
| 10 | Qwen2.5-Coder-32B-Instruct | 2.62 |
| 13 | Claude-3.7-Sonnet | 2.57 |
| 14 | GPT-5 | 2.56 |
| 14 | DeepSeek-R1(0528) | 2.56 |
| 14 | DeepSeek-R1 | 2.56 |
| 17 | Qwen3-30B-A3B | 2.55 |
| 18 | Grok-3-Mini(High) | 2.52 |
| 18 | Claude-Sonnet-4 | 2.52 |
| 20 | Claude-3.5-Sonnet-20241022 | 2.47 |
| 21 | Qwen3-32B | 2.37 |
| 22 | Llama-3.3-70B-Instruct | 2.33 |
| 23 | Claude-3.5-Haiku-20241022 | 2.31 |
| 24 | DeepSeek-V3 | 2.23 |
| 25 | GPT-4-turbo-2024-04-09 | 2.22 |
| 26 | Qwen3-235B-A22B | 1.62 |

Table 23: Contamination detection results

| Task | Code Generation | Code Summuration | Code Translation | Code Review | Code Reasoning | Vulnerability Detection | Test Generation |
|------|-----------------|------------------|------------------|-------------|----------------|-------------------------|-----------------|
| Min-k | -7.72 | -7.54 | -4.68 | -7.38 | -6.62 | -6.60 | -6.17 |

that were refined through LLM assistance. We have verified that all LLM-produced content complies with academic integrity standards and does not constitute plagiarism or scholarly misconduct.

