# OpenReview forum: "TREAT: A Code LLMs Trustworthiness / Reliability Evaluation and Testing Framework"
_ICLR.cc/2026/Conference — Submitted to ICLR 2026_

### Official Review · Reviewer_rx75 · 2025-10-25

**Soundness:** 3
**Presentation:** 3
**Contribution:** 2
**Rating:** 4
**Confidence:** 4

**Summary:**

## Paper Summary

This paper proposes TREAT, a benchmark and evaluation framework designed to assess trustworthiness and reliability of Code LLMs in realistic software-engineering scenarios. It covers:
1. 10+ tasks spanning code generation, summarization, reasoning, review, test generation, vulnerability detection, etc.
2. Multi-language support (up to 12 languages)
3. Multi-modality evaluation including UI-to-Code tasks
4. Robustness evaluation using code transformations and misleading cues
5. Multi-prompt evaluation to mitigate prompt bias
The authors evaluate 26+ state-of-the-art models, uncovering notable weaknesses in robustness, multi-modal UI tasks, and cross-task consistency.

**Strengths:**

## Strengths

1. Comprehensive coverage of SE tasks. Holistic lifecycle-oriented evaluation instead of narrow codegen-only benchmarks.

2. Incorporates robustness testing. Perturbations reveal vulnerability to misleading comments and code structure changes.

3. Large-scale model coverage. Commercial + open-source, reasoning vs non-reasoning LLMs.

**Weaknesses:**

## Weaknesses

1. Lack of novelty in dataset construction. Many task datasets are combinations or resampling from existing public benchmarks such as PRIMEVUL, EvalPlus, DESIGNBENCH, GEEKSFORGEEKS/HackerRank, GitHub mined code, etc. There is no clearly new task definition, only integration. The framework is large and ambitious but conceptual novelty is thin: It aggregates, expands, and re-labels existing evaluation components instead of introducing fundamentally new methodology.


2. Imcomplete and misleading claim in Table 1. First Table 1 miss some releated work, including some dataset used in this paper. Second, Multi Prompt techniques is used in DyCodeEval [1], Multi-Modality is used in SWE-Bench-Multimodal [2], and Robustness Evaluation seems to be a plugin appraoch can be applied to any seed dataset [3, 4], which does not belong to the contribution of this work.

3. Many details are pushed into Appendix. Insufficient explanation for: (1) sample selection (2) deduplication, (3) data quantity per task/language
and (4) contamination filtering and verification.


4. Data contamination risk is acknowledged but not enforced or measured. Given that many evaluated models have been trained on GitHub/HackerRank, results may overestimate model capability. Specifically, some evaluated models in Table 2 were released after certain dataset releases; considering the lack of transparency of the pretraining data of the evaluated models and the potential inclusion of benchmark data in model training, reported scores may be inflated.-level. Even authors admit lacking repository-level realism (dependencies, build systems). Software engineering isn’t only function puzzles. I would suggest the auhtor also consider some repository task, such as SWEBench.


5. Missing related work. The paper would benefit from acknowledging and comparing against recent benchmarks that also focus on code LLMs.




[1]. Dynamic Benchmarking of Reasoning Capabilities in Code Large Language Models Under Data Contamination (ICML 2025)

[2]. SWE-bench Multimodal: Do AI Systems Generalize to Visual Software Domains? (ICLR 2025)

[3]. DynaCode: A Dynamic Complexity-Aware Code Benchmark for Evaluating Large Language Models in Code Generation (ACL 2025 Finding)

[4] Is Your Benchmark (Still) Useful? Dynamic Benchmarking for Code Language Models

[6]. Codereval: A benchmark of pragmatic code generation with generative pre-trained models (ICSE 2024)

[7]. EvoCodeBench: An Evolving Code Generation Benchmark Aligned with Real-World Code Repositories (Neurips 2025)

[8]. Evaluating and Improving LLM-based Competitive Program Generation

[9]. PPM: Automated Generation of Diverse Programming Problems for Benchmarking Code Generation Models (FSE 2024)

**Questions:**

1. How do you justify TREAT as a “novel benchmark” given that most tasks are resampled or aggregated from existing datasets?

2. Can you provide more details on sample selection, deduplication, and contamination filtering? How consistent is data quantity across tasks and languages, and how might this affect cross-task comparisons?

3. Have you measured or verified contamination in the datasets with respect to the pretraining data of the evaluated models?

4. Given that evaluation is function-level, how do you ensure that TREAT reflects real-world software engineering challenges like repository-level dependencies and build systems?

---

> ### Author Response · Authors · 2025-11-22
> **Response to Reviewer rx75**
>
> Thank you for your hard work reviewing! Your critical suggestions have helped us strengthen the paper. For your highlighted concerns, we will address them one by one:
>
> ---
> >[W1/Q1] Lack of novelty in dataset construction. Many task datasets are combinations or resampling from existing public benchmarks. How do you justify TREAT as a "novel benchmark"?
> ---
>
> Thanks for your question, we want to first clarify that TREAT is not merely an aggregation of existing datasets. Instead, we collected new data for most tasks, and only reused a small portion of existing datasets for tasks where automatic annotation is challenging. Specifically:
> - We **did not use EvalPlus** data mentioned in your comment.
> - For GeeksforGeeks/HackerRank, they were not existing datasets. We did not reuse existing datasets. Instead, we **crawled new data** from these platforms.
> - In total, **81.6%** of TREAT’s data is newly collected or created by us, while only **18.4%** is aggregated from existing datasets (400 samples from PRIMEVUL, 200 from CODAMOSA, 900 from DESIGNBENCH, 328 from PolyHumanEval).
> Besides, we believe that reusing a subset of existing data to ensure comprehensive evaluation is a **common and widely accepted practice in existing works** [1,2,3]. Similar to those general-domain LLM benchmarks, TREAT is the **first work** to conduct a comprehensive evaluation of code LLMs across the entire software development lifecycle. This holistic evaluation reveals novel findings such as the substantial performance variation across tasks. We believe this cross-task, lifecycle-oriented benchmark fills a critical gap in code LLM evaluation and contributes valuable insights to the community.
>
>
>
> ---
> >[W2] Incomplete and misleading claim in Table 1. Missing related work and misattributing contributions.
> ---
>
> We thank you for pointing out this issue. We selected some representative benchmarks for comparison before and have updated Table 1 to include all relevant work mentioned in your comment (e.g., DyCodeEval, SWE-Bench-Multimodal). Additionally, we clarify the differences between our multi-prompt strategy and DyCodeEval’s:
> - DyCodeEval modifies the problem statement itself to mitigate data contamination.
> - Our multi-prompt strategy modifies non-task-related parts of the prompt to avoid model bias toward specific prompt phrasings.
> Regarding robustness evaluation, we acknowledge that it can be applied as a plugin to existing datasets. So we followed your suggestion and removed the "Robustness Evaluation" column from Table 1 to avoid misattribution. However, we believe that integrating robustness assessment into the evaluation is a contribution of TREAT as it reveals the limitation of current code LLMs.

---

> > ### Author Response · Authors · 2025-11-22
> > **Response to Reviewer rx75 (2)**
> >
> > ---
> > >[W3/Q2] Many details are pushed into Appendix. Insufficient explanation for sample selection, deduplication, data quantity, and contamination filtering.
> > ---
> >
> > Thanks for your comments. We agree that these details are critical for transparency, and we have updated the main text (Section 3.1). Due to page limitations and the extensive technical details involved, we only present the high-level principles of our data processing pipeline. The specific details of each task are detailed in the appendix. The key details are as follows:
> > - **Sample Selection**: For newly collected data: We prioritize quality and diversity by sourcing from high-quality platforms (GitHub repositories with ≥100 stars and active mainstream programming platforms) and cover different languages. For aggregated data: We select representative benchmarks after 2024 in each task domain to ensure alignment with current model capabilities.
> > - **Data Filtering and Deduplication**: We first remove exact duplicates via string matching, then follow existing work [4,5] and apply task-specific cleaning pipelines to ensure data integrity. For aggregated benchmarks, we validate that the selected benchmarks have undergone rigorous filtering and deduplication in their original publications.
> > - **Data Quantity**: The following table shows the data volume per task, which we have added to the appendix:
> >
> >
> > | Task                  | Code Generation | Code Summarization | Code Translation | Code Reasoning | Code Review | Test Generation | Vulnerability Detection | Multi-modality |
> > |-----------------------|----------------|-------------------|-----------------|---------------|-------------|----------------|----------------------|-------------|
> > | Data Volume           | 1664            | 2000              | 744             | 2000          | 2000        | 200            | 400                  | 900         |
> >
> >
> >
> > |   Language    | C    | C++       | C#   | Java              | Go   | JavaScript | TypeScript | PHP  | Python                 | Ruby | Html | Css  |
> > |--------------|------|-----------|------|-------------------|------|------------|------------|------|------------------------|------|------|------|
> > |   Data Volume   | 400  | 800       | 400  | 2604              | 400  | 1072       | 400        | 400  | 2804                   | 400  | 228  | 228  |
> >
> >
> > While data volume varies across tasks, we ensure the fairness of cross-task comparison by first calculating the average performance per task, then computing final model rankings based on average task rankings. This approach mitigates the impact of uneven data volume on cross-task comparisons.
> > - **Contamination Filtering**: We have added contamination detection with details in our response to [W4/Q3].
> >
> > ---
> > >[W4/Q3] Data contamination risk is acknowledged but not enforced or measured. Have you measured or verified contamination with respect to the pretraining data of evaluated models?
> > ---
> >
> >
> > Thanks for your comment. Data contamination is a widespread challenge in benchmarking internet-scale LLMs, especially since most models do not disclose their training data for auditing. To mitigate this problem, for most tasks, we make the data collection and process stage an automated pipeline and plan to make the benchmark live, which could avoid the data contamination problem. Besides, to validate the contamination probability of current data, we adopted the widely-used min-k [6] method to detect potential contamination. We used Qwen3-30B-A3B for detection given its recent release (May 2025) and open-source nature, which enables access to the model's output probabilities.  The results are shown in the following table:
> >
> > | Task | Code Generation   | Code Summuration   | Code Translation  | Code Review | Code Reasoning | Vulnerability Detection | Test  Generation |
> > |-------|-------|-------|--------|--------|-----------|------------|--------|
> > |Min-k| -7.72 | -7.54 | -4.68  | -7.38  | -6.62     | -6.60      | -6.17  |
> >
> > We can find that all the results indicate that even newer models did not exhibit data leakage on our dataset. We have added this analysis to the appendix to ensure research transparency.

---

> > > ### Author Response · Authors · 2025-11-22
> > > **Response to Reviewer rx75 (3)**
> > >
> > > ---
> > > >[W5/Q4] Given that evaluation is function-level, how do you ensure that TREAT reflects real-world software engineering challenges like repository-level dependencies and build systems?
> > > ---
> > >
> > >
> > > We thank you for this insightful suggestion. We also agree that repo-level evaluation and agent evaluation are important. In fact, some datasets in TREAT (e.g., test generation) already contain data beyond function-level, but we adopted a unified context extraction method for evaluation instead of the agent-based evaluation approach used in benchmarks like SWE-Bench. This is because we believe evaluating models’ ability to directly answer given questions remains crucial and agent-based evaluation cannot replace these question-answering evaluations. On one hand, most existing LLM services operate in a direct question-answering mode, thus direct evaluation is essential for model development. On the other hand, agent-based evaluation has application limitations, as it is not suitable for assessing models in the early stages of training. Model training typically follows a multi-stage process [1,2,3], where agent capabilities are usually introduced in the mid-to-late training phases (e.g., the fourth stage in Qwen’s training). Thus, when multiple checkpoints are generated in the initial stages and the best one needs to be selected for the next training phase, only such question-answering evaluations can be used. We have discussed this limitation in the future work section and plan to incorporate repo-level agent evaluations for all tasks in subsequent iterations.
> > >
> > > ---
> > > >[W6] Missing related work. The paper would benefit from acknowledging and comparing against recent benchmarks that also focus on code LLMs.
> > > ---
> > >
> > > Following your suggestion, we have updated the related work section (Appendix D.3: LARGE LANGUAGE MODELS EVALUATION) to include detailed comparisons with recent code LLM benchmarks (e.g., Codereval, EvoCodeBench, PPM). The following is newly added part and is highlighted with blue:
> > >
> > > DYCODEEVAL (Chen et al.) introduces dynamic benchmarking that deliberately controls contamination to assess reasoning capabilities in code LLMs. DYNACODE (Hu et al., 2025) proposes a dynamic complexity-aware framework that automatically adjusts problem difficulty, enabling finer-grained and adaptive evaluation of code generation skills. EVOCODEBENCH (Li et al., 2024a) is an evolving benchmark tightly aligned with real-world GitHub repositories. It continuously incorporates new commits and domain-specific tasks to prevent leakage and maintain relevance. MMCode (Li et al., 2024c) and SWE-BENCH MULTIMODAL (Yang et al., 2024c) extends the original SWE-bench by adding visual inputs such as screenshots, UI mockups, design files, showing that current multimodal code models suffer large performance drops without visual context and highlighting a generalization gap to visual software domains. CODEREVAL (Yu et al., 2024) and DevEval (Li et al., 2024b) emphasize repository-level code generation drawn from real open-source repositories. PPM (Chen et al., 2024d) presents an automated pipeline that uses LLMs themselves to synthesise diverse programming problems, facilitating scalable and varied benchmark creation. Recent efforts have also evaluated and proposed improvements for LLM-based competitive programming generation using real 2024 ICPC/CCPC contest problems (Wei et al., 2025).
> > >
> > >
> > >
> > > [1] GLUE: A multi-task benchmark and analysis platform for natural language understanding
> > > [2] A Multitask, Multilingual, Multimodal Evaluation of ChatGPT on Reasoning, Hallucination, and Interactivity
> > > [3] Beyond the Imitation Game: Quantifying and extrapolating the capabilities of language models
> > > [4] Are we building on the rock? on the importance of data preprocessing for code summarization.
> > > [5] Automating code review activities by large-scale pre-training
> > > [6] Detecting Pretraining Data from Large Language Models

---

> > > > ### Comment · Reviewer_rx75 · 2025-11-24
> > > > **Response to authors rebuttal**
> > > >
> > > > Thanks for the authors’ feedback.
> > > >
> > > > 1. Regarding the novelty of the dataset construction, I appreciate the authors’ clarification of my earlier misunderstanding. However, I still feel that the claim *“TREAT is the first work to conduct a comprehensive evaluation of code LLMs across the entire software development lifecycle”* is somewhat overstated. First, I do not believe that function-level benchmarks can represent the *entire* software development lifecycle. Second, given that no new tasks are proposed in this benchmark, it seems possible to reuse multiple existing benchmarks to achieve similar evaluation goals, although the final benchmark results may differ.
> > > >
> > > > 2. The authors’ response addressed my concern.
> > > >
> > > > 3. I hope the authors can revise their manuscript to include these detailed descriptions, **as they are essential for readers to fully understand the benchmark construction process and its intended scope**.
> > > >
> > > > 4. The authors’ response regarding data contamination does not fully convince me. First, I am uncertain about the reliability of the cited paper in detecting data contamination. Second, I do not understand what the numbers in the table represent and why these numbers indicate that the model is not contaminated. Third, I am not sure that using only **Qwen3** as an example is sufficient **to generalize the conclusion to all evaluated models**.
> > > >
> > > > 5. The authors’ response did not fully address my concern regarding whether function-level benchmarks can faithfully represent real-world software development.
> > > >
> > > > 6. The authors added the missing related work and thus addressed my concern.

---

### Official Review · Reviewer_4vGk · 2025-10-31

**Soundness:** 3
**Presentation:** 3
**Contribution:** 3
**Rating:** 6
**Confidence:** 4

**Summary:**

This paper introduce a benchmark for evaluating the trustworthiness of large foundation models in real-world software engineering scenarios. It provides a very comprehensive setting to span the software development lifecycle with more than 10 takss and languages. Also, the holistic evaluation framework try to use different prompts to reduce the bias in evaluation.

**Strengths:**

1. this submission provides lots of detailed experiment results and analysis which comprehensively support the insights behind current large foundation models.
2. the evaluation across multiple programming language and even including UI code generation tasks is very interesting and bridges the visuals and code.
3. multi-prompt strategy and adaptive answer extraction helps to improve the fairness of this submission and real-world alighment.
4. the UI code editing or reparing tasks aligns with real deceloper workflows.
5. robustness testing with strucural, misleading comments, and misleading hints.

**Weaknesses:**

1. Adding more types of level can further enhance the benchmark (e.g., repo-level)

2. Model-as-judge may introduce some bias in evaluation. it's better to add more comparison between gpt-4o and human in some subset of testing and analyze the difference.

3. Pass@1 is the primary metric for some tasks. It is better to consider more like partial correctness or runtime issues.

4. In robustness evaluation, instead of rule-based perturbations, using some real-world git-diffs may aligh more with the real scenerios.

5. Authors can try to analyze the influence of image resolution across different models in benchmarking.

**Questions:**

Please refer to the Weaknesses section.

---

> ### Author Response · Authors · 2025-11-22
> **Response to Reviewer 4vGk**
>
> Thank you for your insightful review! We appreciate that you provided us with such thoughtful feedback. For your highlighted concerns, we will address them one by one:
>
> ---
> >[W1] Adding more types of level can further enhance the benchmark.
> ---
> We thank you for this insightful suggestion. We also agree that repo-level evaluation and agent evaluation are important. In fact, some datasets in TREAT (e.g., test generation) already contain data beyond function-level, but we adopted a unified context extraction method for evaluation instead of the agent-based evaluation approach used in benchmarks like SWE-Bench. This is because we believe evaluating models’ ability to directly answer given questions remains crucial and agent-based evaluation cannot replace these question-answering evaluations. On one hand, most existing LLM services operate in a direct question-answering mode. On the other hand, agent-based evaluation has application limitations, as it is not suitable for assessing models in the early stages of training. Model training typically follows a multi-stage process [1,2,3], where agent capabilities are usually introduced in the mid-to-late training phases (e.g., the fourth stage in Qwen’s training). Thus, when multiple checkpoints are generated in the initial stages and the best one needs to be selected for the next training phase, only such question-answering evaluations can be used. We have discussed this limitation in the future work section and plan to incorporate repo-level agent evaluations for all tasks in subsequent iterations.
>
>
> ---
> >[W2] Model-as-judge may introduce some bias in evaluation. It's better to add more comparison between GPT-4o and human in some subset of testing and analyze the difference.
> ---
>
>
> Following your suggestion, we conducted human evaluation to verify the reliability of LLM-as-judge. Due to time constraints in the rebuttal phase, we conduct a humane evaluation by randomly sampling 60 samples from the code summarization task, using predictions from the top 3 performing models on 20 same samples. Three developers with at least 5 years of experience participated, using the same criteria as the LLM judges. The results are shown below:
>
> | Model               | LLM Judge Score | Human 1 Score | Human 2 Score | Human 3 Score |
> | ------------------- | ----------------|---------------|---------------|---------------|
> | GPT-5               | 65              | 63            | 65            | 63            |
> | Claude-3.5-Sonnet   | 28              | 27            | 26            | 29            |
> | LLaMA-3.3-70B       | 38              | 39            | 38            | 37            |
>
> We calculated the Pearson correlation coefficient between LLM judge scores and average human scores, resulting in a correlation of 0.99 with p-value 0.016 (<0.05) which indicates a high degree of consistency.
>
>
> ---
> >[W3] Pass@1 is the primary metric for some tasks. It is better to consider more like partial correctness or runtime issues.
> ---
>
>
> We agree with your suggestion to adopt more metrics. We have added additional smooth metrics to complement Pass@1 and LLM scores:
> 1. For code generation task: We follow previous work [5,6] and add CodeBleu to measure partial correctness and structural similarity.
> 2. For code summarization task: We follow previous work and use BLEU [7] to evaluate the quality of generated summaries.
>
>
> The results for the code generation task (CodeBLEU score) are shown below:
>
> | Rank | Model | Java | Python | Average |
> | --- | --- | --- | --- | --- |
> | 1 | Claude-3.7-Sonnet | 35.38 | 27.91 | 31.65 |
> | 2 | GPT-4.1-2025-04-14 | 35.35 | 26.96 | 31.16 |
> | 3 | Claude-Sonnet-4 | 34.76 | 26.92 | 30.84 |
> | 4 | Qwen2.5-72B-Instruct | 34.17 | 27.01 | 30.59 |
> | 5 | Claude-3.5-Sonnet-20241022 | 34.16 | 26.90 | 30.53 |
> | 6 | Qwen2.5-Coder-32B-Instruct | 34.05 | 26.63 | 30.34 |
> | 7 | o3-mini (Med) | 34.09 | 26.55 | 30.32 |
> | 8 | GPT-5 | 34.03 | 26.15 | 30.09 |
> | 9 | GPT-4o-2024-11-20 | 33.15 | 26.95 | 30.04 |
> | 10 | DeepSeek-R1 (0528) | 34.42 | 25.59 | 30.01 |
> | 11 | Qwen3-235B-A22B | 33.94 | 25.22 | 29.58 |
> | 12 | DeepSeek-V3 | 30.98 | 27.72 | 29.35 |
> | 13 | Qwen3-32B | 33.83 | 24.85 | 29.33 |
> | 14 | DeepSeek-R1 | 34.03 | 24.66 | 29.33 |
> | 15 | Qwen3-30B-A3B | 33.54 | 24.56 | 29.06 |
> | 16 | GPT-4-turbo-2024-04-09 | 31.50 | 26.54 | 29.02 |
> | 17 | Grok-3-Mini (High) | 32.52 | 25.51 | 29.01 |
> | 18 | o4-mini (Med) | 31.92 | 25.84 | 28.88 |
> | 19 | Llama-3.3-70B-Instruct | 32.15 | 25.31 | 28.73 |
> | 20 | GPT-3.5-turbo-0125 | 31.90 | 25.49 | 28.69 |
> | 21 | Claude-3.5-Haiku-20241022 | 30.96 | 26.03 | 28.49 |
> | 22 | Llama-4-Scout-17B-16E-Instruct | 31.29 | 25.38 | 28.34 |
> | 23 | Llama-3.1-70B-Instruct | 31.71 | 24.76 | 28.24 |
> | 24 | Gemini-2.5-Pro-05-06 | 29.80 | 25.76 | 27.78 |
> | 25 | Gemma-3-27B-Instruct | 28.93 | 25.65 | 27.29 |
> | 26 | Llama-3.1-8B-Instruct | 30.33 | 23.29 | 26.81 |

---

> > ### Author Response · Authors · 2025-11-22
> > **Response to Reviewer 4vGk (2)**
> >
> > The results for the code summarization task (BLEU score) are shown below:
> >
> > | Rank | Model                                   | BLEU Score |
> > |------|-----------------------------------------|----------------|
> > | 1    | Llama-4-Scout-17B-16E-Instruct          | 3.59           |
> > | 2    | Llama-3.1-8B-Instruct                   | 3.35           |
> > | 3    | GPT-4.1-2025-04-14                      | 3.19           |
> > | 4    | Llama-3.1-70B-Instruct                  | 3.03           |
> > | 5    | Gemini-2.5-Pro-Preview-05-06            | 3.02           |
> > | 6    | GPT-3.5-turbo-0125                      | 2.88           |
> > | 7    | Qwen2.5-72B-Instruct                    | 2.77           |
> > | 8    | GPT-4o-2024-11-20                       | 2.76           |
> > | 9    | o4-mini(Med)                            | 2.72           |
> > | 10   | Gemma-3-27B-Instruct                    | 2.62           |
> > | 10   | o3-mini(Med)                            | 2.62           |
> > | 10   | Qwen2.5-Coder-32B-Instruct              | 2.62           |
> > | 13   | Claude-3.7-Sonnet                       | 2.57           |
> > | 14   | GPT-5                                   | 2.56           |
> > | 14   | DeepSeek-R1(0528)                       | 2.56           |
> > | 14   | DeepSeek-R1                              | 2.56           |
> > | 17   | Qwen3-30B-A3B                           | 2.55           |
> > | 18   | Grok-3-Mini(High)                       | 2.52           |
> > | 18   | Claude-Sonnet-4                         | 2.52           |
> > | 20   | Claude-3.5-Sonnet-20241022              | 2.47           |
> > | 21   | Qwen3-32B                               | 2.37           |
> > | 22   | Llama-3.3-70B-Instruct                  | 2.33           |
> > | 23   | Claude-3.5-Haiku-20241022               | 2.31           |
> > | 24   | DeepSeek-V3                              | 2.23           |
> > | 25   | GPT-4-turbo-2024-04-09                  | 2.22           |
> > | 26   | Qwen3-235B-A22B                         | 1.62           |
> >
> > From this table, we can observe that the model's performance under these metrics is relatively low. This is because even if the generated code or comments exhibit significant textual differences from the ground truth, they may still be consistent in terms of functionality and semantics. We have included the complete set of supplementary metrics along with detailed analyses in the appendix.
> >
> >
> > ---
> > >[W4] In robustness evaluation, instead of rule-based perturbations, using some real-world git-diffs may align more with the real scenarios.
> > ---
> >
> >
> > We appreciate your suggestion and would like to clarify the rationale behind our choice of rule-based perturbations. We adopted the perturbation method from CodeCrash as it is a recent and comprehensive work on code robustness and systematically simulates perturbations to code structure, comments, and reasoning components. As a benchmarking work, we prioritized controlled experiments to fairly compare the impact of different types of perturbations. Rule-based synthesis allows precise control over perturbation variables, which is challenging with git-diffs (as they often involve uncontrolled changes).
> > We also agree that real-world git-diffs are valuable for evaluating model performance under in-the-wild code refactoring. We plan to study this in future work.
> >
> > ---
> > >[W5] Authors can try to analyze the influence of image resolution across different models in benchmarking.
> > ---
> >
> > Thanks for your suggestion. As an evaluation work, we follow the settings of existing work [4] and set a unified resolution to compare model performance. In general, we believe reducing the resolution may indeed lead to a decline in model performance. Since multimodal evaluation is only a sub-module of this paper, we prioritize the model’s performance under clear resolution to assess its assistance for real-world front-end development. We will incorporate an analysis of image resolution in future work.
> >
> > [1] Qwen3 Technical Report
> > [2] Kimi K2: Open Agentic Intelligence
> > [3] LongCat-Flash Technical Report
> > [4] Designbench: A comprehensive benchmark for mllm-based front-end code generation
> > [5] Codegeex: A pre-trained model for code generation with multilingual benchmarking on humaneval-x
> > [6] Self-planning code generation with large language models
> > [7] Source code summarization in the era of large language models

---

### Official Review · Reviewer_JQn4 · 2025-11-01

**Soundness:** 2
**Presentation:** 3
**Contribution:** 1
**Rating:** 2
**Confidence:** 5

**Summary:**

This paper presents TREAT, an evaluation framework for assessing LLMs on code intelligence tasks. The framework evaluates 26 models across 10+ tasks spanning code generation, summarization, translation, reasoning, review, test generation, vulnerability detection, and UI-based tasks. The work incorporates multi-language support (Python and Java primarily but extended 12 languages), robustness evaluation using semantically-preserving perturbations, and multi-prompt evaluation. Key findings include performance variation across tasks, robustness issues, and bottlenecks in multimodal tasks.

**Strengths:**

- The paper evaluates 26 state-of-the-art models across diverse tasks, languages, and modalities, offering insights and providing extensive empirical data that could be useful for the community.

- The use of three prompts per task demonstrates prompt sensitivity, though the magnitude and practical implications remain unclear (see notes below).

- The paper provides detailed appendices with experimental setup, promises code/data release, and supporting reproducibility.

**Weaknesses:**

The paper lacks both novelty and practicality.

First, the paper is fundamentally an ensemble of existing benchmarks and methods with minimal innovation. The benchmark directly samples from or reuses PolyHumanEval, HumanEval+, MBPP, PrimeVul, SymPrompt/CodaMosa, DesignBench, and CodeCrash without significant modification. For instance, code generation uses problems from GeeksforGeeks and HackerRank with EvalPlus-style test augmentation, code translation uses PolyHumanEval and GeeksforGeeks, vulnerability detection directly adopts PrimeVul, multimodal tasks use DesignBench, and robustness evaluation uses CodeCrash. The multi-prompt approach, while useful, is a straightforward extension that simply runs three paraphrased prompts and averages results. No novel evaluation metrics, methodologies, or theoretical frameworks are introduced. The paper reads as a benchmark aggregation exercise rather than advancing evaluation methodology.

Seccond, there is critical data contamination problem rendering the benchmark impractical. The paper's heavy reliance on public platforms and existing benchmarks creates severe contamination issues that fundamentally undermine its utility: GeeksforGeeks and HackerRank are explicitly known to be in training corpora of major LLMs. The HumanEval series of benchmark (e.g., PolyHumanEval and HumanEval+) are also known to be extensively contaminated. Similarly, GitHub repositories are likely in training data given pre-training data scraping practices. These all making performance metrics potentially measure memorization rather than capability. While the paper mentions contamination as a "limitation", it failed to provide viable mitigation strategy, making the benchmark impractical for evaluating current or future models.

Finally, some of the tasks heavily rely on LLM as judge, and there was no proposal on how to mitigate the internal biases of these models, making results unreliable and untrusted especially the scale of performance difference is rather small. For example, the authors acknowledge GPT-4o's bias when GPT-5 scores lowest (26.9%) on code review despite excelling at other tasks, attributing this to preferring "simpler outputs" in Appendix. This admission directly contradicts using it as an objective judge.

**Questions:**

Please see "Weaknesses" for the major ones. In addition,

1. Could the authors add statistical significance when presenting results?

2. What is the actual performance changes across prompts? While ranking is interesting, it'd be good to see the actual performance and variation to understand whether this is a problem that we should worry about at all.

3. There lacks sufficient literature review on robustness in machine learning for code and LLM in general.

---

> ### Author Response · Authors · 2025-11-22
> **Response to Reviewer JQn4**
>
> Thank you for your detailed review and valuable feedback! We appreciate your recognition of our extensive empirical data and commitment to reproducibility. Your comments have helped us clarify critical points about novelty, data contamination, and evaluation reliability. Below, we address each concern one by one.
>
> ---
> >[W1] The paper lacks both novelty and practicality. It is fundamentally an ensemble of existing benchmarks and methods with minimal innovation.
> ---
>
> Thanks for your comment, we want to first clarify that TREAT is **not merely an aggregation of existing datasets**. Instead, we collected new data for most tasks, and only reused a small portion of existing datasets for tasks where automatic annotation is challenging. Specifically:
> - We did **not use HumanEval+ or MBPP** data mentioned in your comment.
> - For GeeksforGeeks/HackerRank, they were not existing datasets. We did not reuse existing datasets. Instead, we **crawled new data** from these platforms.
> - For CodeCrash, we only adopted its perturbation method (not its data).
> - In total, **81.6%** of TREAT’s data is newly collected or created by us, while only **18.4%** is aggregated from existing datasets (400 samples from PRIMEVUL, 200 from CODAMOSA, 900 from DESIGNBENCH, 328 from PolyHumanEval).
> Besides, we believe that reusing a subset of existing data to ensure comprehensive evaluation is a **common and widely accepted practice in existing works** [1,2,3]. Similar to those general-domain LLM benchmarks, TREAT is the **first work** to conduct a comprehensive evaluation of code LLMs across the entire software development lifecycle. This holistic evaluation reveals novel findings such as the substantial performance variation across tasks. We believe this cross-task, lifecycle-oriented benchmark fills a critical gap in code LLM evaluation and contributes valuable insights to the community.
>
> ---
> >[W2] Critical data contamination problem rendering the benchmark impractical.
> ---
>
> Thanks for your comment. Data contamination is a widespread challenge in benchmarking internet-scale LLMs, especially since most models do not disclose their training data for auditing. To mitigate this problem, for most tasks, we make the data collection and process stage an automated pipeline and plan to make the benchmark live, which could avoid the data contamination problem. Besides, to validate the contamination probability of current data, we adopted the widely-used min-k [1] method to detect potential contamination. We used Qwen3-30B-A3B for detection given its recent release (May 2025) and open-source nature, which enables access to the model's output probabilities.  The results are shown in the following table:
>
> | Task | Code Generation   | Code Summuration   | Code Translation  | Code Review | Code Reasoning | Vulnerability Detection | Test  Generation |
> |-------|-------|-------|--------|--------|-----------|------------|--------|
> |Min-k| -7.72 | -7.54 | -4.68  | -7.38  | -6.62     | -6.60      | -6.17  |
>
> We can find that all the results indicate that even recent-released models did not exhibit data leakage on our dataset. We have added this analysis to the appendix to improve research transparency.

---

> > ### Author Response · Authors · 2025-11-22
> > **Response to Reviewer JQn4 (2)**
> >
> > ---
> > >[W3] Some of the tasks heavily rely on LLM as judge, and there was no proposal on how to mitigate the internal biases of these models.
> > ---
> >
> > Thanks for your suggestion, to mitigate this issue, we have added two additional models Gemini-2.5-Flash and Claude-4-Sonnet from different model families as alternative judges for code summarization and code review tasks. The detailed comparison results are shown in the following table:
> >
> > | Model                          | Gemini-2.5-Flash | Rank | Claude-4 | Rank | GPT-4o  | Rank |
> > |--------------------------------|------------------|------|----------|------|---------|------|
> > | Claude-3.5-Haiku-20241022      | 32.20            | 16   | 39.58    | 17   | 30.56   | 12   |
> > | Claude-3.5-Sonnet-20241022     | 34.16            | 2    | 39.54    | 18   | 29.98   | 20   |
> > | Claude-3.7-Sonnet              | 33.46            | 3    | 40.38    | 8    | 30.48   | 14   |
> > | Claude-Sonnet-4                | 32.84            | 6    | 41.04    | 3    | 31.04   | 8    |
> > | DeepSeek-R1                    | 32.26            | 15   | 40.48    | 7    | 27.38   | 25   |
> > | DeepSeek-R1 (0528)             | 32.38            | 11   | 41.12    | 2    | 31.30   | 5    |
> > | DeepSeek-V3                    | 32.56            | 9    | 39.60    | 16   | 30.52   | 13   |
> > | GPT-3.5-turbo-0125             | 29.60            | 26   | 34.72    | 26   | 29.64   | 22   |
> > | GPT-4-turbo-2024-04-09         | 32.30            | 13   | 39.36    | 22   | 29.70   | 21   |
> > | GPT-4.1-2025-04-14             | 32.92            | 5    | 40.80    | 4    | 29.38   | 23   |
> > | GPT-4o-2024-11-20              | 31.46            | 21   | 39.42    | 20   | 30.46   | 15   |
> > | GPT-5                          | 29.98            | 25   | 42.78    | 1    | 26.62   | 26   |
> > | Gemini-2.5-Pro-05-06           | 32.68            | 8    | 40.30    | 11   | 31.46   | 3    |
> > | Gemma-3-27B-Instruct           | 32.72            | 7    | 40.62    | 6    | 31.74   | 1    |
> > | Grok-3-Mini (High)             | 34.54            | 1    | 40.34    | 10   | 30.90   | 10   |
> > | Llama-3.1-70B-Instruct         | 31.32            | 22   | 38.74    | 24   | 30.14   | 19   |
> > | Llama-3.1-8B-Instruct          | 30.42            | 24   | 37.36    | 25   | 30.20   | 18   |
> > | Llama-3.3-70B-Instruct         | 32.34            | 12   | 38.80    | 23   | 30.70   | 11   |
> > | Llama-4-Scout-17B-16E-Instruct | 32.30            | 13   | 39.66    | 15   | 30.36   | 16   |
> > | Qwen2.5-72B-Instruct           | 32.50            | 10   | 39.40    | 21   | 31.42   | 4    |
> > | Qwen2.5-Coder-32B-Instruct     | 31.80            | 20   | 39.52    | 19   | 31.22   | 6    |
> > | Qwen3-235B-A22B                | 32.14            | 17   | 40.38    | 8    | 31.10   | 7    |
> > | Qwen3-30B-A3B                  | 31.84            | 19   | 40.30    | 11   | 31.70   | 2    |
> > | Qwen3-32B                      | 32.10            | 18   | 40.28    | 13   | 30.36   | 16   |
> > | o3-mini (Med)                  | 33.10            | 4    | 39.68    | 14   | 31.00   | 9    |
> > | o4-mini (Med)                  | 30.60            | 23   | 40.76    | 5    | 29.02   | 24   |
> >
> > | Model | Gemini-2.5-Flash | Rank | Claude-4 | Rank | GPT-4o | Rank |
> > |---|---|---|---|---|---|---|
> > | Claude-3.5-Haiku-20241022 | 41.82 | 19 | 57.82 | 16 | 85.24 | 15 |
> > | Claude-3.5-Sonnet-20241022 | 42.74 | 12 | 59.20 | 4 | 96.54 | 2 |
> > | Claude-3.7-Sonnet | 43.44 | 1 | 59.62 | 3 | 88.10 | 11 |
> > | Claude-Sonnet-4 | 43.44 | 1 | 60.38 | 1 | 93.76 | 5 |
> > | DeepSeek-R1 | 42.04 | 17 | 58.72 | 7 | 90.64 | 7 |
> > | DeepSeek-R1 (0528) | 41.98 | 18 | 58.72 | 7 | 90.64 | 7 |
> > | DeepSeek-V3 | 42.30 | 15 | 57.86 | 14 | 92.82 | 6 |
> > | GPT-3.5-turbo-0125 | 42.66 | 13 | 55.00 | 25 | 71.18 | 24 |
> > | GPT-4-turbo-2024-04-09 | 42.28 | 16 | 57.30 | 19 | 89.94 | 10 |
> > | GPT-4.1-2025-04-14 | 42.42 | 14 | 57.30 | 19 | 80.28 | 21 |
> > | GPT-4o-2024-11-20 | 42.82 | 9 | 57.84 | 15 | 87.88 | 12 |
> > | GPT-5 | 40.50 | 25 | 58.34 | 9 | 98.28 | 1 |
> > | Gemini-2.5-Pro-Preview-05-06 | 43.16 | 4 | 58.76 | 6 | 78.88 | 22 |
> > | Gemma-3-27B-Instruct | 43.12 | 5 | 57.96 | 13 | 82.96 | 19 |
> > | Grok-3-Mini (High) | 42.82 | 9 | 59.64 | 2 | 85.10 | 16 |
> > | Llama-3.1-70B-Instruct | 42.86 | 6 | 58.32 | 10 | 74.52 | 23 |
> > | Llama-3.1-8B-Instruct | 42.84 | 8 | 55.90 | 24 | 64.20 | 25 |
> > | Llama-3.3-70B-Instruct | 42.80 | 11 | 58.96 | 5 | 96.00 | 3 |
> > | Qwen2.5-72B-Instruct | 43.28 | 3 | 58.02 | 11 | 86.54 | 14 |
> > | Qwen2.5-Coder-32B-Instruct | 42.86 | 6 | 58.02 | 11 | 86.86 | 13 |
> > | Qwen3-235B-A22B | 40.88 | 23 | 56.84 | 21 | 95.12 | 4 |
> > | Qwen3-30B-A3B | 41.10 | 22 | 56.46 | 22 | 81.64 | 20 |
> > | Qwen3-32B | 41.40 | 20 | 57.74 | 17 | 90.10 | 9 |
> > | o3-mini (Med) | 40.60 | 24 | 56.24 | 23 | 84.26 | 18 |
> > | o4-mini (Med) | 41.14 | 21 | 57.70 | 18 | 84.38 | 17 |

---

> > > ### Author Response · Authors · 2025-11-22
> > > **Response to Reviewer JQn4 (3)**
> > >
> > > From these tables, we can observe that while different judging models exert a certain impact on the result rankings, the three models generally maintain a certain degree of consistency. For example, on code review, the Kendall's W coefficient is 0.4936, indicating moderate consistency. We have updated the relevant results and analysis in the paper and updated the results in the main table to improve the reliability of the evaluation.
> > >
> > >
> > > ---
> > > >[Q1] Could the authors add statistical significance when presenting results?
> > > ---
> > >
> > > Thanks for your question. In fact, as an evaluation paper, since statistical significance is a pair-wise metric, it is difficult to provide them for all models and it is rare to show this in previous evaluation/evaluation work. However, we want to clarify that although without this, our results are also reliable and convincing. First, we use three distinct prompts to conduct the experiments three times and report the average performance, which mitigates random noise to a large extent. Second, we set the temperature to 0 during inference to avoid the impact of model fluctuations. Finally, each task has a sufficient sample size, meaning the model must correctly solve multiple questions to achieve improved performance. Thus, we believe that our results are reliable and convincing.
> > >
> > >
> > >
> > > ---
> > > >[Q2] What is the actual performance changes across prompts?
> > > ---
> > >
> > >
> > > Thanks for your question. We add the detailed results of each prompt in the appendix. Here we present the results on vulnerability detection. As can be seen from this table, the performance of different models under various prompts varies significantly not only in terms of ranking but also in their actual numerical performance. This highlights the importance of employing multiple prompts to provide a more comprehensive and reliable evaluation of model capabilities.
> > >
> > > | Model | Prompt 1 F1 | Prompt 2 F1 | Prompt 3 F1 |
> > > |------|-------------|-------------|-------------|
> > > | Grok-3-Mini (High) | 0.5455 | 0.5882 | 0.5778 |
> > > | GPT-4o | 0.6746 | 0.7104 | 0.6721 |
> > > | DeepSeek-R1 (0528) | 0.5158 | 0.5797 | 0.5972 |
> > > | Claude-3.5-Sonnet-20241022 | 0.5299 | 0.6332 | 0.5680 |
> > > | Qwen2.5-72B-Instruct | 0.3597 | 0.0721 | 0.2034 |
> > > | o4-mini (Med) | 0.4348 | 0.4762 | 0.4810 |
> > > | Gemma-3-27B-Instruct | 0.6444 | 0.6842 | 0.7054 |
> > > | LLaMA-3.1-70B-Instruct | 0.6950 | 0.6863 | 0.6692 |
> > > | GPT-3.5-turbo | 0.1138 | 0.2432 | 0.0714 |
> > > | GPT-4.1 | 0.6457 | 0.6220 | 0.5787 |
> > > | Claude-3.7-Sonnet | 0.4906 | 0.5989 | 0.6011 |
> > > | LLaMA-4-Scout-17B-16E-Instruct | 0.2519 | 0.1774 | 0.1639 |
> > > | Qwen3-235B-A22B | 0.5701 | 0.5714 | 0.6047 |
> > > | DeepSeek-R1 | 0.6099 | 0.6293 | 0.6071 |
> > > | Qwen3-32B | 0.4649 | 0.5492 | 0.5079 |
> > > | GPT-4-turbo | 0.7055 | 0.6870 | 0.7045 |
> > > | LLaMA-3.3-70B-Instruct | 0.7020 | 0.6214 | 0.6781 |
> > > | Claude-3.5-Haiku | 0.6320 | 0.7111 | 0.7099 |
> > > | Qwen3-30B-A3B | 0.4713 | 0.4151 | 0.3421 |
> > > | GPT-5 | 0.6981 | 0.6944 | 0.6827 |
> > > | o3-mini (Med) | 0.1653 | 0.2047 | 0.2636 |
> > > | Claude-Sonnet-4 | 0.7225 | 0.7289 | 0.7572 |
> > > | Gemini-2.5-Pro-05-06 | 0.6787 | 0.6875 | 0.6760 |
> > > | LLaMA-3.1-8B-Instruct | 0.2462 | 0.4852 | 0.5165 |
> > > | Qwen2.5-Coder-32B-Instruct | 0.1538 | 0.2656 | 0.1983 |
> > > | DeepSeek-V3 | 0.2222 | 0.2815 | 0.2500 |

---

> > > > ### Author Response · Authors · 2025-11-22
> > > > **Response to Reviewer JQn4 (4)**
> > > >
> > > > ---
> > > > >[Q3] There lacks sufficient literature review on robustness in machine learning for code and LLM in general.
> > > > Following your suggestion, we have updated the related work section (Appendix D.4: ROBUSTNESS OF CODE LLMS) to include introduction with recent code LLM robustness. The following is newly added part and is highlighted with blue:
> > > > ---
> > > >
> > > > Robustness in Code LLMs has become an important research question, as models that excel on standard benchmarks like HumanEval and MBPP often degrade sharply when exposed to real-world variations such as semantically-preserving code transformations and natural-language prompt perturbations (Mastropaolo et al., 2023; Zhuo et al., 2023). ReCode (Wang et al., 2023a) introduced the first robustness benchmark for code generation, applying 12 functionality-preserving perturbations including variable renaming, unused code insertion, and control-flow flattening. CCTest (Li et al., 2023b) focuses on real-world code completion tasks, efficiently testing and fixing inconsistency bugs in real products including Github copilot. NLPerturbator (Chen et al., 2024c) shifted focus to natural-language prompt variations, categorizing different real-world perturbation types derived from practitioner surveys and showing average pass@1 drops of 4.8–6.1% across StarCoder, CodeLlama, and DeepSeek-Coder. RobGen (Li et al., 2025) revealed that 35% of LLM-generated code is less robust than human references due to missing conditional checks and proposed a lightweight decoding-time framework that boosts robustness by 10% while preserving functional correctness. RobuNFR (Lin et al., 2025) extended evaluation to non-functional requirements including design, readability, reliability, performance, demonstrating that expressing the same NFR differently causes high output variability and up to 39% correctness loss. Recent work CodeCrash (Lam et al., 2025) comprehensively test LLMs in code reasoning under structural and NL-embedded perturbations. To mitigate this problem, many techniques such as structure-aware model training and robustness training are also introduced to improve code LLM’s robustness (Tipirneni et al., 2024; Pei et al., 2022; Oh & Yoo, 2024).
> > > >
> > > > [1] Detecting Pretraining Data from Large Language Models

---

### Official Review · Reviewer_2H6n · 2025-11-02

**Soundness:** 2
**Presentation:** 2
**Contribution:** 2
**Rating:** 2
**Confidence:** 4

**Summary:**

This paper presents a comprehensive evaluation framework for assessing the holistic performance of code LLMs. The framework integrates multiple code-related tasks, including code generation, code review, test generation, and code summarization, and extends to multi-language and multi-modality settings. It also incorporates robustness evaluation by applying semantically-preserving perturbations to the inputs. Experimental results reveal that existing large language models have notable robustness issues in coding tasks.

**Strengths:**

- The framework is comprehensive, covering diverse coding-related tasks, multiple programming languages, and multimodal settings. This provides a broad and unified view of code LLM capabilities.
- The inclusion of robustness evaluation through semantically-preserving perturbations is practical and relevant. The finding that existing models degrade significantly under prompt perturbations is insightful and highlights an important real-world limitation.

**Weaknesses:**

- The size and scale of each sub-benchmark are not clearly reported, making it difficult to assess coverage and statistical significance.

- When enhancing the prompt diversity, the prompt diversification process relies on manual validation, which limits scalability and reproducibility.

- On certain tasks such as Code Review, the evaluation results fail to effectively distinguish between models of different sizes or architectures, suggesting limited sensitivity of the metric or dataset.

- The contribution appears incremental. The framework largely aggregates existing datasets from prior work without introducing substantial new task design, annotation, or evaluation methodology.

**Questions:**

Many datasets are sourced from public websites, which are also used in the pretraining data of large code models. How does the paper ensure that evaluation data are not contaminated by training overlap?

---

> ### Author Response · Authors · 2025-11-22
> **Response to Reviewer 2H6n**
>
> Thank you for your hard work reviewing! We appreciate your recognition of our framework’s comprehensiveness and the practical value of robustness evaluation.Your comments have helped us identify critical areas for improvement, and we address each concern below.
>
> ---
> >[W1] The size and scale of each sub-benchmark are not clearly reported.
> ---
>
> Thanks for your comments. Following your suggestion, we add this in the paper. The size of each sub-benchmark is shown as follows:
> | Task                  | Code Generation | Code Summarization | Code Translation | Code Reasoning | Code Review | Test Generation | Vulnerability Detection | Multi-modal |
> |-----------------------|----------------|-------------------|-----------------|---------------|-------------|----------------|----------------------|-------------|
> | Data Volume           | 1664            | 2000              | 744             | 2000          | 2000        | 200            | 400                  | 900         |
>
>
> ---
> >[W2] When enhancing the prompt diversity, the prompt diversification process relies on manual validation, which limits scalability and reproducibility.
> ---
>
> Thanks for your comments. We want to clarify that the choice of manual validation for prompts is based on the following points:
> 1. Manual validation ensures **prompt quality and reliability which is critical for evaluation**, as ambiguous prompts can distort model performance.
> 2. This is a popular and widely accepted practice in existing works and aligns with prior benchmark work [1,2,3].
> 2. We believe that scalability of the used prompt is not an important factor for a benchmark, as expanding the dataset only requires adding new task instances and does not require adding new prompts, so manual prompt validation is a **one-time effort**.
> 4. For reproducibility, we have made all used prompts publicly available in the appendix, allowing other researchers to replicate our evaluation.
>
>
> ---
> >[W3] On certain tasks such as Code Review, the evaluation results fail to effectively distinguish between models of different sizes or architectures.
> ---
>
> Thanks for your comments. We use LLM-as-a-judge to evaluate LLMs’ performance on code review as it is a most advanced and widely used approach in this task [4,5,6]. As for limited distinction between models, we believe it reflects the challenge in code review tasks: Most LLMs struggle to effectively understand code diffs and provide actionable feedback. This aligns with recent findings in the field [7,8] where no significant differences exist between large models and small models and their absolute values are relatively low.
>
>
> ---
> >[W4] The contribution appears incremental. The framework largely aggregates existing datasets.
> ---
>
> Thanks for your question, we want to first clarify that TREAT is **not merely an aggregation of existing datasets**. Instead, we collected new data for most tasks, and only reused a small portion of existing datasets for tasks where automatic annotation is challenging. In total, **81.6%** of TREAT’s data is newly collected or created by us, while only **18.4%** is aggregated from existing datasets (400 samples from PRIMEVUL, 200 from CODAMOSA, 900 from DESIGNBENCH, 328 from PolyHumanEval).
> Besides, we believe that reusing a subset of existing data to ensure comprehensive evaluation is a **common and widely accepted practice in existing works** [1,2,3]. Similar to those general-domain LLM benchmarks, TREAT is the **first work** to conduct a comprehensive evaluation of code LLMs across the entire software development lifecycle. This holistic evaluation reveals novel findings such as the substantial performance variation across tasks. We believe this cross-task, lifecycle-oriented benchmark fills a critical gap in code LLM evaluation and contributes valuable insights to the community.

---

> > ### Author Response · Authors · 2025-11-22
> > **Response to Reviewer 2H6n (2)**
> >
> > ---
> > >[Q1]  Many datasets are sourced from public websites, which are also used in the pretraining data of large code models. How does the paper ensure that evaluation data are not contaminated?
> > ---
> >
> > Thanks for your comment. Data contamination is a widespread challenge in benchmarking internet-scale LLMs, especially since most models do not disclose their training data for auditing. To mitigate this problem, for most tasks, we make the data collection and process stage an automated pipeline and plan to make the benchmark live, which could avoid the data contamination problem. Besides, to validate the contamination probability of current data, we adopted the widely-used min-k [9] method to detect potential contamination. We used Qwen3-30B-A3B for detection given its recent release (May 2025) and open-source nature, which enables access to the model's output probabilities.  The results are shown in the following table:
> >
> > | Task | Code Generation   | Code Summuration   | Code Translation  | Code Review | Code Reasoning | Vulnerability Detection | Test  Generation |
> > |-------|-------|-------|--------|--------|-----------|------------|--------|
> > |Min-k| -7.72 | -7.54 | -4.68  | -7.38  | -6.62     | -6.60      | -6.17  |
> >
> > We can find that all the results indicate that even recent-released models did not exhibit data leakage on our dataset. We have added this analysis to the appendix to improve research transparency.
> >
> >
> >
> >
> > [1] ClassEval: A Manually-Crafted Benchmark for Evaluating LLMs on Class-level Code Generation; [2] MMLU-Pro: A more robust and challenging multi-task language understanding benchmark
> > [3] GPQA: A graduate-level google-proof q&a benchmark
> > [4] Deep assessment of code review generation approaches: Beyond lexical similarity
> > [5] On the Effectiveness of LLM-as-a-Judge for Code Generation and Automated Code Review
> > [6] CRScore: Grounding Automated Evaluation of Code Review Comments with LLM-as-a-Judge
> > [7] Benchmarking and Studying the LLM-based Code Review
> > [8] Benchmarking LLMs for Fine-Grained Code Review with Enriched Context in Practice
> > [9] Detecting Pretraining Data from Large Language Models

---

> > > ### Comment · Reviewer_2H6n · 2025-11-25
> > > **Response to the authors**
> > >
> > > Thanks for the clarification.
> > > 1.  I acknowledge that many data instances were newly collected. However, my concern remains that the task and data collection methods closely mirror prior work. In the paper, the data collection section largely follows existing task definitions and established annotation pipelines to new data sources, with limited new task design or methodological innovation. Thus, even with new data points, the overall contribution still feels incremental, unless clearer evidence of novel task design or annotation methodology is provided.
> > > 2. About the code review task performance: While I understand the authors’ interpretation that the uniformly low scores across models indicate that the dataset is inherently challenging, I am not fully convinced that this conclusion is sufficiently supported by the presented evidence. When models with substantially different sizes and capabilities, including frontier-level LLMs (e.g., GPT-5 and Claude Sonnet-4) and much smaller models (e.g., Llama-3.1-8B, Qwen3-30B-A3B), all converge to similar performance, a natural alternative explanation is that the dataset or metric lacks discriminative power. To support the claim that the dataset is challenging rather than insensitive, additional evidence would be needed. For example, human performance, qualitative error analysis, or demonstrations that more capable models produce meaningfully better outputs even if the metric fails to capture it.

---

### Official Review · Reviewer_RBNc · 2025-11-02

**Soundness:** 3
**Presentation:** 3
**Contribution:** 3
**Rating:** 6
**Confidence:** 3

**Summary:**

The paper introduces TREAT, a framework for holistic evaluation of coding capabilities of LLMs, covering different coding tasks (like generation, summarization, test generation, etc), languages, multi-modality and robustness. They benchmarked this framework across different LLMs and noticed that no one model excels at all the tasks, while also showing that these LLMs are not robust to semantic code perturbations.

**Strengths:**

- Holistic benchmark with coverage across different tasks, languages, containing multimodal and robustness assessments.
- Section 5.3 and 5.4 present interesting findings, where models with thinking exhibit better robustness to code perturbations and that model's evaluation results are sensitive to changes in prompt.

**Weaknesses:**

- Using GPT-4o as the only LLM judge may bias the scores towards GPT* models, for tasks beyond code correctness. Why did the authors not think of an ensemble based ranking?
- In page 2, the authors mentioned "Current state-of-the-art models exhibit substantial performance variation and specialization across
different programming tasks" to be one of the novel findings. However, LiveCodeBench paper also discusses similar findings in Figure 4 (https://arxiv.org/pdf/2403.07974). This seems to be a misrepresentation.

**Questions:**

- Why is only GPT4o used as an LLM Judge?

---

> ### Author Response · Authors · 2025-11-22
> **Response to Reviewer RBNc**
>
> Thank you for your hard work reviewing! We appreciate that you provided us with such thoughtful feedback. For your concerns, we will address them one by one.
>
> ---
> > [W1/Q1] Using GPT-4o as the only LLM judge may bias the scores towards GPT* models, for tasks beyond code correctness. Why did the authors not think of an ensemble based ranking?
> ---
>
> Thanks for your suggestion, to mitigate this issue, we have added two additional models Gemini-2.5-Flash and Claude-4-Sonnet from different model families as alternative judges for code summarization and code review tasks. The detailed comparison results are shown in the following table:
>
> ### Code Review
> | Model                          | Gemini-2.5-Flash | Rank | Claude-4 | Rank | GPT-4o  | Rank |
> |--------------------------------|------------------|------|----------|------|---------|------|
> | Claude-3.5-Haiku-20241022      | 32.20            | 16   | 39.58    | 17   | 30.56   | 12   |
> | Claude-3.5-Sonnet-20241022     | 34.16            | 2    | 39.54    | 18   | 29.98   | 20   |
> | Claude-3.7-Sonnet              | 33.46            | 3    | 40.38    | 8    | 30.48   | 14   |
> | Claude-Sonnet-4                | 32.84            | 6    | 41.04    | 3    | 31.04   | 8    |
> | DeepSeek-R1                    | 32.26            | 15   | 40.48    | 7    | 27.38   | 25   |
> | DeepSeek-R1 (0528)             | 32.38            | 11   | 41.12    | 2    | 31.30   | 5    |
> | DeepSeek-V3                    | 32.56            | 9    | 39.60    | 16   | 30.52   | 13   |
> | GPT-3.5-turbo-0125             | 29.60            | 26   | 34.72    | 26   | 29.64   | 22   |
> | GPT-4-turbo-2024-04-09         | 32.30            | 13   | 39.36    | 22   | 29.70   | 21   |
> | GPT-4.1-2025-04-14             | 32.92            | 5    | 40.80    | 4    | 29.38   | 23   |
> | GPT-4o-2024-11-20              | 31.46            | 21   | 39.42    | 20   | 30.46   | 15   |
> | GPT-5                          | 29.98            | 25   | 42.78    | 1    | 26.62   | 26   |
> | Gemini-2.5-Pro-05-06           | 32.68            | 8    | 40.30    | 11   | 31.46   | 3    |
> | Gemma-3-27B-Instruct           | 32.72            | 7    | 40.62    | 6    | 31.74   | 1    |
> | Grok-3-Mini (High)             | 34.54            | 1    | 40.34    | 10   | 30.90   | 10   |
> | Llama-3.1-70B-Instruct         | 31.32            | 22   | 38.74    | 24   | 30.14   | 19   |
> | Llama-3.1-8B-Instruct          | 30.42            | 24   | 37.36    | 25   | 30.20   | 18   |
> | Llama-3.3-70B-Instruct         | 32.34            | 12   | 38.80    | 23   | 30.70   | 11   |
> | Llama-4-Scout-17B-16E-Instruct | 32.30            | 13   | 39.66    | 15   | 30.36   | 16   |
> | Qwen2.5-72B-Instruct           | 32.50            | 10   | 39.40    | 21   | 31.42   | 4    |
> | Qwen2.5-Coder-32B-Instruct     | 31.80            | 20   | 39.52    | 19   | 31.22   | 6    |
> | Qwen3-235B-A22B                | 32.14            | 17   | 40.38    | 8    | 31.10   | 7    |
> | Qwen3-30B-A3B                  | 31.84            | 19   | 40.30    | 11   | 31.70   | 2    |
> | Qwen3-32B                      | 32.10            | 18   | 40.28    | 13   | 30.36   | 16   |
> | o3-mini (Med)                  | 33.10            | 4    | 39.68    | 14   | 31.00   | 9    |
> | o4-mini (Med)                  | 30.60            | 23   | 40.76    | 5    | 29.02   | 24   |
>
> ### Code Summarization
> | Model | Gemini-2.5-Flash | Rank | Claude-4 | Rank | GPT-4o | Rank |
> |---|---|---|---|---|---|---|
> | Claude-3.5-Haiku-20241022 | 41.82 | 19 | 57.82 | 16 | 85.24 | 15 |
> | Claude-3.5-Sonnet-20241022 | 42.74 | 12 | 59.20 | 4 | 96.54 | 2 |
> | Claude-3.7-Sonnet | 43.44 | 1 | 59.62 | 3 | 88.10 | 11 |
> | Claude-Sonnet-4 | 43.44 | 1 | 60.38 | 1 | 93.76 | 5 |
> | DeepSeek-R1 | 42.04 | 17 | 58.72 | 7 | 90.64 | 7 |
> | DeepSeek-R1 (0528) | 41.98 | 18 | 58.72 | 7 | 90.64 | 7 |
> | DeepSeek-V3 | 42.30 | 15 | 57.86 | 14 | 92.82 | 6 |
> | GPT-3.5-turbo-0125 | 42.66 | 13 | 55.00 | 25 | 71.18 | 24 |
> | GPT-4-turbo-2024-04-09 | 42.28 | 16 | 57.30 | 19 | 89.94 | 10 |
> | GPT-4.1-2025-04-14 | 42.42 | 14 | 57.30 | 19 | 80.28 | 21 |
> | GPT-4o-2024-11-20 | 42.82 | 9 | 57.84 | 15 | 87.88 | 12 |
> | GPT-5 | 40.50 | 25 | 58.34 | 9 | 98.28 | 1 |
> | Gemini-2.5-Pro-Preview-05-06 | 43.16 | 4 | 58.76 | 6 | 78.88 | 22 |
> | Gemma-3-27B-Instruct | 43.12 | 5 | 57.96 | 13 | 82.96 | 19 |
> | Grok-3-Mini (High) | 42.82 | 9 | 59.64 | 2 | 85.10 | 16 |
> | Llama-3.1-70B-Instruct | 42.86 | 6 | 58.32 | 10 | 74.52 | 23 |
> | Llama-3.1-8B-Instruct | 42.84 | 8 | 55.90 | 24 | 64.20 | 25 |
> | Llama-3.3-70B-Instruct | 42.80 | 11 | 58.96 | 5 | 96.00 | 3 |
> | Qwen2.5-72B-Instruct | 43.28 | 3 | 58.02 | 11 | 86.54 | 14 |
> | Qwen2.5-Coder-32B-Instruct | 42.86 | 6 | 58.02 | 11 | 86.86 | 13 |
> | Qwen3-235B-A22B | 40.88 | 23 | 56.84 | 21 | 95.12 | 4 |
> | Qwen3-30B-A3B | 41.10 | 22 | 56.46 | 22 | 81.64 | 20 |
> | Qwen3-32B | 41.40 | 20 | 57.74 | 17 | 90.10 | 9 |
> | o3-mini (Med) | 40.60 | 24 | 56.24 | 23 | 84.26 | 18 |
> | o4-mini (Med) | 41.14 | 21 | 57.70 | 18 | 84.38 | 17 |

---

> ### Author Response · Authors · 2025-11-22
> **Response to Reviewer RBNc (2)**
>
> From these tables, we can observe that while different judging models exert a certain impact on the result rankings, the three models generally maintain a certain degree of consistency. For example, on code review, the Kendall's W coefficient is 0.4936, indicating moderate consistency. We have updated the relevant results and analysis in the paper and updated the results in the main table to improve the reliability of the evaluation.
>
>
> ---
> > [W2] The claim that "Current state-of-the-art models exhibit substantial performance variation and specialization across different programming tasks" is similar to findings in LiveCodeBench, leading to misrepresentation.
> ---
>
> We appreciate your critical observation and have conducted a quantitative comparison to clarify the differences of findings between our work and LiveCodeBench. We used the **rank variance** (standard deviation of model rankings across tasks) as a metric to measure performance variation, and the results are as follows:
>
> ## Results in LiveCodeBench
> | Results in LiveCodeBench| GPT-4-Turbo | Claude3-O | GPT-4 | Gemini-Pro-1.5 | Gemini-Flash-1.5 | Mistral-L | LLama-ins-70B | Claude-3-S | DS-ins-33B | Phind-34B | GPT-3.5-Turbo | DS-ins-6.7B |
> |-------------------------|-------------|-----------|-------|----------------|------------------|-----------|---------------|------------|------------|-----------|---------------|-------------|
> | Average Rank            | 1.25        | 2.00      | 3.50  | 3.50           | 5.75             | 6.50      | 6.75          | 7.00       | 9.75       | 10.00      | 10.00         | 11.75       |
> | Rank Variance           | 0.50        | 0.82      | 0.58  | 1.29           | 0.96             | 1.73      | 0.96          | 1.15       | 0.50       | 1.00      | 1.15         | 0.50        |
>
> ## Results in TREAT
> | Model | Average Rank | Rank Std Dev |
> |-------|--------------|--------------|
> | GPT-5 | 5.0 | 8.45 |
> | Claude-Sonnet-4 | 5.0 | 3.65 |
> | Claude-3.7-Sonnet | 8.4 | 3.99 |
> | DeepSeek-R1 (0528) | 8.7 | 3.73 |
> | o3-mini | 9.3 | 8.40 |
> | GPT-4.1 | 9.4 | 6.21 |
> | Qwen3-235B-A22B | 10.4 | 3.60 |
> | o4-mini | 10.4 | 8.10 |
> | Grok-3-Mini | 10.4 | 7.85 |
> | DeepSeek-R1 | 11.1 | 6.64 |
> | GPT-4o | 12.0 | 4.36 |
> | Claude-3.5-Sonnet | 12.3 | 7.91 |
> | DeepSeek-V3 | 12.6 | 4.79 |
> | Gemini-2.5-Pro | 12.6 | 9.07 |
> | Qwen3-32B | 13.3 | 4.03 |
> | Qwen3-30B-A3B | 14.4 | 6.13 |
> | GPT-4-turbo | 14.4 | 4.39 |
> | LLaMA-3.3-70B | 15.0 | 9.07 |
> | Gemma-3-27B | 16.1 | 9.30 |
> | Qwen2.5-72B | 16.7 | 4.54 |
> | Qwen2.5-Coder-32B | 17.1 | 2.54 |
> | Claude-3.5-Haiku | 17.9 | 6.23 |
> | LLaMA-4-Scout | 19.7 | 5.28 |
> | LLaMA-3.1-70B | 20.4 | 5.22 |
> | GPT-3.5-turbo | 23.0 | 4.58 |
> | LLaMA-3.1-8B | 24.0 | 4.04 |
>
>
>
> As shown in the tables, most models in LiveCodeBench have a rank variance of less than 1, indicating only minor performance variation across tasks. In contrast, models in TREAT exhibit significantly larger variance (ranging from 2.5 to 9.3), especially on tasks not covered by LiveCodeBench such as code review and vulnerability detection. To address the misrepresentation concern, we have revised the statement in the paper to be more rigorous: "Current state-of-the-art models exhibit substantial performance variation and specialization across different programming tasks, especially on tasks like code review and vulnerability detection."

---

### Meta-Review · Area_Chair_tiLW · 2026-01-05

**Summary:**

Reviewers unanimously agree that the goal of this paper is valuable: proposing a comprehensive framework for evaluating LLM coding capabilities, covering multiple tasks, programming languages, multimodal scenarios, and robustness tests. The paper features a large-scale experimental design (evaluating 26 models) and offers insightful findings (e.g., poor robustness under prompt perturbations, and models with explicit "thinking" processes being more robust).

Reviewers also acknowledge that the paper covers tasks spanning the entire software engineering lifecycle. The experimental results are comprehensive, providing extensive empirical data. Finally, the evaluation of perturbations is meaningful.

Reviewers have raised the following primary concerns:

1. The benchmark collection appears more like a broad aggregation from diverse sources, lacking strict standards for data decontamination and inclusion control, which undermines reviewer confidence regarding contamination issues.

2. The use of LLM-as-a-judge lacks rigorous criteria and reproducibility, necessitating more stringent quantitative metrics for evaluation.

**Reviewer Concerns:**

In response, the authors have supplemented the paper with extensive experiments, partially addressing some of the reviewers' concerns. However, they have not taken effective concrete actions to resolve concerns regarding benchmark collection and decontamination controls.

**Reviewer Scores:**

While appreciative of the additional experiments, reviewers maintain that the core concerns about benchmark collection and decontamination remain unresolved. As a result, they are inclined to retain their current negative evaluations.

---

### Decision · Program_Chairs · 2026-01-26

Reject